# Building compositional tasks with shared neural subspaces

Sina Tafazoli[1✉], Flora M. Bouchacourt[1], Adel Ardalan[1], Nikola T. Markov[1], Motoaki Uchimura[1], Marcelo G. Mattar[2], Nathaniel D. Daw[1,3] & Timothy J. Buschman[1,3✉]

Cognition is highly flexible—we perform many different tasks[1] and continually adapt our behaviour to changing demands[2,3]. Artificial neural networks trained to perform multiple tasks will reuse representations[4] and computational components[5] across tasks. By composing tasks from these subcomponents, an agent can flexibly switch between tasks and rapidly learn new tasks[6,7]. Yet, whether such compositionality is found in the brain is unclear. Here we show the same subspaces of neural activity represent task-relevant information across multiple tasks, with each task flexibly engaging these subspaces in a task-specific manner. We trained monkeys to switch between three compositionally related tasks. In neural recordings, we found that task-relevant information about stimulus features and motor actions were represented in subspaces of neural activity that were shared across tasks. When monkeys performed a task, neural representations in the relevant shared sensory subspace were transformed to the relevant shared motor subspace. Monkeys adapted to changes in the task by iteratively updating their internal belief about the current task and then, based on this belief, flexibly engaging the shared sensory and motor subspaces relevant to the task. In summary, our findings suggest that the brain can flexibly perform multiple tasks by compositionally combining task-relevant neural representations.

Humans and other animals can combine simple behaviours to create more complex behaviours[8–11]. For example, learning to discriminate whether a piece of fruit is ripe can be used as a component of a variety of foraging, cooking and eating tasks. The ability to compositionally combine behaviours is thought to be central to generalized intelligence in humans[12] and a necessary component for artificial neural networks to achieve human-level intelligence[13–15]. When artificial neural networks are trained to perform multiple tasks, they reuse representations and computational components in different tasks[4–6]. Whether the brain similarly reuses sensory, cognitive and/or motor representations across tasks remains unclear. Furthermore, we do not yet understand how the brain flexibly engages these representations to continually adapt to the changing demands of the environment[16]. To address these questions, we trained two monkeys to flexibly switch between three different compositionally related tasks.

## Composing tasks from subtasks

All three tasks followed the same general structure: the monkeys were presented with a visual stimulus and had to indicate its category with an eye movement (Fig. 1a). The stimuli were parametric morphs, independently varying in both shape and colour (Fig. 1b and Methods). The monkeys performed three categorization tasks. In the shape–axis 1 (S1) task, they categorized the shape of the stimulus and then responded on axis 1: when the shape was more similar to a 'bunny', the monkey made a saccade to the upper-left (UL) target, and when the shape

was more similar to a 'tee', the monkey saccaded to the lower-right (LR) target (Fig. 1c (top row)). The colour–axis 2 (C2) task required the monkey to categorize the colour of the stimulus and respond on axis 2: if the colour was 'red', they saccaded to upper-right (UR), and if 'green', they saccaded to lower-left (LL; Fig. 1c (bottom row)). Finally, in the colour–axis 1 (C1) task, the monkeys categorized the colour of the stimulus (as in the C2 task) and responded on axis 1 (as in the S1 task; red = LR, green = UL; Fig. 1c (middle row)). In this way, the tasks were compositionally related—C1 can be considered as combining the colour categorization subtask of C2 with the motor response subtask of S1.

Both monkeys performed all three tasks well (Fig. 1e; monkey Si/Ch: S1, 81%/77%; C1, 83%/78%; C2, 92%/92%; all $P < 0.001$, binomial test, the performance of individuals is shown in Extended Data Fig. 1). The monkeys performed the same task for a block of trials (Fig. 1d). When they reached a behavioural criterion (performance ≥ 70%), the task would change (Methods). The monkeys were not instructed as to the identity of the new task. Thus, they had to learn which task was in effect on each new block. Note that this learning was not de novo but reflected a process of discovering the current task (also known as apparent learning[17] or task inference[18]). This was reflected in the monkey's behaviour, which improved over the first 75 trials of the S1 and C1 tasks (Fig. 1f (left); S1/C1 increased from 0.47/0.62 at trial 15 to 0.71/0.77 at trial 75; both $P < 0.001$, $\chi^2$ test). Despite performing both tasks well, the monkeys were initially biased towards the C1 task (Extended Data Fig. 2e,f; performance difference for first 15 trials was 15.1%, $P = 3.3 \times 10^{-16}$,

[1]Princeton Neuroscience Institute, Princeton University, Princeton, NJ, USA. [2]Department of Psychology, New York University, New York, NY, USA. [3]Department of Psychology, Princeton University, Princeton, NJ, USA. ✉e-mail: tafazoli@princeton.edu; tbuschma@princeton.edu

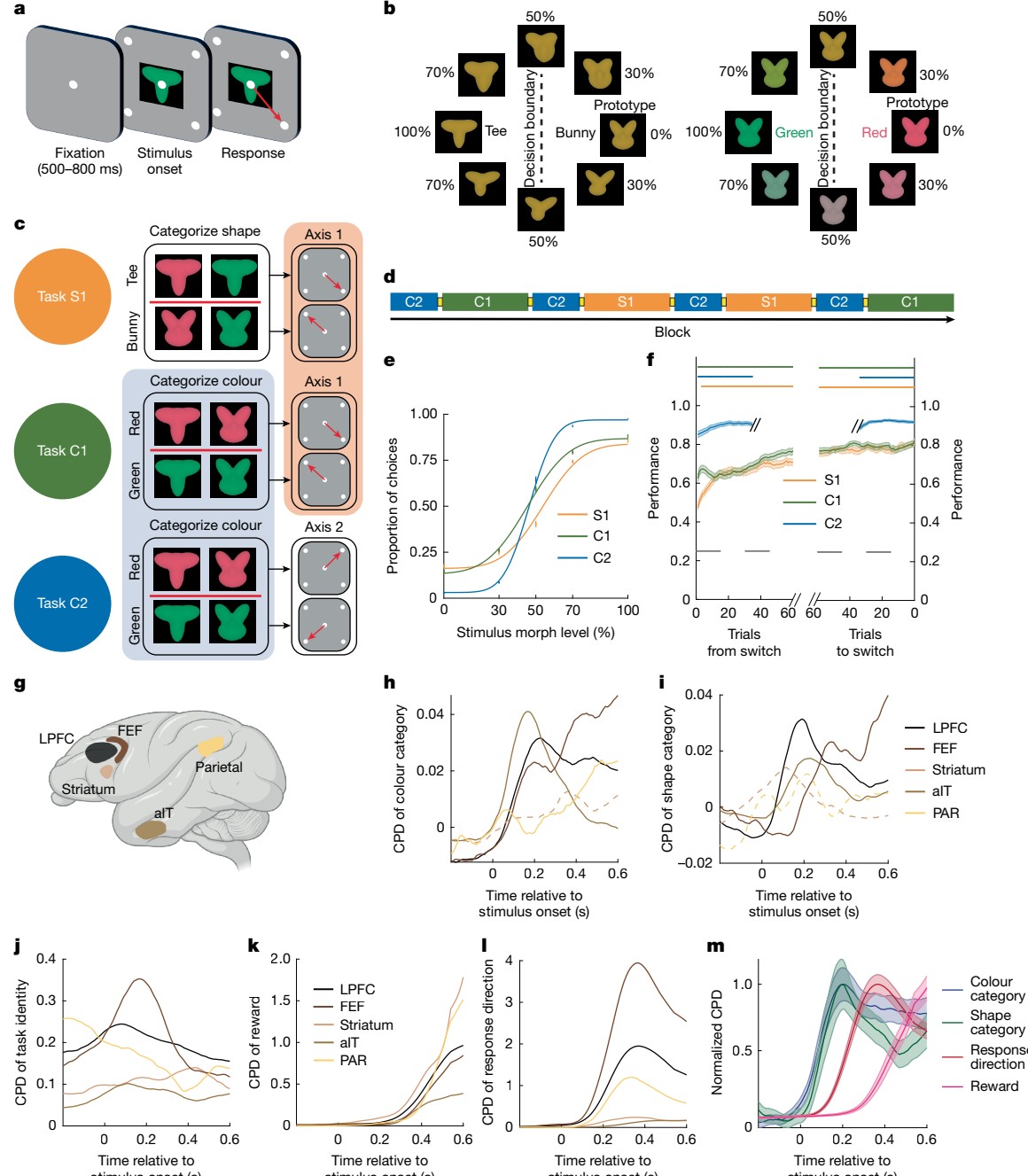

**Fig. 1 | The monkeys performed three compositional tasks. a**, Schematic of the task timeline. After fixation, a visual stimulus and four response targets were presented. The monkeys reported the stimulus category by saccading to one of the targets. **b**, Stimuli were morphs in a two-dimensional feature space, independently varying in shape (left) and colour (right). Categories are indicated by dashed lines and labels. **c**, Schematic of the task design. Task S1 required categorizing stimuli by shape and responding on axis 1. Task C2 required categorizing by colour and responding on axis 2. Task C1 required categorizing by colour and responding on axis 1. The coloured backgrounds highlight shared subtasks: colour categorization for C1 and C2 (blue) and response axis for S1 and C1 (orange). **d**, Example sequence of tasks. The task switched when performance was ≥70%. The monkeys were not cued to the upcoming task, although the response axis always switched between blocks. Recordings included 94, 97 and 189 blocks of S1, C1 and C2. **e,f**, Behavioural performance. **e**, Psychometric curve

for both animals for the 102, 102 and 51 trials before the switch for S1, C1 and C2 tasks, respectively. **f**, The performance before and after a switch in task. Data are mean ± s.e.m. The horizontal line indicates *P* < 0.001, as determined using a one-sided uncorrected binomial test. **g**, The locations of neural recordings. The diagram was created using BioRender. **h–l**, Information about task-relevant variables for all regions, estimated as the proportion of variance in each neuron's activity uniquely explained by each variable (using the coefficient of partial determination, CPD; Methods). The CPD for colour (**h**), shape (**i**), task identity (**j**), reward (**k**) and response direction (**l**) is shown. The lines show the mean CPD across neurons per region. The dashed lines indicate that a region did not have a significant number of neurons encoding that cognitive variable. **m**, Time course of the average CPD across all recorded neurons, normalized to the maximum value to show temporal order. Data are mean ± s.e.m.

$\chi^2$ test). This suggests that the monkeys initially expected to perform the C1 task, possibly because they categorized colour on two out of three tasks (C1 and C2).

The monkeys performed the C2 task accurately immediately after the switch (Fig. 1f; 0.85 at trial 15; *P* < 0.001, binomial test). This was because the axis of response always changed between blocks and the C2 task was

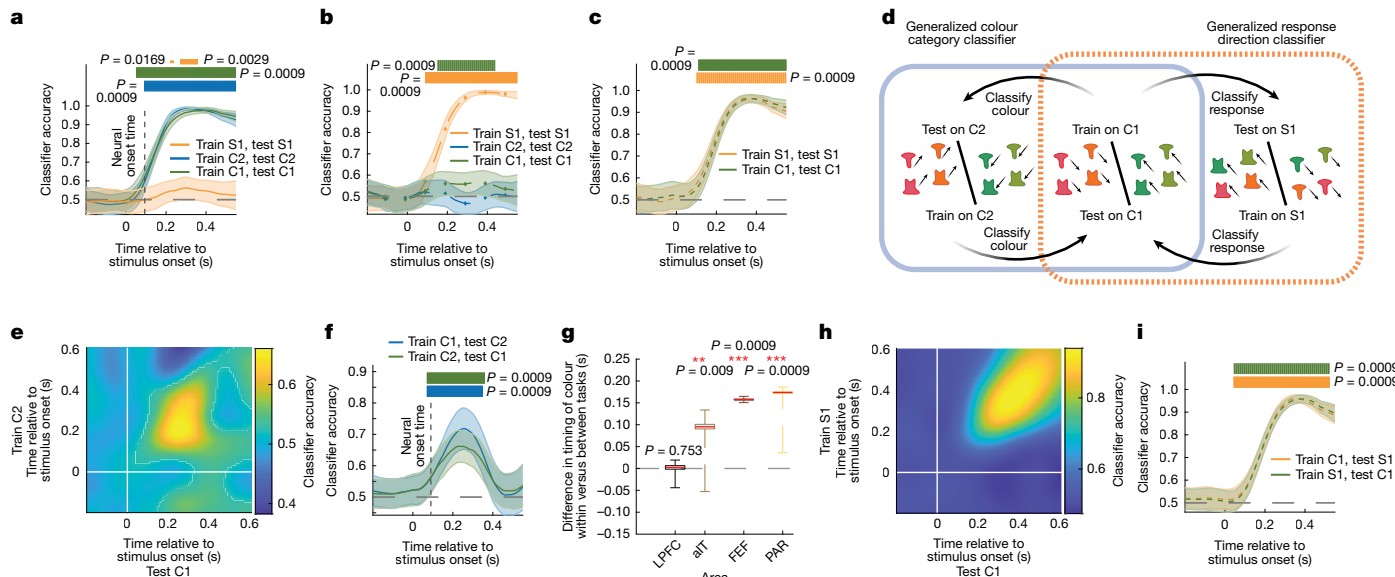

**Fig. 2 | Colour category and response representations were shared across tasks. a–c**, The accuracy of classifiers trained to decode colour category (**a**), shape category (**b**) and motor response (**c**) from LPFC neural activity. Classifiers were trained and tested on withheld trials of the S1 task (orange), C2 task (blue) and C1 task (green). Data are mean ± s.e.m. over 250 classifiers, trained and tested on $n$ = 40 randomly sampled trials (Methods). The horizontal bars indicate above-chance classification ($P < 0.05$, 0.01 and 0.001 for thin, medium and thick lines, respectively; two-sided permutation test, corrected for multiple comparisons over time). **d**, Schematic of the classifiers used to test whether colour category and response location information were shared across tasks (Methods). **e**, Cross-temporal cross-task classification accuracy of the colour classifier trained to decode the colour category from LPFC neural activity during the C2 task and tested on the C1 task. **f**, Cross-task classification accuracy of colour when the colour classifier was trained on the C1 task (blue) or C2 task (green) and then tested on the other task. Data are mean ± s.e.m. over 250 classifiers. **g**, Shared

colour information is delayed in the aIT, FEF and PAR. The difference in the time of onset of colour information during the C1 task when decoded from classifiers trained on the C1 task versus generalized from the C2 task (vertical dashed lines in **a** and **f**, respectively) is shown. Statistical analysis was performed using two-sided uncorrected bootstrap tests; *$P < 0.05$, **$P < 0.01$ and ***$P < 0.001$; LPFC ($P = 0.753$), aIT ($P = 0.0099$), FEF ($P = 0.0009$), PAR ($P = 0.0009$). The box limits represent the interquartile range (25th to 75th percentile), the red horizontal line indicates the median, and the whiskers indicate the full extent of data. **h**, Cross-temporal cross-task classification accuracy of the response classifier trained to decode response direction from LPFC neural activity during the S1 task and tested on the C1 task. **i**, Cross-task classification accuracy of response when the response classifier was trained on the C1 task (orange) or S1 task (green) and then tested on the other task. Data are mean ± s.e.m. over 250 classifiers.

the only one to use axis 2, reducing the monkeys' uncertainty about the task (and leading to more C2 blocks than C1 or S1 blocks, although there were more trials of C1 or S1 than C2; Methods). While overall behavioural performance was higher on C2 than C1 trials (Fig. 1e), this difference was probably due to the increased uncertainty about the task on C1 trials; there was no difference in performance for congruent trials or when the shape was ambiguous (Extended Data Fig. 2a–d). Together, the animals' behaviour suggests that they rapidly identified the change in response axis but slowly integrated feedback to learn whether S1 or C1 was in effect[19].

## Representations were shared across tasks

To understand the neural representations used during each task, we simultaneously recorded neural activity from five cortical and subcortical regions (Fig. 1g): the lateral prefrontal cortex (LPFC; 480 neurons), frontal eye fields (FEF; 149 neurons), parietal cortex (PAR; 64 neurons), anterior inferior temporal cortex (aIT; 239 neurons) and striatum (caudate nucleus; 149 neurons). All five regions represented task-relevant cognitive variables[20,21], including the identity of the current task, the colour and shape of the stimulus, the response direction and whether a reward was received (Fig. 1h–l). The identity of the task was consistently represented throughout the trial (Fig. 1j). After stimulus onset, information about the colour category[22] (Fig. 1h) and shape category of the stimulus (Fig. 1i) was followed by information about the direction of the animal's response (Fig. 1l), and then the reward received (Fig. 1k; see Fig. 1m and Extended Data Fig. 3a for the average time course for all regions). Individual neurons were

'mixed selective' in all regions, representing multiple task variables[23,24] (Extended Data Fig. 3b–k).

To understand how each task variable was represented in the neural population, we trained classifiers to decode the stimulus colour, shape and motor response from the pattern of neural activity in each region (neurons from all recording sessions were combined into a pseudopopulation[24]; Methods). Classifiers trained on LPFC neural activity accurately decoded the category of the stimulus' colour in the C1 and C2 tasks (Fig. 2a; 79 ms, 87 ms after stimulus onset for C1 and C2, respectively; similar results were seen when controlling for motor response; Extended Data Fig. 4a, $P < 0.001$, permutation test). In this way, the classifier defined a subspace within the high-dimensional space of neural activity that represented the colour category of the stimulus input. Similarly, classifiers trained on LPFC activity decoded the category of the stimulus' shape during the S1 task (Fig. 2b; 100 ms after the stimulus onset; $P < 0.001$, permutation test).

Task-irrelevant stimulus information was attenuated. In the LPFC, classification accuracy about the shape of the stimulus was reduced during the C1 and C2 tasks (Fig. 2b) and information about the colour of the stimulus was reduced during the S1 task (this was also true at the end of the block, when the animal was more certain about the identity of the task; Fig. 2a and Extended Data Fig. 4c,d). Similar to LPFC, other regions represented task-relevant, but not task-irrelevant, information (Extended Data Fig. 4e,f).

The direction of the monkey's response within each task's response axis could be decoded from the LPFC (Fig. 2c and Methods). Information about the response occurred after the stimulus' category (115 ms, 133 ms after stimulus onset in S1 and C1 tasks, respectively;

both $P < 0.001$, permutation test; see Extended Data Fig. 4b for the C2 task). Similar results were seen in other regions (Extended Data Fig. 4g). Together, these results show that the stimulus colour category, stimulus shape category and response direction are broadly represented at the population level in multiple recorded regions.

To test whether representational subspaces were reused across tasks, we quantified how well a classifier trained to decode the stimulus' colour category or the animals' motor response in one task generalized to the other tasks (Fig. 2d and Methods). Consistent with a shared representation in the LPFC, a classifier trained to decode the colour category of a stimulus during the C2 task was able to significantly decode the stimulus' colour category during the C1 task (65 ms after stimulus onset in the LPFC; $P < 0.001$, permutation test; similar results were seen when using a nonlinear classifier; Fig. 2e,f and Extended Data Fig. 5a). The reverse was also true: a classifier trained on C1 could decode the stimulus' colour category during the C2 task (Fig. 2f and Extended Data Fig. 5b, 84 ms after stimulus onset; $P < 0.001$, permutation test). Importantly, the two tasks required different motor responses and therefore the shared representation reflected the colour category of the stimulus and not the motor response (further controls for movement are shown in Extended Data Fig. 5c,d). Although colour information was reduced in the S1 task, the weak colour category representation that did exist also generalized between the C1 and S1 tasks (Extended Data Fig. 5e).

While colour category information was represented in a shared subspace in the LPFC, it was represented in task-specific subspaces in other brain regions (Extended Data Fig. 6a). Generalization was weaker in the FEF, PAR and aIT (Extended Data Fig. 6b,c) and was delayed with respect to task-specific sensory information (Fig. 2g and Extended Data Fig. 6d). There was no significant generalization in the STR (although the stimulus colour category could be decoded; Extended Data Fig. 4e).

Motor response representations also generalized across tasks. A classifier trained to decode response direction in the S1 task generalized to decode response direction in the C1 task (and vice versa; Fig. 2h,i, 128/128 ms after stimulus onset in the LPFC, when training on S1 or C1 tasks and testing on C1 or S1 tasks, respectively; $P < 0.001$, permutation test). In contrast to stimulus information, the motor representation was shared in all regions (Extended Data Fig. 4h; perhaps reflecting a widely broadcasted motor signal[25], but see ref. 26).

## Tasks sequentially used shared subspaces

If the representation of both the category of the stimulus and the motor response were shared across tasks then performing a task requires selectively transforming representations from one subspace to another[27]. Consistent with this, there was a sequential representation of the stimulus colour in the shared colour subspace followed by the motor response in the shared motor subspace during the C1 task (Fig. 3a; 63 ms difference in onset time of colour response using C2 classifier and motor response using S1; $P < 0.001$, $t$-test).

To directly test whether this reflected the transformation of information between subspaces, we tested whether information about the stimulus colour in the shared colour subspace predicted the response in the shared response subspace on a trial-by-trial basis (Fig. 3b). Figure 3c shows the correlation between the representation of the stimulus' colour category in the C1 task, decoded using the classifier from the C2 task, and the representation of the motor response in the C1 task, decoded using the response classifier from the S1 task (Methods). Correlation was measured across all possible pairs of timepoints, quantifying whether colour or motor response representations at one timepoint were correlated with representations at a future timepoint. The correlation was shifted upward with respect to the diagonal line (36 ms before saccade start; Fig. 3c,e (blue line)), indicating that the encoding in the shared colour subspace predicted the future encoding in the shared response subspace. Notably, there was a negative correlation between the sensory and motor responses early in the trial (Fig. 3c), which may reflect suppression of the motor response during fixation and initial integration of the stimulus input.

Importantly, the transformation of stimulus information into response was specific to the task: during the C1 task, the shared colour representation did not predict the associated response direction along axis 2 (8 ms after saccade start; Fig. 3d,e (red line)). This is consistent with the shared colour representation being selectively transformed into a motor response along axis 1, and not axis 2, when the monkey was performing the C1 task. By contrast, when the animals performed the C2 task, the shared colour representation was transformed into a motor response along axis 2, and not axis 1 (Extended Data Fig. 7a–c). To visualize the dynamics of this transformation, we used targeted dimensionality reduction (TDR)[21] to project neural activity onto dimensions encoding the colour category (red versus green) and the motor response along both axis 1 and axis 2 (Fig. 3f and Extended Data Fig. 7d–g). Consistent with the results from the classifiers, neural activity during the C1 and C2 tasks initially evolved along the colour axis according to the stimulus' category before transforming onto the axis 1 or axis 2 dimensions for the C1 and C2 tasks, respectively.

Together, these results suggest tasks sequentially engaged shared subspaces, selectively transforming stimulus representations into motor representations in a task-specific manner.

## Task belief engaged shared subspaces

Theoretical modelling suggests that shared subspaces could facilitate cognitive flexibility by allowing the brain to engage previously learned, task-appropriate representations and/or computations[4,5]. If true, then this predicts that task-appropriate shared subspaces should be engaged as the animal discovers the task in effect.

To begin to test this hypothesis, we first measured the monkey's internal representation of the current S1 or C1 task (that is, their 'belief' about the task, as encoded by the neural population). To do so, we trained a classifier to decode the identity of the S1 and C1 tasks using neural activity in the LPFC during the fixation period (before stimulus onset[28]). Training was restricted to the last 75 trials of each task block, when behavioural performance was high (Fig. 1f). We then applied the task classifier to the beginning of the C1 task blocks to measure the animals' internal belief about the task as they discovered which task was in effect (Fig. 4b (1) and Methods). As noted above, monkeys slowly discovered whether the S1 or C1 task was in effect[19]. Notably, the monkey's rate of learning depended on the sequence of tasks. When switching into a C1 task, animals learned more quickly when the previous axis 1 task was C1 compared to S1 (Fig. 4a (purple, C1–C2–C1 task sequence; black, S1–C2–C1 task sequence); $\Delta = 10.26\%$ change in performance over the first 20 trials, $P < 0.001$, $\chi^2$ test). A similar pattern was seen for S1: performance on task sequence S1–C2–S1 was greater than C1–C2–S1 (Extended Data Fig. 8a; $\Delta = 4.99\%$, $P = 0.032$, $\chi^2$ test). As the C2 task had a unique response axis, it was not affected by the preceding task (Extended Data Fig. 8b; $\Delta = 1.42\%$ between the S1–C2 and C1–C2 sequences; $P = 0.234$, $\chi^2$ test).

As the animals discovered the current task, the performance of the task classifier increased (Fig. 4c). During the S1–C2–C1 sequences, the classifier was slightly below chance initially, suggesting a slight encoding of the S1 task, before increasing as the monkeys learned (Fig. 4c,d; from 42% to 54% from trial 40 to 110, $P = 0.0079$ permutation test). On C1–C2–C1 task sequences, classifier performance was high immediately after the switch to C1 (67% classifier accuracy on trial 40; $P < 0.001$, permutation test) and increased with trials (Fig. 4d (purple line); 13% increase from trial 40 to 110, $P = 0.0079$, permutation test). Moreover, the task classifier during the C1 task in S1–C2–C1 sequences was correlated with behavioural performance on the colour categorization task (Fig. 4e; $P = 0.0239$, permutation test) and anti-correlated with how much behaviour depended on the shape category (Extended Data

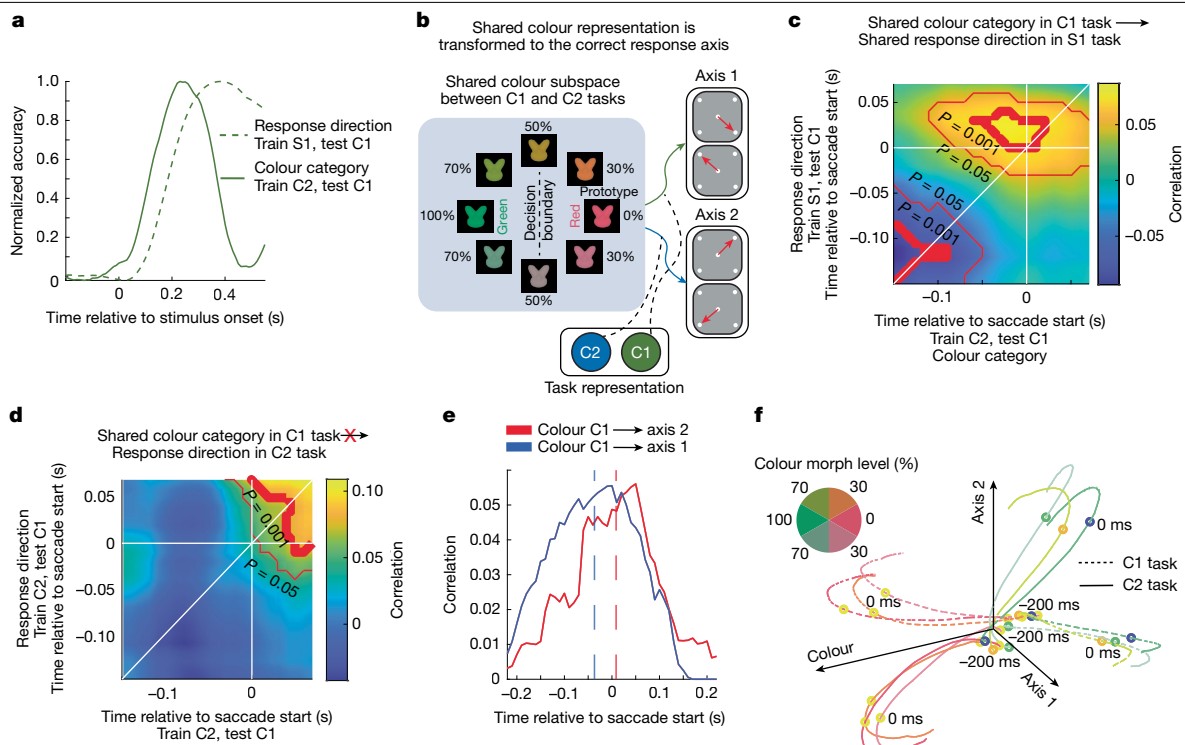

**Fig. 3 | Shared representations were transformed into shared motor representations during the task. a**, Sequential processing of shared colour category information and shared response direction information in the LPFC. Classifier accuracy was normalized to the range from the baseline 200 ms before stimulus to the maximum accuracy. **b**, Schematic showing the prediction that shared colour representation is transformed to the axis 1 and axis 2 response axis during the C1 and C2 tasks, respectively. **c**, Cross-temporal, trial-by-trial correlation of shared colour category encoding (trained on the C2 task, tested on the C1 task) and shared response location encoding (trained on the S1 task, tested on the C1 task). The thin and thick red lines indicate $P \leq 0.05$ and $P \leq 0.001$, respectively, as determined using uncorrected two-sided $t$-tests. **d**, Cross-temporal, trial-by-trial correlation of shared colour category encoding (trained on the C2 task, tested on the C1 task) and response direction encoding on axis 2

(trained on the C2 task, tested on the C1 task). Lines as in **c**. **e**, The average cross-temporal correlation between colour category and response direction on axis 1 (blue line, anti-diagonal axis from **c**) and between colour category and response on axis 2 (red line, from **d**). The shift of the curve's centre towards negative time values reflects the extent to which encoding in the shared colour subspace predicts future encoding in the shared response subspace. The dotted lines show mean of the Gaussian fit to each curve. **f**, Dynamics of LPFC population activity projected onto dimensions of neural activity encoding colour category, response on axis 1 and response on axis 2 for the C1 (dashed line) and C2 (solid line) tasks, showing task-dependent sensory–motor transformation. The line colours match the colour of the stimuli. Time stamps denote the time from saccade start.

Fig. 8c, $P = 0.00796$, permutation test; similar results were seen during the C1–C2–C1 sequence; Extended Data Fig. 8d,e).

So far, our results suggest the monkeys tracked whether the S1 or C1 task was in effect, and that this belief was maintained during the intervening C2 task. Consistent with this, the task classifier trained during the S1 and C1 tasks could decode the identity of the previous task during the subsequent C2 task (Fig. 4b (2); 62%, $P = 0.0009$, permutation test) and into the start of the next S1 or C1 task (Fig. 4b (3); 64%, $P = 0.0009$, permutation test). Similar results were seen in the FEF, but not in other regions (Extended Data Fig. 8f).

We next examined whether the animal's internal representation of the task predicted the engagement of the shared colour category and shared motor response subspaces. The strength of the representation in the shared colour subspace increased as the monkeys discovered the C1 task during the S1–C2–C1 sequences (Fig. 4f,g and Methods; 64% to 69%, $\Delta = 5\%$ from trial 40 to 110; $P = 0.0478$, permutation test). This increase in the shared colour category representation of the stimulus was predicted by the strength of internal task representation in the LPFC during fixation (Fig. 4h and Methods; $P = 0.0478$, permutation test). Furthermore, the colour category subspace in C1 became aligned with the colour category subspace in C2 as the animal discovered that the C1 task was in effect (Extended Data Fig. 9a–c). Together, these results suggest that the shared colour subspace was dynamically engaged as the monkey updated its belief about the current task.

Like task representations, colour representations depended on the task sequence (Fig. 4g). While shared representations increased when discovering C1 after a S1–C2–C1 sequence, they remained stable during C1–C2–C1 sequences (63% to 62%, $\Delta = -1\%$ from trial 40 to 110; $P = 0.6852$, permutation test). While this is consistent with better performance at the beginning of the C1 task during C1–C2–C1 sequences, it does not explain the improvement in behaviour over trials during the C1–C2–C1 sequence (Fig. 4a). This difference in performance was associated with an increase in colour information in the FEF during the C1–C2–C1 sequence (Extended Data Fig. 8g; the difference in accuracy between C1–C2–C1 and S1–C2–C1 was 12.7%; $P = 0.004$), possibly reflecting the consolidation of colour category information from the LPFC into FEF as the animal became more certain about the current task.

The animal's behaviour and internal representation of the task suggest that they initially expected to perform the S1 task in the S1–C2–C1 task sequences. If true, and task representations modulate the strength of shared subspace representations, then shape information should initially be strong and then decay as the animal learns the C1 task is in effect. To test this, we trained a classifier to decode the stimulus' shape category during the last 75 trials of the S1 task, and then tested it as the animal discovered the C1 task (Fig. 4i and Methods for details). In contrast to colour, shape representation was significant immediately after the switch to the C1 task during S1–C2–C1 sequences (53%, $P = 0.02$, permutation test) and then decreased, and delayed in time,

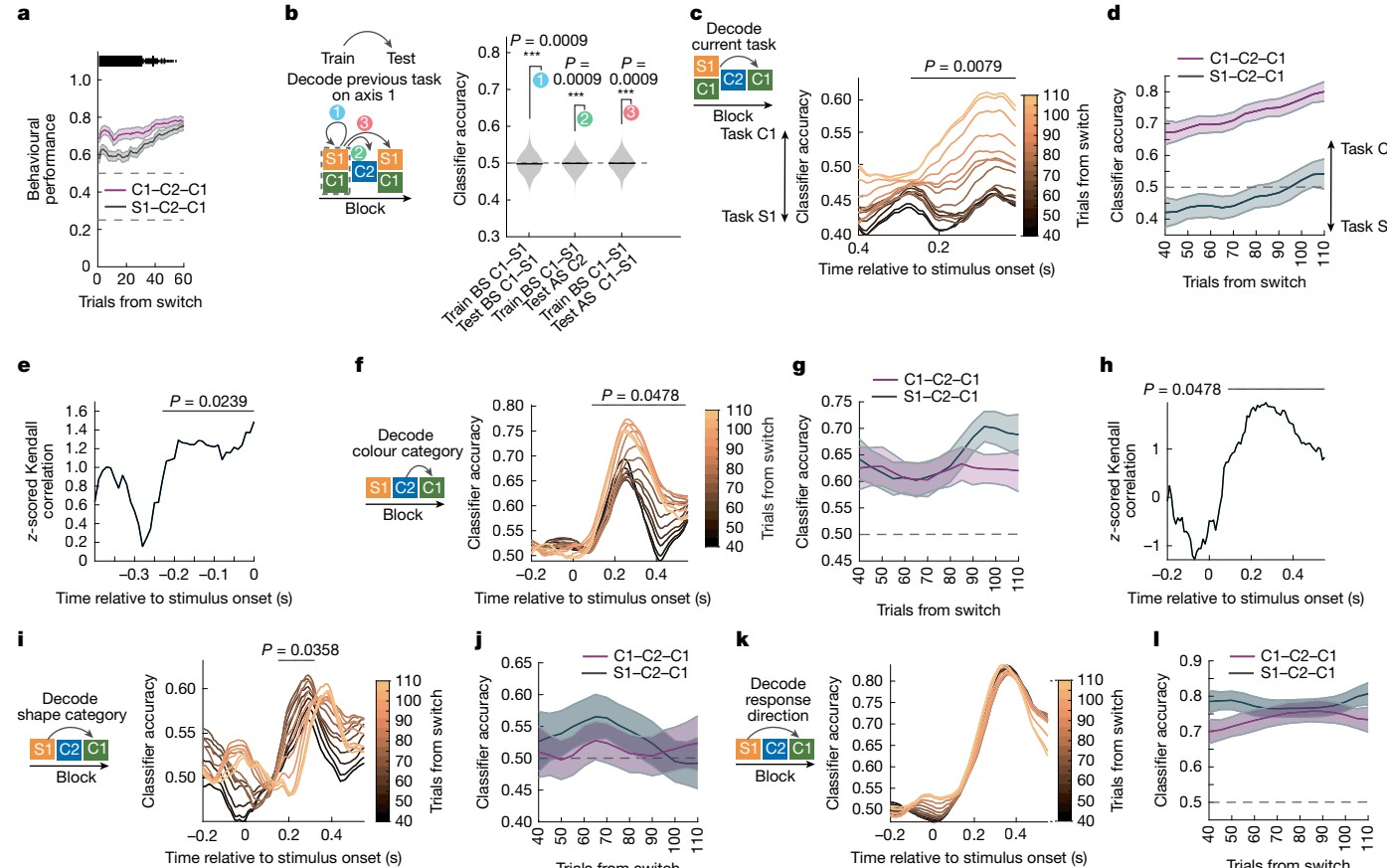

**Fig. 4 | Shared subspaces are dynamically engaged during task discovery.**
**a**, Behavioural performance during the C1 task after C1–C2–C1 (purple) and
S1–C2–C1 (black) sequences of tasks. Data are mean ± s.e.m. over 44 and
53 blocks of C1–C2–C1 and S1–C2–C1 tasks. The horizontal bars indicate
significant differences: $P < 0.05, 0.01$ and $0.001$ for thin, medium and thick lines,
respectively, as determined using one-sided uncorrected $\chi^2$ tests. The dashed
lines mark chance: 25% overall, 50% within axis 1. **b**, The PFC encodes history of
task identities. Left, schematic of the classifier decoding task. The classifier
was trained on 75 trials of C1 or S1 before the switch at end of the block and was
tested on (1) withheld trials of C1 or S1 before the switch; (2) C2 trials to decode
the identity of the previous C1 or S1 block; (3) C1 or S1 trials after the switch to
decode the identity of the previous C1 or S1 block. Right, the numbered circles
show the accuracy of classifiers, and the grey violin plot shows the null
distribution. $P$ values were calculated using two-sided permutation tests.
AS, after the switch; BS, before the switch. **c**, Schematic of the classifier for the
decoding task (left). Right, the accuracy in decoding identity of C1 task over
time, during the S1–C2–C1 sequence. The line colours show progression over

the block (window of 40, shifted by 5 trials). The lines show the mean accuracy
of 250 classifier iterations. The horizontal bars indicate an above chance trend
in classifier accuracy across trials, as determined using two-sided permutation
tests, corrected for multiple comparisons. **d**, Accuracy in decoding C1 task
after switching into C1, during the C1–C2–C1 (purple) and S1–C2–C1 (black)
sequences. Data are mean ± s.e.m. estimated from 250 classifier iterations.
**e**, Correlation of behavioural performance and decoded task belief during the
S1–C2–C1 sequence. $z$-scored Kendall's tau with two-sided permutation test,
corrected for multiple comparisons. **f**, Accuracy in decoding of colour category
over time, during the S1–C2–C1 sequence. Line colours as in **c**. **g**, Accuracy of the
colour category classifier over trials in the C1–C2–C1 (purple) and S1–C2–C1
(black) sequences. Data are mean ± s.e.m. estimated from 250 classifier
iterations. **h**, Correlation of task belief and colour category during the S1–C2–C1
sequence. Lines and statistics as in **e**. **i**–**j**, As in **f** and **g**, respectively, but showing
accuracy in decoding of shape category. **k**,**l**, As in **f** and **g**, respectively, but
showing accuracy in decoding of response direction.

as the animal learned (Fig. 4i,j; 49%, $\Delta = -4\%$, $P = 0.0358$, permutation
test). By contrast, shape information was reduced overall and remained
stable during the C1–C2–C1 task sequences (Fig. 4j (purple); 51% on
trial 40, $\Delta = 1\%$, $P = 0.1035$, permutation test). Finally, consistent with
task belief modulating engagement of shape representations, belief
about the C1 task was inversely correlated with the representation of
shape (Extended Data Fig. 8h).

In contrast to shared colour and shape subspaces, the animals'
motor response was stably decoded in a shared subspace through-
out the block (Fig. 4k,l; $\Delta = 3\%$ ($P = 0.2868$) and $\Delta = 3\%$ ($P = 0.6733$) for
the S1–C2–C1 and C1–C2–C1 sequences, respectively; permutation
tests; Methods and Extended Data Fig. 9d). The representation in the
shared response subspace was not correlated with the strength of
internal task representation in the LPFC ($P = 0.546$, permutation test;
Extended Data Fig. 8i). This makes sense, as both S1 and C1 tasks used
the same motor response.

Together, these results suggest the animals maintained an internal
belief about whether they were performing the S1 or C1 task and then
used this internal belief to selectively engage the relevant shared colour
and shape subspaces during the C1 task. This information was then
transformed into the motor subspace to perform the task.

## Task belief scaled shared subspaces

Previous research suggests tasks may modulate the gain of stimulus
features, depending on their relevance for the current task[23,29,30]. Simi-
larly, we found stimulus information was attenuated when irrelevant
(Fig. 2a,b). Furthermore, this attenuation depended on the animals'
internal belief about the task: as they discovered the C1 task was in
effect, the representation of colour category was magnified (Fig. 5a),
while shape representation was attenuated and delayed in time (Fig. 5b
and Extended Data Fig. 8j).

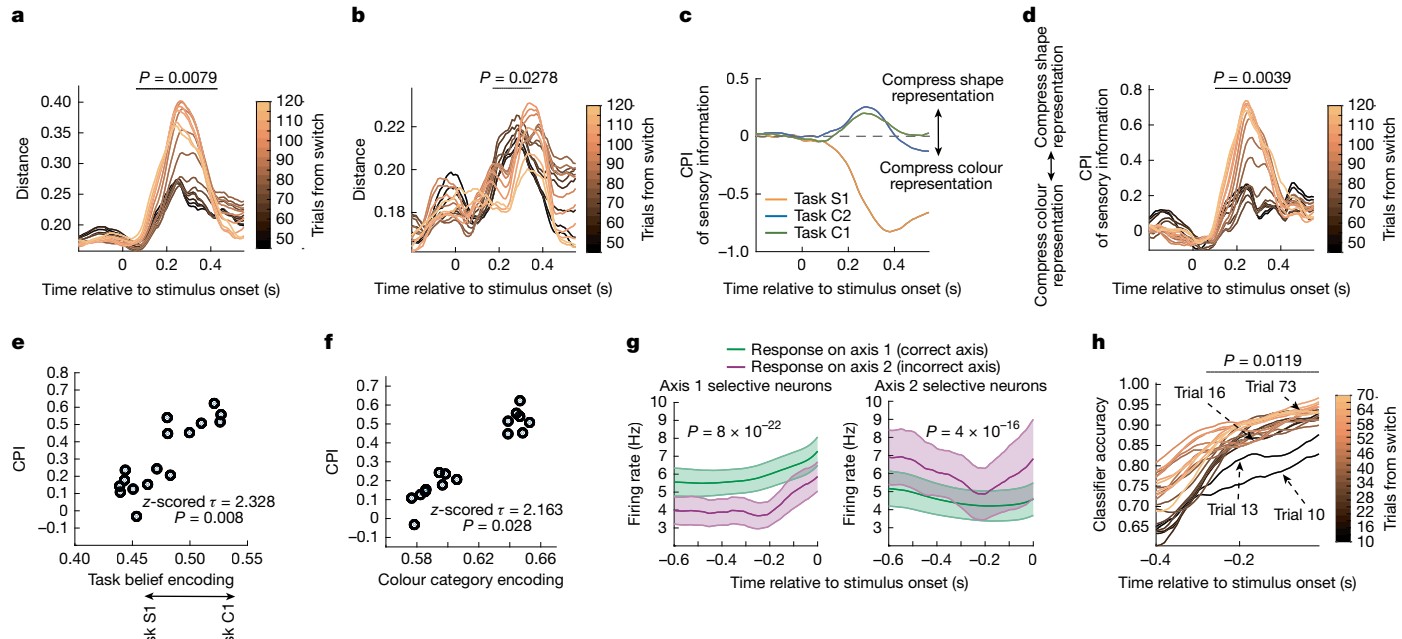

**Fig. 5 | Irrelevant sensory and motor representations were suppressed during flexible behaviour. a**, The distance between neural response to stimuli from each colour category, projected onto shared colour category encoding axis (trained on C2, tested on C1, following the S1–C2–C1 sequence). Line colour shows progression in block (window of 45 trials, shifted by 5 trials). Lines show the mean value for 250 iterations of classifiers. The horizontal bar indicates a significant trend in distance over trials, as determined using two-sided permutation tests, corrected for multiple comparisons. **b**, Distance between neural response to stimuli from each shape category, projected onto shape category encoding axis (trained on S1, balanced for response direction, tested on C1, following the S1–C2–C1 sequence). Lines as in **a**. **c**, Suppression of irrelevant sensory information in the LPFC. The CPI over time for all three tasks is shown. CPI is equal to the log of the ratio of the separability of stimuli in colour and shape subspaces. **d**, CPI over trials after the switch into C1 (following the S1–C2–C1 sequence). Irrelevant sensory information is gradually suppressed during task discovery. Lines as in **a**. **e**, Correlation between task belief encoding and the

average CPI during 100–300 ms after stimulus onset. Two-sided uncorrected permutation test. **f**, Correlation between colour category encoding and average CPI during 100–300 ms after stimulus onset. Two-sided uncorrected permutation test. **g**, The average firing rate (FR) for axis-1-selective (left) and axis-2-selective (right) neurons in the LPFC during the C1 task when animals responded on axis 1 (correct axis, green) or axis 2 (incorrect axis, purple). Irrelevant response axis information is suppressed during task discovery. Data are mean ± s.e.m. over time. $P$ values were determined using two-sided paired $t$-tests, comparing the difference in FR measured over entire time period. **h**, Accuracy of the classifier decoding response axis across trials after the switch into C1 (window of 10 trials, shifted by 3 trials; following the S1–C2–C1 sequence). The response axis representation is quickly updated after the switch. Lines as in **a**. The horizontal bar indicates a significant increase in classifier accuracy from trial 10 to trial 16, as determined using a permutation test with cluster mass correction.

Amplitude modulation can be thought of as a geometric scaling of stimulus representations in feature space[23,29]. To quantify this, we defined a compression index (CPI) as the log of the ratio between the separability of stimuli in (1) the colour subspace and (2) the shape subspace. Colour and shape representations were scaled in all three tasks. There was greater separation of colour representation during C1 and C2 tasks, and greater separation of shape representations during the S1 task (similar results were found when controlling for motor response; Fig. 5c and Extended Data Fig. 9e,f). Stimulus representations were also scaled in the FEF and PAR, but not in the aIT, suggesting that the aIT maintains a veridical representation of the stimulus[22,23] (Extended Data Fig. 9e–g).

The scaling of representations changed as animals learned which task was in effect (Fig. 5d). CPI was positively correlated with the strength of the task representation and colour category encoding in the LPFC (Fig. 5e,f; $z$-scored Kendall's $\tau = 2.328$, $P = 0.008$, permutation test and $z$-scored Kendall's $\tau = 2.163$, $P = 0.028$, permutation test, respectively).

In addition to selecting sensory information, the three tasks also require selecting the appropriate motor response: the animal must suppress responses on axis 1 during the C2 task and axis 2 during the C1 task. Consistent with scaling acting as a general mechanism for modulating task-(ir)relevant information, neurons selective for each motor axis were suppressed when the animal performed a task requiring a response on the other axis (Fig. 5g and Extended Data Fig. 10a,b;

see Extended Data Fig. 10c–e for further controls and Extended Data Fig. 10f–i for other regions). However, in contrast to the gradual suppression of sensory representations during task discovery, the incorrect response axis representation in the LPFC was quickly suppressed after a change in task (within 3–6 trials; Fig. 5h; Δ(classifier performance in trial 10 and trial 16) = 6.88%, $P = 0.0119$, permutation test; see Extended Data Fig. 10j,k for other regions). This reflected the rapid inference of response axis seen in the animals' behaviour (Extended Data Fig. 2g,h).

Together, these results suggest the monkeys' belief about the current task dynamically adjusted the magnitude of sensory and motor representations[31–33]. Scaling neural representations was a common mechanism used in all the three tasks and could facilitate task-relevant associations between stimulus category and motor response, while preventing task-irrelevant associations.

## Discussion

Our results suggest the brain can perform a task by compositionally engaging a series of representational subspaces. Subspaces of neural activity within prefrontal cortex represented task-relevant information, including the colour category and shape category of the stimulus[21,34] and the motor action[35]. Consistent with computational models[4–7], these subspaces were shared across multiple tasks (Fig. 2), suggesting they act as task components[21,36–38]. Subspaces were sequentially engaged, such

that information from the relevant sensory subspace was transformed into the appropriate motor response subspace[39] (Fig. 3). In this way, a task can be constructed by sequencing together a series of task components. For example, performing the C1 task engaged the subspace representing the colour category of the stimulus (shared with the C2 task) and then transformed this information into the motor subspace encoding response along axis 1 (shared with the S1 task).

Although our study is limited to three tasks, the underlying mechanism has the ability to be highly expressive—flexibly sequencing task components together could implement a wide variety of behaviours (a form of sequential compositionality[4]). In this framework, the representation of the task acts as a control input that selects the appropriate representations and computations[4,5,40] (Figs. 4 and 5). If, as suggested by our results, the brain can reuse representations and computations across tasks, then this could allow one to rapidly adapt to changes in the environment, either by learning the appropriate task representation through reward feedback[41,42] or by recalling it from long-term memory[43].

Performing multiple tasks can lead to either shared representations or task unique representations[23,44]. Several previous studies have found different tasks use representations of sensory and motor information that are specific to that task[24,45]. Such task-unique representations have the advantage of minimizing interference when learning and performing multiple tasks[45,46], but limit the ability to generalize learning across tasks[3,23]. By contrast, shared subspaces increase interference but could speed learning by allowing knowledge to generalize across tasks[6]. For example, learning the C2 task shapes the neural computations needed to categorize the colour of a stimulus, which could generalize to other tasks that involve categorizing colour, such as C1. Future computational and experimental work is needed to test whether such transfer of knowledge facilitates learning new tasks.

Representations in the prefrontal cortex generalized across tasks[38,47] (Fig. 2), although other regions, such as the hippocampus, are probably involved[34]. Less generalization in the PAR, visual cortex and striatum suggests that neural representations are more task-unique in these regions. Differences between regions may allow the brain to use the complementary advantages of both sharing representations (generalizing learning) and task-unique representations (avoiding interference)[23,48].

Whether a network uses a shared or task-unique representation depends on training curricula[32], initial conditions[48] and biological factors. Given that the animals were trained for months on the tasks, one might expect the brain to form independent task representations to reduce interference[4,44,45]. Instead, we found the brain reduced interference by dynamically amplifying task-relevant sensory and motor representations and suppressing task-irrelevant dimensions[31,37,49] (Figs. 4 and 5).

One reason we found shared representations (rather than task-unique ones) may be because our task required the animal to continuously learn. Across trials, the monkeys used feedback to infer the task context, updating their internal belief about the task, and flexibly mapping stimuli onto motor responses (Figs. 4 and 5; a form of class-incremental continual learning[16]). This may enable the brain to flexibly interpolate through a range of behaviours: updating its representation of the current task along a task manifold[41] to parametrically modulate how colour category and shape category influence decision-making and to ensure the animal performed the appropriate response (for example, responding on axis 1, not axis 2, during C1 and vice versa for C2).

As noted above, shared representations facilitate continual learning by allowing knowledge to generalize between tasks[50]. In addition to reducing interference, the scaling of neural representations may also facilitate continual learning by constraining learning to those representations that are currently task relevant[51–53]. Neural learning rules are activity dependent[54] and gated by reward[55] and therefore suppressing irrelevant representations may limit learning to task-relevant representations (addressing the credit assignment problem[56]). Future work is

needed to understand how suppressing irrelevant features supports continual learning and consider alternative mechanisms[16], including replaying experiences across tasks[57,58] and recalling context-specific associations from long-term memory[43].

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

## Methods

### Monkeys

Two adult male rhesus macaques (*Macaca mulatta*) participated in the experiment. The number of monkeys (2) follows previous work using similar approaches[21]. Monkeys Si and Ch were between 8 and 11 years old and weighed approximately 12.7 and 10.7 kg, respectively. All of the experimental procedures were approved by the Princeton University Institutional Animal Care and Use Committee (protocol, 3055) and were in accordance with the policies and procedures of the National Institutes of Health.

### Behavioural task

Each trial began with the monkeys fixating on a dot at the centre of the screen. During a fixation period lasting 500 ms–800 ms, the monkeys were required to keep their gaze within a circle with a radius of 3.25 degrees of visual angle around the fixation dot. After the fixation period, the stimulus and all four response locations were simultaneously displayed.

Stimuli were morphs consisting of both a colour and shape (Fig. 1a). The stimuli were rendered as three-dimensional models using POV-Ray and MATLAB (MathWorks) and displayed using Psychtoolbox on a Dell U2413 LCD monitor positioned 58 cm from the animal. Stimuli were morphed along circular continua in both colour and shape (that is, drawn from a four-dimensional 'Clifford' torus; Fig. 1b). Colours were drawn from a photometrically isoluminant circle in the CIELAB colour space, connecting the red and green prototype colours. Shapes were created by circularly interpolating the parameters defining the lobes of the 'bunny' prototype to the parameters defining the corresponding lobes of the 'tee' prototype. The mathematical representation of the morph levels adhered to the equation $X_1^2 + X_2^2 + X_3^2 = P^2$ where $X$ represents the parameter value in a feature dimension (for example, $L$, $a$, $b$ values in CIELAB colour space). We chose the radius ($P$) to ensure sufficient visual discriminability between morph levels. The deviation of each morph level from prototypes (0% and 100%) was quantified using percentage, corresponding to positions on the circular space from $-\pi$ to 0 and 0 to $\pi$. Morph levels were generated at eight levels: 0%, 30%, 50%, 70%, 100%, 70%, 50%, 30%, corresponding to 0, $\pi/6$, $\pi/2$, $5\pi/6$, $\pi$, $-5\pi/6$, $-\pi/2$ and $-\pi/6$, respectively. 50% morph levels for one feature were only generated for prototypes of the other feature (that is, 50% colours were only used with 0% or 100% shape stimuli and vice versa). Stimuli were presented at fixation and had a diameter of 2.5 degrees of visual angle.

The monkeys indicated the colour or shape category of the stimulus by saccading to one of the four response locations, positioned 6 degrees of visual angle from the fixation point at 45, 135, 225 and 315 degrees, relative to the vertical line. The reaction time was taken as the moment of leaving the fixation window relative to the time of stimulus onset. Trials with a reaction time below 150 ms were terminated, followed by a brief timeout (200 ms). Correct responses were rewarded with juice, while incorrect responses resulted in short timeouts lasting 1 s for monkey Ch and 5 s for monkey Si. After the trial finished, there was an intertrial interval lasting 2–2.5 s before the next trial began.

The animals performed three category-response tasks (Fig. 1c). The S1 task required the monkeys to categorize the stimulus by its shape. For a stimulus with a shape that was closer to the 'bunny' prototype, the animals had to make a saccade to the UL location to get rewarded. For a stimulus with a shape closer to the 'tee' prototype, the animals had to make a saccade to the LR location to be rewarded. For ease of notation, we refer to the combination of the UL and LR target locations as axis 1. The C1 task required the monkeys to categorize the stimulus by its colour. When a stimulus' colour was closer to 'red', the animals made an eye movement to the LR location and when it was closer to 'green' the animals made an eye movement to the UL location. Finally, in the C2 task, the monkeys again categorized stimuli based on their colour

but responded to the UR location for red stimuli and the LL location for green stimuli. Together, the UR and LL targets formed axis 2. This set of three tasks was designed to be related to one another: the C1 and C2 tasks both required categorizing the colour of the stimulus while the C1 and S1 tasks both required responding on axis 1.

The monkeys were not explicitly cued as to which task was in effect. However, they did perform the same task for a block of trials allowing the animal to infer the task based on the combination of stimulus, response and reward feedback. Tasks switched when the monkeys' performance reached or exceeded 70% on the last 102 trials of task S1 and task C1 or the last 51 trials of task C2. For monkey Si, block switches occurred when their performance reached or exceeded the 70% threshold for all morphed and prototype stimuli in the relevant task dimension independently. For monkey Ch, block switches occurred when their average performance at each morph level in the relevant task dimension exceeded the 70% threshold (that is, average of all 30% morphs, average of all 70% morphs, and 0% and 100% prototypes in colour dimension were all equal or greater than 70% accuracy in the C1 and C2 tasks). Moreover, on a subset of recording days, to prevent monkey Ch from perseverating on one task for an extended period of time, the threshold was reduced to 65% over the last 75 trials for S1 and C1 tasks after the monkey had already done 200 or 300 trials on that block.

When the animal hit the performance threshold, the task switched. This was indicated by a flashing yellow screen, a few drops of reward and a delay of 50 s.

To ensure even sampling of tasks despite the limited number of trials each day, the axis of response always changed following a block switch. During axis 1 blocks, either S1 task or C1 task was pseudorandomly selected, interleaved with C2 task blocks. Pseudorandom selection within axis 1 blocks avoided three consecutive blocks of the same task, ensuring the animal performed at least one block of each task during each session. On average, animals performed of 560, 558 and 301 trials and 2.68, 2.77 and 5.4 blocks per day for the S1, C1 and C2 tasks, respectively. Task orders and trial conditions were randomized across trials within each session.

Monkeys Si and Ch underwent training for 36 and 60 months, respectively. Both animals were trained in the same order of tasks: S1, C2 and then C1. Each animal underwent exposure to every task manipulation. As all of the animals were allocated to a single experimental group, blinding was neither necessary nor feasible during behavioural training. Electrophysiological recordings began when the monkeys consistently executed five or more blocks daily. Further details on the behavioural methods and results have been previously reported[19].

### Congruent and incongruent stimuli

During the S1 and C1 tasks red-tee stimuli and green-bunny stimuli were 'congruent' as they required a saccade to the LR and UL locations, respectively, in both tasks. By contrast, stimuli in the red-bunny and green-tee portion of stimulus space were 'incongruent', as they required different responses during the two tasks (UL and LR in S1 task; LR and UL in C1 task, respectively). To ensure the animals performed the task well, 80% of the trials included incongruent stimuli. Note that, as the C2 task was the only one to use axis 2, there were no congruent or incongruent stimuli on those blocks.

### Analysis of behavioural data

Psychometric curves plot the fraction of trials in which the animals classified a stimulus with a specific morph level as being a member of the 'green' category for the C1 and C2 tasks or the 'tee' category for the S1 task. The fraction of responses for a given morph level of the task-relevant stimulus dimension was averaged across all morph levels of the task-irrelevant dimension during the last 102/102/51 trials of the S1/C1/C2 task (for example, for the C1 task, the fraction of responses for 70% green stimuli were averaged across all shape morph levels). As the stimulus space was circular, we averaged the behavioural response for

the two stimuli at each morph level on each side of the circle (Fig. 1b). Psychometric curves were quantified by fitting the mean datapoints with a modified Gauss error function (erf):

$$F = \theta + \lambda \times \left( \text{erf}\left(\frac{x-\mu}{\sigma}\right) + 1 \right) \Big/ 2,$$

where erf is the error function, $\theta$ is a vertical bias parameter, $\lambda$ is a squeeze factor, $\mu$ is a threshold, and $\sigma$ is a slope parameter. Fitting was done in MATLAB with the maximum likelihood method (fminsearch.m function).

To calculate performance during task discovery (Fig. 1f), we used a sliding window of 15 trials, stepped 1 trial, to calculate a running average of performance immediately after a switch or right before a switch. Performance was estimated using trials from all blocks of the same task, regardless of the identity of the stimulus. To test whether behavioural performance differed significantly between the C1–C2–C1 and S1–C2–C1 sequences during task discovery (Fig. 4a; S1–C2–S1 or C1–C2–S1 in Extended Data Fig. 8a and C1–C2 or S1–C2 in Extended Data Fig. 8b), we applied $\chi^2$ test using the chi2test.m function as implemented previously[59].

To compare distribution of task performance after switch and before switch for C1 and S1 tasks (Extended Data Fig. 2e,f), we computed behavioural performance in first 15 trials after switch and last 15 trials before switch for each block, respectively.

To compare colour categorization performance based on the stimulus shape morph level in C1 and C2 tasks (Extended Data Fig. 2c), for each colour morph level, we computed colour categorization psychometric curves using combination of each colour morph level and three sets of shape morph levels: ambiguous, 50% and 150% (that is, $-\pi/2$); intermediate, 30%, 70%, 130% (that is, $-5\pi/6$) and 170% (that is, $-\pi/6$); and prototype, 0% and 100%.

### Surgical procedures and electrophysiological recordings

Monkeys were implanted with a titanium headpost to stabilize their head during recordings and two titanium chambers (19 mm diameter) placed over frontal and parietal cortices that provided access to the brain. Chamber positions were determined using 3D model reconstruction of the skull and brain using structural MRI scans. We recorded neurons from the LPFC (Brodmann area 46 d and 46 v, 480 neurons), aIT (area TEa, 239 neurons), PAR (Brodmann area 7, 64 neurons), the FEF (Brodmann area 8, 149 neurons) and the striatum (STR, caudate nucleus, 149 neurons). The number of monkeys and the number of neurons recorded per region were chosen to follow previous work using similar approaches[38].

Two types of electrodes were used during recordings. To record from the LPFC and FEF we used epoxy-coated tungsten single electrodes (FHC). Pairs of single electrodes were placed in a custom-built grid with manual micromanipulators that lowered electrode pairs using a screw. This enabled us to record 20–30 neurons simultaneously from cortical areas near the surface. To record from deeper regions (PAR, aIT and STR) we used 16ch or 32ch Plexon V-Probes (Plexon). These probes were lowered using the same custom-built grid through guide tubes. During lowering, we used both structural MRI scans and the characteristics of the electrophysiological signal to track the position of the electrode. A custom-made MATLAB GUI tracked electrode depth during the recording session and marked important landmarks until we found the brain region of interest.

Recordings were acute; up to 50 single electrodes and three V-Probes were inserted into the brain each day (in some recording sessions one additional V-Probe was also inserted in the LPFC). Single electrodes and V-probes were lowered though the intact dura at the beginning of recording session and allowed to settle for 2–3 h before recording, improving the stability of recordings. Neurons were recorded without bias. Electrodes were positioned to optimize the signal-to-noise ratio of the electrophysiological signal without consideration of neural type

or selectivity. Experimenters were blinded to experimental conditions while recording neurons. We did not simultaneously record from all five regions in all recording sessions. We began recording from the LPFC and FEF for the first 5–10 days and then added PAR, STR and aIT on successive days.

Broadband neural activity was recorded at 40 kHz using a 128-channel OmniPlex recording system (Plexon). We performed 15 recording sessions with monkey Ch and 19 recording sessions with monkey Si. After all recordings were complete, we used electrical microstimulation in monkey Si, and structural MRI and microstimulation in monkey Ch, to identify the FEF. Electrical stimulation was delivered as a train of anodal-leading biphasic pulses with a width of 400 μs and an interpulse frequency of 330 Hz. A site was identified as the FEF when electrical microstimulation of around 50 μA evoked a stereotyped eye-movement on at least 50% of the stimulation attempts. In monkey Ch, untested electrode locations were classified as FEF if they fell between two FEF sites (as confirmed with electrical stimulation) in a region that was confirmed as being FEF using MRI.

Eye position was recorded at 1 kHz using an Eyelink 1000 Plus eye-tracking system (SR Research). Sample onset was recorded using a photodiode attached to the screen. Eye position, photodiode signal and behavioural events generated during the task were all recorded alongside neural activity in the OmniPlex system.

### Signal preprocessing

Electrophysiological signals were filtered using a 4-pole high pass 300 Hz Butterworth filter. To reduce common noise, the median of the signals recorded from all single electrodes in each chamber was subtracted from the activity of all single electrodes. For V-Probe recordings, we subtracted the median activity for all channels along the probe from each channel. To detect spike waveforms, a $4\sigma_n$ threshold was used where $\sigma_n$ is an estimate of s.d. of noise distribution of signal $x$ defined as:

$$\sigma_n = \text{median}\left(\frac{|x|}{0.6745}\right)$$

Timepoints at which the electrophysiological signal ($x$) crossed this threshold with a negative slope were identified as putative spiking events. Spike waveforms were saved (total length was 44 samples, 1.1 ms, of which 16 samples, 0.4 ms, were pre-threshold). Repeated threshold crossing within 48 samples (1.2 ms) was excluded. All waveforms recorded from a single channel were manually sorted into single units, multiunit activity or noise using Plexon Offline Sorter (Plexon). Units that were partially detected during a recording session were also excluded. Experimenters were blinded to the experimental conditions while sorting waveforms into individual neurons. All analyses reported in this Article were performed on single units.

For all reported electrophysiology analysis, saccade time was calculated as the moment at which the instantaneous eye speed exceeded a threshold of 720 degrees of visual angle per second. Instantaneous eye speed was calculated as $\sqrt{\left(\frac{dx}{dt}\right)^2 + \left(\frac{dy}{dt}\right)^2}$, where $x$ and $y$ are the position of the eye on the monitor at time $t$.

### Statistics and reproducibility

Independent experiments were performed on two monkeys and the data were combined for subsequent analyses. As described below, statistical tests were corrected for multiple comparisons using two-tailed cluster correction unless stated otherwise. Unless otherwise noted, nonparametric tests were performed using 1,000 iterations; therefore, exact $P$ values are specified when $P > 0.001$. Unless stated otherwise, all data were smoothed with a 150 ms boxcar. To compare the onset timing differences between within and shared-colour representations across brain regions (Fig. 2g and Extended Data Fig. 6d), classifier accuracy for each region was smoothed using a 50 ms boxcar filter.

All analyses were performed in MATLAB 2021b (MathWorks).

## Using cluster mass to correct for multiple comparisons

To assess the significance of observed clusters in the time series data, we used a two-tailed cluster mass correction method[60]. This approach is particularly useful when dealing with multiple comparisons and helps identify clusters of contiguous timepoints that exhibit statistically significant deviations from the null distribution.

We first generated a null distribution (NullDist) by shuffling the observed data, breaking the relationship between the neural signal and the task parameter of interest (details of how data were shuffled are included with each test below). The observed data were also included as a 'shuffle' in the null distribution and the $z$-score of the null distribution for each timepoint was calculated. To define significant moments in time, we computed the upper and lower thresholds based on the null distribution. The thresholds were determined non-parametrically using percentiles. The resulting thresholds, denoted as $P_{\text{threshold upper}}$ and $P_{\text{threshold lower}}$, serve as critical values for identifying significant deviations in both tails.

Timepoints of significant signal were identified by finding the moments when the value in each shuffle within the null distribution or the observed data exceeded the computed thresholds. These timepoints were then clustered in time, such that contiguous values above the threshold were summed together. The sum was calculated on the $z$-transformed values of each time series. This resulted in a mass of each contiguous cluster in the data. To correct for multiple comparisons, we took the maximum absolute value of the cluster mass across time for each shuffle. This resulted in a distribution of maximum cluster masses. Finally, the two-tailed $P$ value of each cluster in the observed data was determined by comparing its mass to the distribution of maximum cluster masses in the null distribution.

## FR calculation

In all analyses, we estimated the FR of each neuron by averaging the number of action potentials within a 100 ms window, stepping the window every 10 ms. Changing the width of the smoothing window did not qualitatively change our results. The time labels in the figures denote the trailing edge of this moving window (that is, 0 ms would be a window from 0 ms to 100 ms).

## GLM

To estimate how individual neurons represented task variables, we used a generalized linear model (GLM) to relate the activity of each neuron ($y$) to the task variables at each moment of time. The full model was:

$$y = \beta_0 + \beta_1 \times \text{stimulus colour category}$$
$$+ \beta_2 \times \text{stimulus shape category} + \beta_3 \times \text{time} + \beta_4 \times \text{task identity}$$
$$+ \beta_5 \times \text{motor response direction} + \beta_6 \times \text{reward}$$

where $y$ is the FR of each neuron, normalized to the maximum FR across all trials, and $\beta_0, \beta_1, \ldots, \beta_6$ are the regression coefficients corresponding to each predictor. Predictors were: stimulus colour category, indicating the colour category of the stimulus (categorical variable); stimulus shape category, indicating the shape category of the stimulus (categorical variable); time, indicating the temporal progression within a recording session, normalized between 0 and 1 (continuous variable); task identity, indicating the identity of the task (categorical variable); motor response direction, indicating the direction of the motor response (categorical variable; UL, UR, LL or LR); reward, indicating whether a reward was received following the response (binary variable; 1 for reward, 0 for no reward).

The GLM coefficients ($\beta$) were estimated using maximum-likelihood estimation as implemented by MATLAB's lassoglm.m function. Independent models were fit for each timepoint. To address potential overfitting, we applied Lasso ($L^1$) regularization to the GLM weights with

regularization coefficients (lambda) values of [0, 0.0003, 0.0006, 0.0009, 0.0015, 0.0025, 0.0041, 0.0067, 0.0111, 0.0183]. In a separate set of runs, we fit the GLM models to 80% of data and tested on the remaining 20%. The lambda value with maximum $R^2$ on the withheld data was used when estimating the CPD.

## CPD calculation

To quantify the unique contribution of each predictor to a neuron's activity, we used the CPD. This metric quantifies the percent of explained variance that is lost when a specific factor is removed from the full model. The CPD was computed by initially fitting the GLM with all the factors (full model) as described above. Each factor was then sequentially removed to create a set of reduced models, each of which were fit to the data. The CPD for each predictor $X(t)$ was calculated as:

$$\text{CPD}(X(t)) = \frac{\text{SSE}_{\text{reduced}} - \text{SSE}_{\text{full model}}}{\text{SSE}_{\text{reduced}}} \times 100$$

where $\text{SSE}_{\text{reduced}}$ is the sum of square error due to predictor $X$, $\text{SSE}_{\text{full model}}$ is the sum of square error of the full model with all predictors[61]. Note that, because the CPD statistic estimates the additional explained variance captured by each term, it controls for potential covariation between terms. Nevertheless, it is important to note that, as with all neurophysiological studies, it is difficult to distinguish between the encoding of a specific cognitive variable (for example, task) and its effect on the representation of other cognitive variables (for example, the observed suppression of sensory and/or motor representations).

We used a permutation test to assess whether CPDs for each predictor, at each timepoint, and for each neuron, was significantly larger than expected by chance. To compute a null distribution for CPD for each predictor, we generated 1,000 permuted datasets by randomly permuting the predictor values relative to the neural activity and refitting the full and reduced models (as above). The likelihood of the observed CPD was then estimated by computing the proportion of permuted datasets that yielded a CPD greater than or equal to that of the observed dataset. To account for multiple comparisons across timepoints, we used cluster correction (detailed above) to estimate corrected $P$ values for each predictor. Neurons that had at least one significant cluster ($P < 0.05$) were considered to significantly carry information about a task variable. To assess whether an area had a significant number of neurons for a given task variable, we used a binomial test against the alpha-level 0.05.

For Fig. 1h–l, we compensated for a baseline CPD by subtracting the average permuted CPD from the observed CPD. For Fig. 1m, we averaged the CPD for each factor across all recorded neurons, subtracted the average baseline in −200–0 ms as the stimulus onset period and then normalized the resulting CPD curve to its maximum value. For Extended Data Fig. 3a, we averaged the CPD for task identity factor across all recorded neurons then normalized the resulting CPD curve to its maximum value. This highlighted the timing of different factors during the trial.

## Quantifying population overlap for encoding of task variables

To determine whether task variables were encoded by overlapping neural populations (Extended Data Fig. 3b–k), we examined neurons for each task variable pair (for example, task identity and reward) to identify those that significantly encoded either or both variables using the GLM model explained above. Using a permutation test, we calculated the $P$ value for neurons encoding both variables. For a neural population of size $N_{\text{Total}}$, with $N_{\text{TaskVar1}}$ neurons encoding the first task variable, $N_{\text{TaskVar2}}$ neurons encoding the second, and $N_{\text{TaskVar1\&TaskVar2}}$ neurons encoding both, we iterated 10,000 random samples of $N_{\text{TaskVar1}}$ and $N_{\text{TaskVar2}}$ neurons from $N_{\text{Total}}$. In each iteration, we counted neurons encoding both variables. The likelihood of the observed $N_{\text{TaskVar1\&TaskVar2}}$ was estimated by calculating the proportion of permutations

yielding an equal or greater count of neurons encoding both task variables.

## Classifiers

To understand how task variables were represented in the neural population, we trained a set of binary classifiers to discriminate the vector of FRs across the neural population for two different categories of task variables (depending on the variable of interest). Classifiers were trained with the logistic regression algorithm (as implemented in MATLAB, fitclinear.m function). In brief, the linear classifier relates $\mathbf{x}$ (vector of neural responses) to $y$ (task labels, either +1 or −1) through a linear equation, with weights ($w$) and intercept ($b$):

$$f(x) = w^T \mathbf{x} + b$$

where $w$ and $b$ are optimized to minimize the logistic loss function: $\mathcal{L}[y, f(x)] = \frac{-1}{N} \sum_{i=1}^{N} [y_i \log(f(x_i)) + (1 - y_i) \log(1 - f(x_i))]$, where $N$ is the number of samples. Ridge ($L^2$) regularization with a regularization coefficient $\lambda = 1/60$ was used to minimize over-fitting. The classifier was trained with the Broyden–Fletcher–Goldfarb–Shanno algorithm.

## Construction of pseudopopulations

All classifiers were trained on pseudopopulation of responses constructed from neurons across all recording sessions. A separate pseudopopulation was constructed for each classifier analysis based on the trial types of interest (described below). To be included in the pseudopopulation, each neuron had to be recorded for a minimum number of trials ($N_{train} + N_{test}$) of each type of trial. The FR of neurons were concatenated to form a vector of neuron FRs (the pseudopopulation). Neural activity was aligned in time relative to either the sample onset or saccade onset, depending on the analysis.

To combine neurons across recording sessions into a single 'trial' of the pseudopopulation, we drew trials for each neuron with matching experimental conditions (that is, matched in terms of reward, colour and shape morph level, task identity and response direction). For example, when constructing a pseudopopulation for classifying the colour category, the first trial was constructed by concatenating the neural responses of neurons on trials that were rewarded, had a colour morph level of 100 and a shape morph level of 30. If a neuron did not have a trial with an exactly matching stimulus, then a trial that matched the colour category and reward would be randomly chosen.

## Cross-validation of classifiers

A separate logistic regression classifier was trained and tested on withheld trials for each timepoint. To estimate variability, the entire analysis was repeated 250 times, with each iteration involving a different partition of trials into training and testing sets. This new set of trials was randomly sampled with replacement (always ensuring test trials were separate from the train trials).

When testing the performance of a classifier trained on the same set of conditions (for example, training and testing colour categorization on the C2 task), the performance of the classifier was taken as the average performance across tenfold cross-validation. However, as detailed in the main text, many of the classifiers were trained and tested across conditions (for example, across different tasks or across trials during learning of the task). In this case, the test trials were randomly resampled ten times for each training set, and the classifier's performance was averaged across folds.

## Balancing classifier conditions

To avoid bias in the classifier, we balanced trial conditions to ensure the observed results were not due to other experimental factors. Conditions were balanced in three ways:

(1) Balanced congruency: classifiers were trained on an equal number of congruent and incongruent trials, resulting in equal number of trials for four stimulus types: green-bunny, green-tee, red-bunny and red-tee.

(2) Balanced reward: classifiers were trained on an equal number of rewarded and unrewarded trials, balancing the stimulus identity of the relevant dimension on each side of the classifier.

(3) Balanced response direction: classifiers were trained on an equal number of trials from each response direction on the task's response axis (for example, trials with response on axis 1 for the C1 task). This necessarily included error trials but balanced the number of each response direction on each side of the classifier.

## Classifying the colour category and shape category of stimuli

Colour and shape classifiers were trained to decode the stimulus colour category or shape category from the vector of activity across the pseudopopulation of neurons (Fig. 2a,b). A balanced number of congruent and incongruent stimuli were included in the training data to ensure colour and shape information could be decoded independently. Classifiers were trained for each task independently.

Most classifiers were trained on correct trials alone. This maximized the number of trials included in the analysis and ensured the animals were engaged in the task. As many of our analyses were tested across tasks, this mitigates concerns that motor response information might confound stimulus category information. However, to ensure response direction did not affect our analyses, we controlled for motor response by balancing response direction in a separate set of analysis (Extended Data Figs. 4a and 5c,d). Although this significantly reduced the number of trials, and required us to include error trials, qualitatively similar results were often observed. The total number of LPFC, STR, aIT, PAR and FEF neurons used for colour category classification was 403, 110, 195, 54 and 116, respectively. The total numbers of LPFC, STR, aIT, PAR and FEF neurons used for shape category classification were 480, 149, 239, 64 and 149, respectively.

## Classifying response direction

A separate set of response classifiers were trained to decode the motor response from the vector of activity across the pseudopopulation of neurons. Response direction was decoded within each axis (for example, UL versus LR for axis 1). When training and testing on the same task (Fig. 2c), we included only trials that the animal responded on the correct axis (for example, axis 1 for the S1 and C1 tasks).

For testing whether response direction could be decoded across axes (Extended Data Fig. 5d), we trained the classifier using trials from task C1, where the animal responded on axis 1, and then tested this classifier using trials from task C2, where the animal responded on axis 2, and vice versa. To control for stimulus-related information, we balanced rewarded and unrewarded trials for each condition and only included incongruent trials. This ensured that trials from the same stimulus category were present in both response locations. Similar results were seen when balancing for congruency and reward simultaneously, although the low number of incorrect trials for congruent trials resulted in a small set of neurons with the minimum number of train and test trials. The total number of LPFC, STR, aIT, PAR and FEF neurons used for response direction classification was 403, 110, 195, 54 and 116, respectively. Owing to limited number of trials, the total number of LPFC neurons for Extended Data Figs. 4b and 5d was 95 (only from animal Si).

## Using permutation tests to estimate the likelihood of classifier accuracy

We used permutation tests to estimate the likelihood the observed classifier accuracy occurred by chance. To create a null distribution of classifier performance, we randomly permuted the labels of training data 1,000 times. Importantly, only the task-variable of interest was permuted — permutations were performed within the set of trials with the same identity of other (balanced) task variables. For example, if

trials were balanced for reward, we shuffled labels within correct trials and within incorrect trials, separately. This ensured that the shuffling broke only the relationship between the task-variable of interest and the activity of neurons. Shuffling of the labels was performed independently for each neuron before building the pseudopopulation and training classifiers. Neurons that were recorded in the same session had identical shuffled labels. To stabilize the estimate of the classifier performance, the performance of the classifier on the observed and each instance of the permuted data were averaged over 10-folds and 20 to 50 novel iterations of each classifier. The null distribution was the combination of the permuted and observed values (total $n = 1,001$). The likelihood of the observed classifier performance was then estimated as its percentile within the null distribution. As described above, we used cluster correction to control for multiple comparisons across time.

### Testing classification across tasks
To quantify whether the representation of colour, shape, or response direction generalized across tasks (Fig. 2e,f,h,i), we trained classifiers on trials from one task and then tested the classifier on trials from another task. For example, to test cross-generalization of colour information across the C1 and C2 tasks, a colour classifier was trained on trials from the C2 task was tested on trials from the C1 task (Fig. 2f). To remove any bias due to differences in baseline FR across tasks, we subtracted the mean FR during each task from all trials of that task (at each timepoint). Similar results were observed when we did not subtract the mean FR.

### Cross-temporal classification
To measure how classifiers generalized across time, we trained classifiers to discriminate colour category (Fig. 2e and Extended Data Fig. 5b) or response direction (Fig. 2h) using 100 ms time bins of FR data, sliding by 10 ms. Classifiers trained on each time bin were then tested on all time bins of the test trials.

### Projection onto the encoding axis of classifier
To visualize the high-dimensional representation of task variables, we used the MATLAB predict.m function to project the FR response on test trials onto the one-dimensional encoding space defined by the vector orthogonal to the classification hyperplane. In other words, the projection measures the distance of the neural response vector to the classifier hyperplane. For example, to measure the encoding of each colour category in the C1 task, we projected the trial-by-trial FR onto the axis orthogonal to the hyperplane of the colour classifier trained during the C1 task.

### Quantifying the impact of task sequence on task discovery
The task-discovery period was defined as the first 110–120 trials after a switch in the task. The monkey's performance increased during this period, as they learned which task was in effect (Fig. 1f). As described in the main text, the animal's behavioural performance depended on the sequence of tasks (Fig. 4a) and we therefore analysed neural representations separately for two different sequences of tasks:
(1) C1–C2–C1 and S1–C2–S1 block sequence (same task transition): the task on axis 1 (C1 or S1) repeated across blocks. As shown in the main text, monkeys tended to perform better during these task sequences, as if they were remembering the previous axis 1 task.
(2) S1–C2–C1 and C1–C2–S1 block sequence (different task transition): the task on axis 1 changed across blocks. As shown in the main text (Fig. 4a), monkeys tended to perform worse on these task sequences.

As we were interested in understanding how changes in neural representations corresponded to the animals' behavioural performance, we divided the 'different task transition' sequences into two further categories: in 'Low Initial Performance' blocks, the animal's behavioural performance during the first 25 trials of the block was less than 50%, while on the 'high initial performance' blocks, the performance was greater than 50%.

Owing to constrains on number of neurons, different but overlapping population of neurons were used to quantify neural representations in C1–C2–C1 and S1–C2–C1 tasks sequences.

### Task belief representation during task discovery
To measure the animal's internal representation of the task (that is, their 'belief' about the task, as represented by the neural population), we trained a task classifier to decode whether the current task was C1 or S1. The classifier was trained on neural data from the last 75 trials of each task block, when the animal's performance was high (reflecting the fact that the animals were accurately estimating the task at the end of the block). The number of congruent and incongruent trials were balanced in the training dataset (32 trials: 4 trials for each of the four stimulus types, in each task). We included all C1 blocks regardless of their task sequence in training set. Only correct trials were used to train and test the classifier. To minimize the effect of neural response to visual stimulus, the classifier was trained on neural activity from the fixation period (that is, before stimulus onset). As we were interested in measuring differences between tasks, we did not subtract the mean firing rate before training the classifier.

The task classifier was tested on trials from the beginning of blocks of the C1 task. Test trials were drawn from windows of 40 trials, slid every 5 trials, during learning (starting from trial 1–40 to trial 71–110). Overlapping test and train trials from the same task were removed. In contrast to training set, testing was done separately for S1–C2–C1 and C1–C2–C1 task sequences (Fig. 4d). As we were interested in focusing on the learning of the C1 task, we only used S1–C2–C1 task sequences with low initial performance (see above). Classifiers were tested on pseudopopulations built from trials within each trial window, with a minimum of four test trials (one trial for each of the four stimulus congruency types from task C1). Neurons that did not include the required number of test trials for all trial windows were dropped. Note that, as we were using a small moving window of trials during task discovery, we had to trade-off the number of included neurons and the number of train/test trials for this analysis. Moreover, although the number of correct trials increased as the animal discovered the task in effect, we kept a constant number of train and test trials in each of the sliding trial windows.

To quantify the animal's belief above the current task, we measured the distance to the hyperplane of the C1/S1 task classifier (task belief encoding). For Fig. 4d, we averaged the performance of the classifier in pre-stimulus processing window (−400 ms to 0 ms).

### Correlation between task belief encoding and behavioural performance
To calculate the correlation between the animal's task belief during task discovery and their behavioural performance (Fig. 4e), we used Kendall's $\tau$ statistic with a permutation test to correct for autocorrelation in the signal (detailed below). This measurement was performed using data from each window of trials and the belief encoding, as estimated from the task classifier, on that same window of trials. As we are working with pseudopopulations of neurons, we estimated the behavioural performance for each window of trials during learning as the average of behavioural performance across all the trials in the window. The behavioural performance for each trial was taken as the average of the animal's behavioural performance during the previous 10 trials. This yielded one average performance for each of the 16 trial windows. The task belief was measured for the same trials, using the average distance to the task classifier hyperplane, averaged over the time period from 400 ms to 0 ms before the onset of the stimulus (note that, as described above, all of these trials are withheld from the

training data). Task belief was then taken as the average distance across all trials in a window of trials.

## Evolution of colour category representation during task discovery

We were interested in quantifying how the shared colour representation was engaged as the animal discovered the C1 task (Fig. 4f,g). To this end, we trained a classifier to categorize colour using the last 75 trials of the C2 task, when the animal's behavioural performance was high, and then tested it during discovery of the C1 task. The classifier was trained on only correct trials and the training data were balanced for congruent and incongruent stimuli (16 trials: 4 trials for each of the four stimulus congruency types). This ensured an equal number of correct trials for each congruency type across all trial windows, controlling for motor response activity during the task discovery period. As cross-axis response decoding between the C1 and C2 tasks was weak (Extended Data Fig. 5d), cross-task classifiers are capturing the representation of the colour category that is shared between tasks.

Similar to the task classifier described above, the shared colour classifier was tested using trials from the C1 task in a sliding window of 40 trials stepped 5 trials, in both the S1–C2–C1 and C1–C2–C1 sequences of tasks (Fig. 4f; as above, only low initial performance blocks were used for the S1–C2–C1 task sequences). We used four trials to test the classifier (one trial for each of the four stimulus congruency types from task C1). For Fig. 4g, we averaged the performance of the classifier in stimulus processing window (100 ms to 300 ms).

## Evolution of shape category representation during task discovery

We were interested in measuring the change in the representation of the stimulus' shape category as the animal learned the C1 task (Fig. 4i). Our approach followed that of the colour category representation described above, and so we only note differences here. We trained a classifier to categorize stimulus shape based on neural responses during the S1 task (limited to the last 75 trials of each block). To ensure the classifier was only responding to shape (and not motor response), we trained the classifier on a balanced set of correct (rewarded) and incorrect (unrewarded) trials (16 trials: 4 trials for each reward condition and for each shape category). The classifier was tested using trials from the C1 task (Fig. 4i). Note that we used the same C1 trials to quantify the representation of shared colour category, shape category and task belief for S1–C2–C1 task sequences. For Fig. 4j, we averaged the performance of the classifier in the window of time when stimuli were processed (100 ms to 300 ms after stimulus onset). The total numbers of LPFC neurons included for S1–C2–C1 and C1–C2–C1 task sequences to quantify the representation of shared colour category, shape category and task belief were 136 and 154, respectively.

## Evolution of response direction representation during task discovery

To measure the change in response direction representation as the monkey's learned the C1 task, we trained a classifier to categorize the response direction based on the neural response during the S1 task (Fig. 4k; last 75 trials of the block). As above, we used a balanced number of correct and incorrect trials to control for information about the stimulus (12 trials: 3 trials for each reward condition and each response location). The classifier was then tested on correct trials from the C1 task during task learning (Fig. 4k; as above: sliding windows of 40 trials, stepped 5 trials, in S1–C2–C1 and C1–C2–C1; tested on 4 trials: 1 for each reward condition and each response location). For Fig. 4l, we averaged the performance of the classifier in the window of time when response location was processed (200 ms to 400 ms after stimulus onset). The total numbers of LPFC neurons included for S1–C2–C1 and C1–C2–C1 task sequences to quantify the representation of shared response direction were 120 and 155, respectively.

## Classifier statistical test to detect trends during task discovery period

To quantify the statistical significance of trends in representations during task discovery (Figs. 4 and 5), we used trend-free prewhitening (TFPW)[62,63], as implemented in MATLAB[64]. This method helps to reduce serial correlation in time-series data to obtain robust statistical inference in the presence of trends. TFPW first detrends the time series by removing Sen's slope. It then prewhitens the time series by modelling autocorrelation with an autoregressive (AR) model (typically AR(1)) to produce residuals free of temporal dependencies. Finally, it adds back the original trend to generate processed time series. Both the Mann–Kendall statistics and Sen's slope were used to estimate the significance of trends in the data.

To ensure that our reported statistics are fully unbiased, we used the estimated Sen's slope on prewhitened data using TFPW to estimate the $P$ value of the observed data. To do so, we used permutation tests to estimate the likelihood the observed trend slope occurred by chance. To create a null distribution of classifier performance, we randomly permuted the labels of test data 250 times across all trial windows (trials 1–110 for Fig. 4 and 1–120 for Fig. 5). To stabilize the estimate of the classifier performance, we tested classifiers using the same set trained classifiers (for example, use same set of classifiers trained to decode colour category in C2 task to test colour category decoding in C1 task during task discovery). Furthermore, the performance of the classifier on the observed and each instance of the permuted data were averaged over 10-folds and 50 novel iterations of each classifier. The slope was calculated for each timepoint across all time trial windows to estimate the trend for observed and permuted classifier performances. The null distribution was the combination of the permuted and observed slope values (total $n = 251$). The likelihood of the observed slope was then estimated as its percentile within the null distribution. As described above, we used cluster correction to control for multiple comparisons across time.

To measure rank correlation between two random variables (for example, correlation between task belief and behavioural performance, Fig. 4e), we used Kendall's $\tau$. As TFPW requires a monotonic time variable and cannot be applied here, we computed Kendall's $\tau$ for both observed and permuted datasets. $z$-score values of observed data against permuted values were reported to account for autocorrelation inflation.

## Transfer of information between subspaces

As described in the main text, we were interested in testing the hypothesis that the representation of the stimulus colour in the shared colour subspace predicted the response in the shared response subspace on a trial-by-trial basis (Fig. 3c–e and Extended Data Fig. 7a–c). To this end, we correlated trial-by-trial variability in the strength of encoding of colour and response along four different classifiers (using Pearson's correlation as implemented in MATLAB's corr.m function).

First, as described above, a shared colour classifier was trained to decode colour category from the C2 task and tested on trials from the C1 task. Training data were balanced for correct and incorrect trials (rewarded and unrewarded trials, 16 trials: 4 trials for each reward condition and each colour category). Test trials were all correct trials (2 trials: 1 trial for each colour category). Both train and test trials were drawn from the last 50 trials from the block (to ensure the animals were performing the task well). The total number of LPFC neurons included for this classifier was 63 (only from monkey Si, owing to constraints on the number of trials).

Similarly, a second shared response classifier was trained on trials from the S1 task and then tested on the same set of test trials as the shared colour classifier.

A third classifier was trained to decode the response direction on axis 2, using a balance of correct and incorrect trials from the C2 task.

This classifier was tested on the same set of test trials as the shared colour classifier.

Finally, a fourth classifier was trained to decode the shared colour representation but was now trained on correct trials from the C1 task (12 trials: 3 trials for each of the four stimulus congruency types) and tested on correct trials from C2 task (2 trials: 1 trial for each colour category). The total number of LPFC neurons included for this classifier was 101 (only from animal Si, owing to constraints on the number of trials).

To account for the arbitrary nature of positive and negative labels, we calculated the magnitude of encoding by flipping the encoding for negative labels. All four classifiers were trained on 2,000 iterations of training set trials. Note, that the first three classifiers were tested on the same test trials, allowing for trial-by-trial correlations to be measured. Furthermore, all four classifiers were trained over time, enabling us to measure the cross-temporal correlation between any pairs of classifiers.

Together, these four classifiers allowed us to test three hypothesized correlations that reflect the transfer of task-relevant stimulus information into representations of behavioural responses. First, one might expect that, on any given trial before the start of saccade, the strength and direction of the shared colour representation, as estimated by the distance to the hyperplane of the first classifier, should be correlated with the shared response, as estimated by the distance to the hyperplane of the second classifier. This correlation is seen in Fig. 3c.

Second, during the performance of the C1 task, one would expect that before the start of saccade the shared colour representation should not be correlated with the response on the Axes 2 predicted by the C2 task. This is quantified by correlating the distance to the hyperplane of the first classifier and the distance to the hyperplane of the third classifier, as seen in Fig. 3d.

However, one would expect these representations to be correlated when the animal is performing the C2 task. This is quantified by correlating the distance to the hyperplane of the fourth classifier and the distance to the hyperplane of the third classifier, as seen in Extended Data Fig. 7a.

### Distance along colour and shape encoding axes during the discovery of the task

To understand how the geometry of the neural representation of the colour and shape of the stimulus evolved during task discovery (Fig. 5), we measured the distance in neural space between the two prototype colours (red and green) and the two prototype shapes (tee and bunny). Distance was measured along the encoding axis for each stimulus dimension (that is, the axis that is orthogonal to the colour and shape classifiers, described above). So, for each test trial, the distance along the colour encoding axis was:

$$\text{colour encoding distance}$$
$$= \text{Avg}\begin{pmatrix} \text{abs(encoding(red bunny)} - \text{encoding(green bunny)),} \\ \text{abs(encoding(red tee)} - \text{encoding(green tee))} \end{pmatrix};$$

This approach enabled us to calculate the distance along red/green colours while controlling for differences across shapes. The colour encoding distance was estimated for 250 iterations of the classifiers, enabling us to estimate the mean and standard error of the distance. A similar process was followed for estimating the shape encoding distance.

To calculate the colour and shape encoding distance during task discovery (Fig. 5a,b), we calculated the distance above in the sliding window of 45 trials, stepped 5 trials, after the switch into the C1 task during the S1–C2–C1 sequences of tasks with low initial performance.

### CPI

CPI was defined as the log of the ratio of average colour encoding distance and average shape encoding distance described above:

$$\text{CPI} = \log\left(\frac{\text{avg(Colour encoding distance)}}{\text{avg(Shape encoding distance)}}\right)$$

To calculate the CPI for each task (Fig. 5c), we computed the shape and colour encoding distance using all trials of a given task. To measure the shape distance for all three tasks, we trained a classifier to categorize stimulus shape based on neural responses during the S1 task (limited to the last 75 trials of each block). To ensure that the classifier was responding only to shape (and not motor response), we trained the classifier on a balanced set of correct (rewarded) and incorrect (unrewarded) trials (20 trials: 5 trials for each reward condition and for each shape category). Shape encoding for each task was calculated using four trials to test the classifier (one trial for each of the four stimulus congruency types).

To measure colour distance in the S1–C2–C1 task, we trained colour category classifier on correct trials of the C1–C1–C2 task, balancing for congruent and incongruent stimuli (20 trials: 5 trials for each of the four stimulus congruency types). Colour encoding for each task was calculated using four trials of that task to test the classifier (one trial for each of the four stimulus congruency types).

To track changes in the relative strength of colour and shape information during task discovery, we calculated CPI in a sliding window of 45 trials (slid every 5 trials starting from trial 1–45 to trial 76–120) after the monkey's switched into the C1 task (during the S1–C2–C1 sequences of tasks, with low initial performance). Note, as we controlled for motor response when calculating shape and colour distance, the CPI is not affected by motor response information[65].

### Correlation between task belief encoding and CPI

To correlate the CPI with task belief encoding, we used the same set of C1 test trials to calculate both CPI and estimate the task identity using the task classifier described above. CPI values in 100 ms–300 ms after stimulus onset and belief encoding values in 400 ms to 0 ms before stimulus onset were averaged for all trials in each trial window and the Mann–Kendall correlation between the resulting two vectors was calculated (Fig. 5e).

### Quantifying suppression of axis representation

We trained a response axis classifier to decode whether the current task axis was axis 1 or axis 2. The classifier was trained on neural data from all trials of C1 and C2 tasks with an equal number of trials from each response direction on task specific axis (for example, equal trials for UL and LR response on axis 1 for C1 task, 36 trials: 9 trials for each response direction, in each task). All correct and incorrect trials were used to train and test the classifier. To capture the response period, the classifiers were trained on the number of spikes between 200 ms and 450 ms after stimulus onset. As we were interested in measuring the difference in neural activity between axis of responses, we did not subtract the mean FR before training the classifier. For Extended Data Fig. 10d,f–i, the classifier was trained to decode response axis using S1 and C2 trials.

To create a null distribution for classifier weights, we randomly permuted the response axis labels for a given response direction (1,000 iterations).

### Selectivity for axis of response

We used the classifier $\beta$ weights to group neurons according to their axis selectivity. Neurons with significantly negative $\beta$ weights were categorized as selectively responding to axis 1. Neurons with significantly positive $\beta$ weights were categorized as selectively responding to axis 2. Neurons without significant $\beta$ weights were categorized as non-selective (Extended Data Fig. 10a). To determine the significance of a neuron's classifier weight, we compared the observed $\beta$ weight to a null distribution (two-tailed permutation test).

To quantify the suppression of neural activity for each category of neurons, we averaged FRs of neurons in each category during trials of the C1 task when the animal responded on axis 1 or axis 2 (Fig. 5g; Extended Data Fig. 10e shows the same analysis for the S1 task). This meant including all trials when the animal responded on axis 1 (both correct and error) and all trials when the animal responded on axis 2 (all errors). As the monkeys rarely responded on incorrect axis (Extended Data Fig. 2g), the number of trials was limited and so neurons without any error trials on axis 2 were excluded.

Note that, although neurons were sorted by their activity during the animal's response (200–450 ms after stimulus onset), we observed suppression across the entire trial (Fig. 5g). Furthermore, similar results were seen when a neuron's axis selectivity was quantified on withheld trials (for example, in Extended Data Fig. 10d–i, axis selectivity was defined on S1 and C2 tasks and applied to C1 trials).

## Decoding response axis during the discovery of the task

We trained a response axis classifier to decode whether the current task axis was axis 1 or axis 2. The classifier was trained on neural data from the last 75 trials of the C1 and C2 tasks, when the monkey's performance was high (reflecting the fact that the animals were accurately responding to the correct axis at the end of the block). The classifier was trained on an equal number of trials from each direction on the response axis (for example, equal trials for UL and LR response on axis 1 for C1 task; 36 trials: 9 trials for each response direction, in each task). Correct and incorrect trials were used to train and test the classifier. To remove the effect of stimulus processing, the classifier was trained on neural activity from the fixation period (that is, −400 to 0 ms before stimulus onset). As we were interested in measuring differences between axis of responses, we did not subtract the mean FR before training the classifier.

The task classifier was tested on trials from the beginning of blocks of the C1 task. Test trials were drawn from windows of 10 trials, slid every 3 trials, during task discovery (starting from trial 1–10 to trial 66–75). Overlapping test and train trials from the same task were removed. Classifiers were tested on pseudopopulations built from trials within each trial window (two trials: one trial for each response direction on axis one from task C1).

## Quantifying the angle between classifier hyperplanes

To quantify the similarity between the decision boundaries of classifiers trained on different tasks, we calculated the angle between the hyperplanes defined by their weight vectors. Each classifier produced a weight vector $\mathbf{w}_i$ corresponding to the hyperplane that separates the data points according to the respective task. We averaged the resulting hyperplane across resampling repetitions of trials (250 iterations).

The angle $\theta$ between two average hyperplanes, defined by weight vectors $\mathbf{w}_i$ and $\mathbf{w}_j$, was calculated using the cosine similarity, which is given by:

$$\cos(\theta) = \frac{\mathbf{w}_i \cdot \mathbf{w}_j}{\|\mathbf{w}_i\| \|\mathbf{w}_j\|}$$

The angle $\theta$ was then obtained by taking the inverse cosine of the cosine similarity. Angles close to 0° indicate that the classifiers' hyperplanes are nearly parallel, implying similar decision boundaries across the tasks. Conversely, angles close to 90° suggest orthogonal hyperplanes, indicating distinct decision boundaries.

To measure the angle within the C1 task, we calculated the angle between classifiers trained to classify colour category on the last 50 trials of C1 task in both the C1–C2–C1 task sequence and the S1–C2–C1 task sequence. This ensured that there were no overlapping trials between training samples. Only correct trials were used in this analysis and training and test trials were balanced for congruent and incongruent stimuli (20 trials: 5 trials for each of the four stimulus congruency types).

To measure the angle between the colour category classifier in the C1 task and the C2 task, we trained the classifier on last 50 trials of task C1 and task C2.

## TDR analysis

The TDR analysis requires multiple steps[21]. Here, we describe each step in turn.

**Linear regression.** We used a GLM to relate the activity of each neuron ($y$) to the task variables at each moment of time. The full model was:

$$\begin{aligned} y = \beta_0 + \beta_1 &\times \text{stimulus colour morph level} + \beta_2 \\ &\times \text{stimulus shape morph level} + \beta_3 \times \text{time} + \beta_4 \times \text{task S1} + \beta_5 \\ &\times \text{task C1} + \beta_6 \times \text{task C2} + \beta_7 \\ &\times \text{motor response direction on axis 1} + \beta_7 \\ &\times \text{motor response direction on axis 2} + \beta_9 \times \text{reward} \end{aligned}$$

where $y$ is the FR of each neuron, $z$ scored by subtracting the mean response from the FR at each timepoint and dividing it by the s.d. The mean and s.d. were computed across all trials and timepoints for each neuron.

$\beta_0, \beta_1, …, \beta_9$ are the regression coefficients corresponding to each predictor. Predictors were: stimulus colour morph level, coded as −1 for 0% morph level, −0.5 for 30% and 170% morph levels, +0.5 for 70% and 130% morph levels, and +1 for 100% morph level; stimulus shape morph level, coded as −1 for 0% morph level, −0.5 for 30% and 170% morph levels, +0.5 for 70% and 130% morph levels and +1 for 100% morph level; time, indicating the temporal progression within a recording session, normalized between 0 and 1 (continuous variable); task identity, indicating the identity of the task (categorical variable: S1, C1, C2); motor response direction on axis 1, indicating the direction of the motor response (categorical variable; −1 for UL, +1 for LR); motor response direction on axis 2, indicating the direction of the motor response (categorical variable; −1 for LL, +1 for UR); reward, indicating whether a reward was received following the response (binary variable; 1 for reward, 0 for no reward).

The GLM coefficients ($\beta$) were estimated using maximum-likelihood estimation as implemented by MATLAB's lassoglm.m function. Independent models were fit for each timepoint. To address potential overfitting, we applied Lasso ($L_1$) regularization to the GLM weights with regularization coefficients (lambda) values of 0.0015. Correct and incorrect trials were used in fitting the model.

**Population average responses.** We estimated the average response of the neural population to each task variable. To calculate the average neural response to each colour and shape morph level, we separately averaged trials for within each morph level (0, 30, 70, 100, 130, 170) for each task. To calculate the average neural response for each response direction, we separately averaged trials for within each response direction (UL, UR, LL or LR) for each task. For all task variables, we smoothed the resulting response in time with a 150 ms Boxcar. Finally, we $z$-scored the smoothed average for a given unit by subtracting the mean across times and conditions, and by dividing the result by the corresponding s.d.

**PCA.** We used principal component analysis (PCA) (as implemented in MATLAB, pca.m function) to find the dimensions in the state space that captured most of the variance of the population average response and to mitigate the effect of noise. We concatenated the average response across conditions to build matrix $X$ of size $(N_{condition} \times T) \times N_{unit}$, where $N_{condition}$ is the total number of conditions and $T$ is the number of time samples. We used the first $N_{PCA}$ that captured 90% of the explained variance to define a de-noising matrix $D$ of size $N_{unit} \times N_{unit}$.

**Regression subspace.** We used the regression coefficients described above to identify dimensions in the state space containing task related variance. We used four task variables to define this space: colour morph level, shape morph level, response direction on axis 1 and response direction on axis 2. For each of these task variables, we built coefficient vectors $\beta_{v,t}(i)$ corresponding to regression coefficient for task variable $v$, time $t$ and unit $i$. We then denoised each variable vector by projecting it to the subspace spanned by $N_{\text{PCA}}$ principal components from the population average defined above.

$$\beta_{v,t}^{\text{PCA}} = D\beta_{v,t}$$

We computed $t_v^{\text{max}}$ for each task variable where norm of $\beta_{v,t}^{\text{PCA}}$ matrix had its maximum value. This defined the time-independent, denoised regression vectors.

$$\beta_v^{\text{max}} = \beta_{v,t_v^{\text{max}}}^{\text{PCA}}$$

With $t_v^{\text{max}} = \text{argmax}_t \|\beta_{v,t}^{\text{PCA}}\|$. To compute orthogonal axes of colour, shape and response direction on axis 1 and response direction on axis 2, we orthogonalized the regression vectors $\boldsymbol{\beta}_v^{\text{max}}$ using $QR$ decomposition.

$$B^{\text{max}} = QR$$

Where the first four columns of $Q$ are orthogonalized regression vectors $\boldsymbol{\beta}_v^{\perp}$ of four task variables that comprise the regression subspace.

To then estimate the representation of task-related variables in the neural population, we projected the average population response (described above) for each colour morph level, shape morph level, and response direction onto these orthogonal axes.

$$p_{v,c} = \boldsymbol{\beta}_v^{\perp T} X_c$$

Where $p_{v,c}$ is the time series with length $T$ for each morph level in a specific task.

Note that because TDR orthogonalizes task features, it can control for motor response when estimating the neural response to colour and shape.

**CPI using TDR.** To measure the distance between the two prototype colours (red and green) and the two prototype shapes (tee and bunny). We first projected the average PSTH for each protype object onto the orthogonal encoding axes of colour and shape.

$$p_{\text{colour,protoype}} = \boldsymbol{\beta}_{\text{colour}}^{\perp T} X_{\text{prototype}}$$

$$p_{\text{shape,protoype}} = \boldsymbol{\beta}_{\text{shape}}^{\perp T} X_{\text{prototype}}$$

Distance was measured along the orthogonal encoding axis of shape and colour using projected responses:

colour encoding distance

$$= \text{Avg} \begin{pmatrix} \text{abs}\big(p_{\text{colour,red bunny}} - p_{\text{colour,green bunny}}\big), \\ \text{abs}\big(p_{\text{colour,red tee}} - p_{\text{colour,green tee}}\big) \end{pmatrix}$$

$$\text{shape encoding distance} = \text{Avg} \begin{pmatrix} \text{abs}\big(p_{\text{shape,red bunny}} - p_{\text{shape,red tee}}\big), \\ \text{abs}\big(p_{\text{shape,green bunny}} - p_{\text{shape,green tee}}\big) \end{pmatrix}$$

CPI was defined as the log of the ratio of the average colour encoding distance and average shape encoding distance as described above:

$$\text{CPI} = \log\left(\frac{\text{avg(colour encoding distance)}}{\text{avg(shape encoding distance)}}\right)$$

Extended Data Fig. 9f reports the average CPI for 100 ms to 300 ms since stimulus onset.

**Quantifying the angle between colour category and response direction subspace across tasks using TDR**

To quantify the angle between colour category subspace we used orthogonal task variable vectors defined by TDR. Similar to fitting a regression model for TDR, we first fit a regression model that included separate colour, shape and response axis weights for each task.

$y = \beta_0 + \beta_1 \times$ C1 task stimulus colour morph level

$\quad + \beta_2 \times$ C2 task stimulus colour morph level

$\quad + \beta_3 \times$ S1 task stimulus colour morph level

$\quad + \beta_4 \times$ C1 task stimulus shape morph level

$\quad + \beta_5 \times$ C2 task stimulus shape morph level

$\quad + \beta_6 \times$ S1 task stimulus shape morph level

$\quad + \beta_8 \times$ S1 task motor response direction on axis 1

$\quad + \beta_9 \times$ C1 task motor response direction on axis 1

$\quad + \beta_{10} \times$ C2 task motor response direction on axis 2 $+ \beta_{11} \times$ time

$\quad + \beta_{12} \times$ task S1 $+ \beta_{13} \times$ task C1 $+ \beta_{14} \times$ task C2 $+ \beta_{15} \times$ reward.

Models were fit on 200 resamples, each randomly drawing 75% of all trials for each individual task. For 'from switch' conditions, the first 50 trials after switch to each task were used and, for 'to switch' conditions, the last 50 trials in the blocks were used. Similar to TDR method described above, we used QR decomposition to find the orthogonal axes encoding each task variable, but now specific to each task (colour morph level, shape morph level, task identity, reward and task specific axis of response):

$$B^{\text{max,C1 task}} = Q_{\text{C1}}R_{\text{C1}}, B^{\text{max,C2 task}} = Q_{\text{C2}}R_{\text{C2}}, B^{\text{max,S1 task}} = Q_{\text{S1}}R_{\text{S1}}$$

For each resampling, the angle between orthogonal axis of colour of pairs of tasks (C1–C2, C1–S1 and C2–S1) was measured as

$$\cos(\theta) = \frac{\boldsymbol{\beta}_{\text{colour,task } i}^{\perp} \cdot \boldsymbol{\beta}_{\text{colour,task } j}^{\perp}}{\left\|\boldsymbol{\beta}_{\text{colour,task } i}^{\perp}\right\| \left\|\boldsymbol{\beta}_{\text{colour,task } j}^{\perp}\right\|}$$

Where $i$ and $j$ are task identity of the tasks. Within-task angles (Extended Data Fig. 9c,d) were computed by randomly taking 200 pairs of resampling regression coefficient repetitions, and finding the orthogonal axes explained above. All angles were wrapped to 90 deg. This was because the process of orthogonalization with QR decomposition might result in vectors that are mathematically equivalent but with a flipped sign compared to the input vectors owing to numerical choices or conventions.

We used a permutation test to assess whether the angle between pairs of tasks was significantly larger than expected by chance. The observed angle was estimated by fitting the regression model explained above to all trials. To compute a null distribution for angles, we generated 1,000 permuted datasets by randomly permuting the predictor values relative to the neural activity and refitting the model. For each permuted dataset, we computed the angle between task pairs as described above. The likelihood of the observed angle was then estimated by computing the proportion of permuted datasets that yielded an angle smaller than or equal to that of the observed angle.

**Quantifying whether task representations transfer across blocks**
To test whether the task representation was maintained across blocks (Fig. 4b), we trained a classifier to discriminate between C1 and S1 trials using the last 50 trials of the block. We then tested this classifier in three ways. First, to quantify the information about the identity of C1 versus S1 task, we tested this classifier on withheld trials from last 50 trials of C1 and S1 tasks (Fig. 4b (1)). Second, to quantify how much the representation of C1 and S1 tasks transferred to the C2 task, we tested whether the classifier could discriminate between C1 and S1 tasks during the first 50 trials of the C2 task (that is, comparing C1–C2 versus S1–C2). Third, to quantify how much the representation of C1 and S1 tasks was transferred to C1 or S1 in the next block, we tested the classifier on the first 50 trials after the switch in C1 and S1 tasks.

For all training and testing, the total number of spikes in the period −400 to 0 ms from stimulus onset for each trial were used to build a pseudopopulation. Only correct trials were included in this analysis, and the training and test trials were balanced for congruent and incongruent stimuli (36 trials: 9 trials for each of the four stimulus congruency types). To compute a null distribution, we generated 1,000 permuted datasets by randomly permuting all task identity values relative to the neural activity and refitting the classifier.

## Reporting summary

Further information on research design is available in the Nature Portfolio Reporting Summary linked to this article.

## Data availability

Processed data are publicly available at FigShare[66] (https://doi.org/10.6084/m9.figshare.30276238.v1).

## Code availability

Custom MATLAB analysis functions are publicly available at Zenodo[67] (https://doi.org/10.5281/zenodo.17274345).

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

**Acknowledgements** We thank B. Morea and N. Rajagopalan for assistance with monkeys; S. Ostojic, C. Brody, T. Engel, C. M. Langdon and H. Maleki for discussions; I. Wahle and T. Engel for help with revision; T. Engel, C. Jahn, Q. He, P. Iamshchinina, I. Wahle, S. Akers-Campbel and J. Park for feedback on the manuscript; and the Princeton Primate Research Resources staff for support. This work was supported by National Institutes of Health (NIH) grants R01MH129492 (to T.J.B.) and NIH 5T32MH065214 (to A.A.).

**Author contributions** T.J.B. and S.T. conceived the project. S.T. collected experimental data with assistance from N.T.M., M.U. and T.J.B.; S.T. analysed the data with input from T.J.B., A.A., F.M.B., M.G.M. and N.D.D.; S.T. wrote first draft of the paper. S.T., T.J.B., F.M.B., M.G.M., A.A., N.D.D., N.T.M. and M.U. edited the paper. T.J.B. acquired funding. T.J.B. supervised the project.

**Competing interests** The authors declare no competing interests.

**Additional information**
**Correspondence and requests for materials** should be addressed to Sina Tafazoli or Timothy J. Buschman.

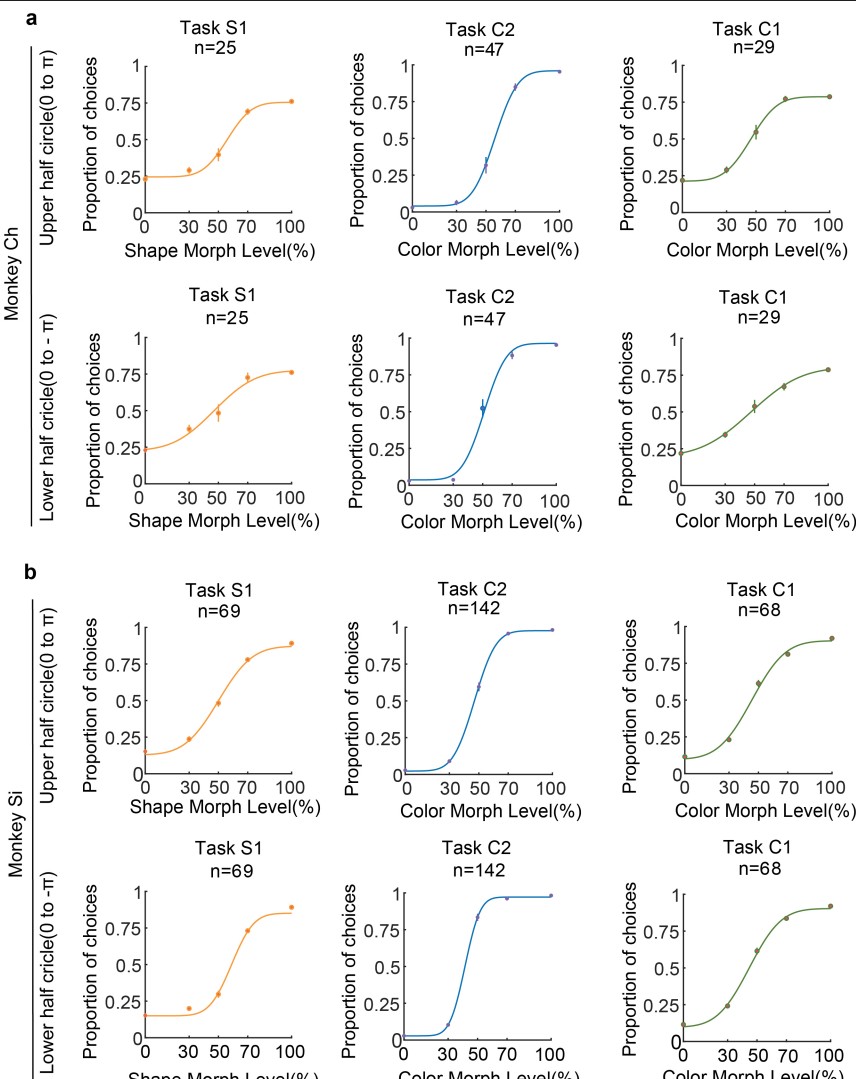

**Extended Data Fig. 1 | Both monkeys perform all three tasks accurately.**
Follows Fig. 1e. Average psychometric curve for (**a**) monkey Ch and (**b**) monkey Si. Behavioural performance is shown for the last 102 trials of blocks of S1 and C1 tasks and last 51 trials of blocks of C2 task. N indicates the number of blocks per task. Error bars show mean ± s.e.m.

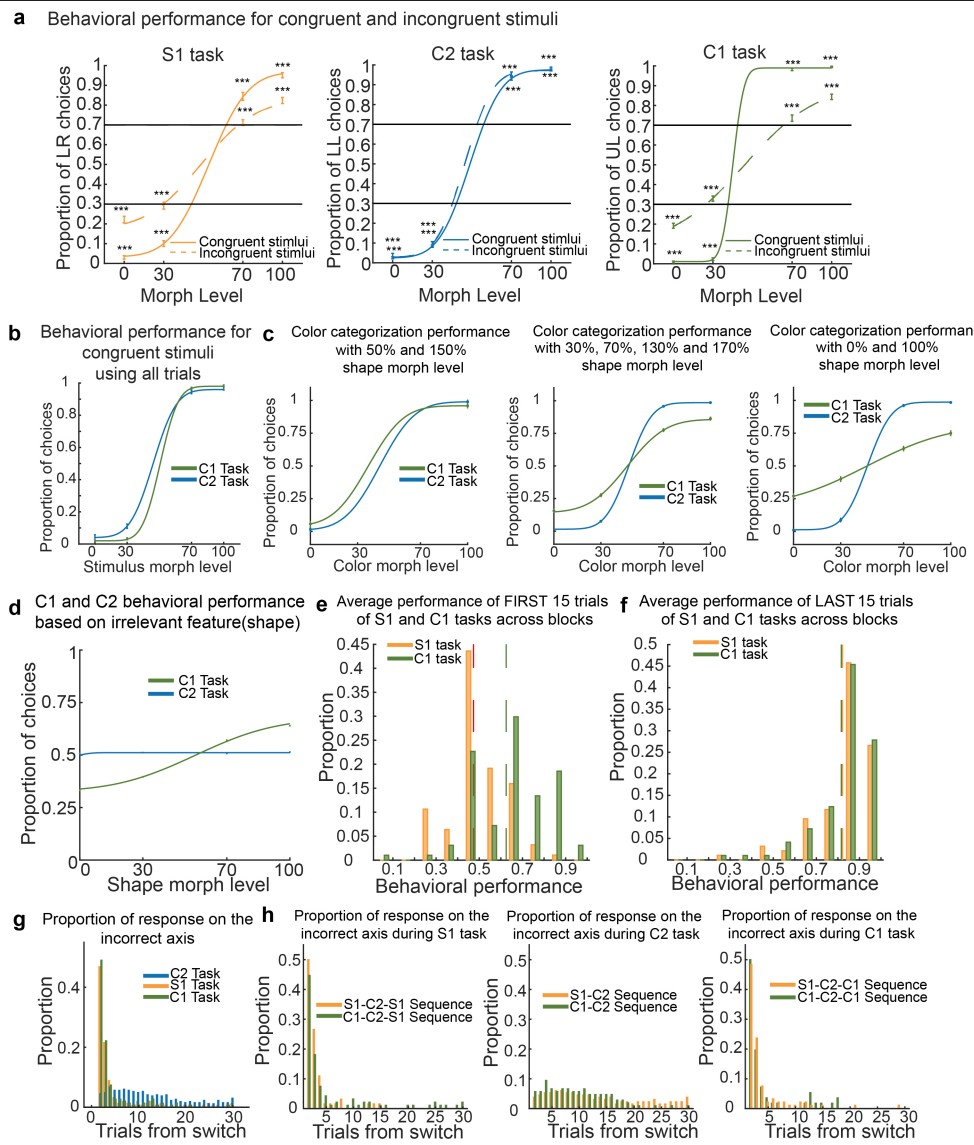

**Extended Data Fig. 2 | Behaviour varied for congruent and incongruent stimuli.** Follows Fig. 1e. **a**, Psychometric curve for S1 (left), C2 (middle), and C1 (right) tasks for incongruent (dashed) and congruent (solid) stimuli. *, **, and *** indicate p < 0.05, 0.01, and 0.001, respectively, one-sided binomial test. p < 10^−16 for all data points. Since congruency is undefined for the 50% morph line, there are no corresponding data points available. Error bars show mean ± s.e.m. n = 94/97/189 blocks of S1/C1/C2 tasks, respectively. **b**, Psychometric curve for congruent trials for C1 and C2 tasks using all trials in the block. Since congruency is undefined for the 50% morph line, there are no corresponding data points available. Behavioural performance on congruent stimuli, where there is no conflict between the C1 and S1 tasks, is similar between the C2 and C1 tasks. This is consistent with the idea that the relatively lower behavioural performance during the C1 task is due to interference from the S1 task (possibly due to uncertainty over which task to perform). Error bars show mean ± s.e.m. n = 94/97/189 blocks of S1/C1/C2 tasks, respectively. **c**, Psychometric curve for

colour categorization performance when the shape morph level of the stimulus was ambiguous (50%/150%, left), intermediate (30%/70%/130%/170%, middle) and prototypical (0%/100%, right). Error bars show mean ± s.e.m. n = 94/97/189 blocks of S1/C1/C2 tasks, respectively. **d**, Psychometric curve for C1 and C2 tasks when proportion of choices was computed based on responses according to irrelevant feature (shape). Shape affects behavioural responses during the C1 task but not the C2 task. This is again consistent with the idea that the decrease in behavioural performance during the C1 task is due to interference from the S1 task. **e-f**, Comparison of behavioural performance of S1 (orange) and C1 (green) tasks **e**, in the first 15 trials of the block (after switch) and **f**, for the last 15 trials of the block (before switch). Red and green dotted line show mean behavioural performance for S1 and C1 tasks, respectively. **g**, Proportion of responses on incorrect axis for C1, C2 and S1 tasks. **h**, Proportion of responses on incorrect axis during the S1 task (left), C2 task (middle) and C1 task (right) did not depend on the sequence of tasks (S1-C2 in orange and C1-C2 in green).

**Extended Data Fig. 3 | Task variables are widely distributed in neurons across multiple brain regions. a**, Follows Fig. 1m. Normalized CPD for task identity averaged for all regions. Lines and shading show mean± s.e.m. Task identity representation increased by 51.5% from baseline during the stimulus processing period. Previous research has reported similar dynamics in task representations, suggesting that as an animal prepares to process and respond to a visual stimulus, task-relevant information increases to facilitate task execution[68]. **b-k**, Individual neurons represent multiple task variables across regions. Each panel shows the proportion of neurons that significantly encode two task variables (left and middle columns) and encode both (right column). For all panels, *, **, and *** indicate p < 0.05, 0.01, and 0.001, respectively, all one-sided permutation tests. **b-e**, Response direction and **b**, reward, **c**, task identity, **d**, colour category, and **e**, shape category. **f-h**, reward and **f**, task identity, **g**, colour category, and **h**, shape category. **i-k**, task identity and **i**, colour category and **j**, shape category. **k**, colour category and shape category.

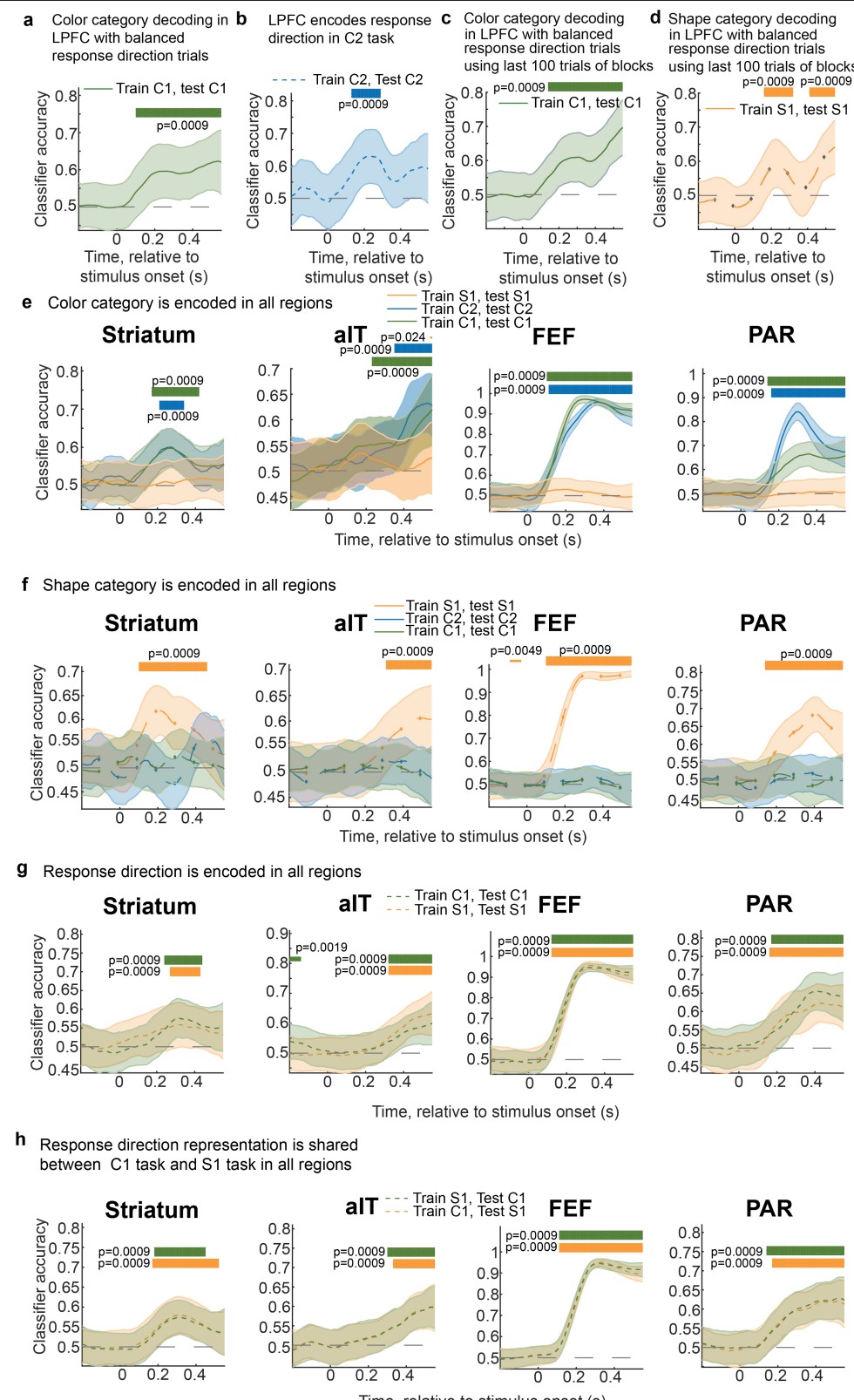

**Extended Data Fig. 4** | See next page for caption.

**Extended Data Fig. 4 | Colour category, shape category and response direction were encoded in all regions. a**, Follows Fig. 2a. Time course of accuracy of classifier trained to decode the colour category of the stimulus during the C1 task based on neural activity in LPFC. To control for movement, the number of trials with each response direction were balanced within each colour category. Only trials with responses on the correct response axis were included (e.g., Axis 1 for task C1). Lines and shading show mean ± s.e.m. classification accuracy after stimulus onset. Distribution reflects 250 iterations of classifiers. Horizontal bars along top indicate above-chance classification (p < 0.05, 0.01, and 0.001 for thin, medium, and thick lines, respectively, two-sided permutation test with cluster mass correction for multiple comparisons). **b**, Follows Fig. 2c. Time course of accuracy of classifier trained to decode response direction during the C2 task from neural activity in LPFC. Classifier was trained on a balance of rewarded and unrewarded trials. Lines and shading show mean ± s.e.m. classification accuracy after stimulus onset. Distribution reflects 250 iterations of classifiers. Horizontal bars along top indicate above-chance classification (p < 0.05, 0.01, and 0.001 for thin, medium, and thick lines, respectively; two-sided permutation test with cluster mass correction for multiple comparisons). **c**, As in panel **a**, but showing time course of accuracy of classifier trained to decode colour category within C1 task. Trials were drawn from the last 100 trials in the block and balanced for reward. **d**, As in panel **c**, but for shape category within S1 task. **e**, Follows Fig. 2a. Time course of accuracy of classifier trained to decode the colour category of the stimulus. Classifiers were trained for each task and tested on withheld trials of the same task (coloured lines). Classifiers were trained separately for each brain region: Striatum, aIT, FEF, and PAR in each column, moving left to right. Lines and shading show mean ± s.e.m. classification accuracy after stimulus onset. Distribution reflects 250 iterations of classifiers. Horizontal bars along top indicate above-chance classification (p < 0.05, 0.01, and 0.001 for thin, medium, and thick lines, respectively; two-sided permutation test with cluster mass correction for multiple comparisons). **f**, Follows Fig. 2b. As in panel **e**, but for shape category information. **g**, Follows Fig. 2c. Time course of accuracy of classifier trained to decode response direction for S1 and C1 tasks. Classifiers were trained separately for each brain region: Striatum, aIT, FEF, and PAR in each column, moving left to right. Classifiers were trained on a balance of rewarded and unrewarded trials. Lines and shading show mean ± s.e.m. classification accuracy after stimulus onset. Distribution reflects 250 iterations of classifiers. Horizontal bars along top indicate above-chance classification (p < 0.05, 0.01, and 0.001 for thin, medium, and thick lines, respectively; two-sided permutation test with cluster mass correction for multiple comparisons). **h**, Follows Fig. 2i. As in panel **g**, but showing time course of accuracy of classifier trained on S1 task (green) or C1 task (orange) and then tested on the other task (C1 and S1, respectively).

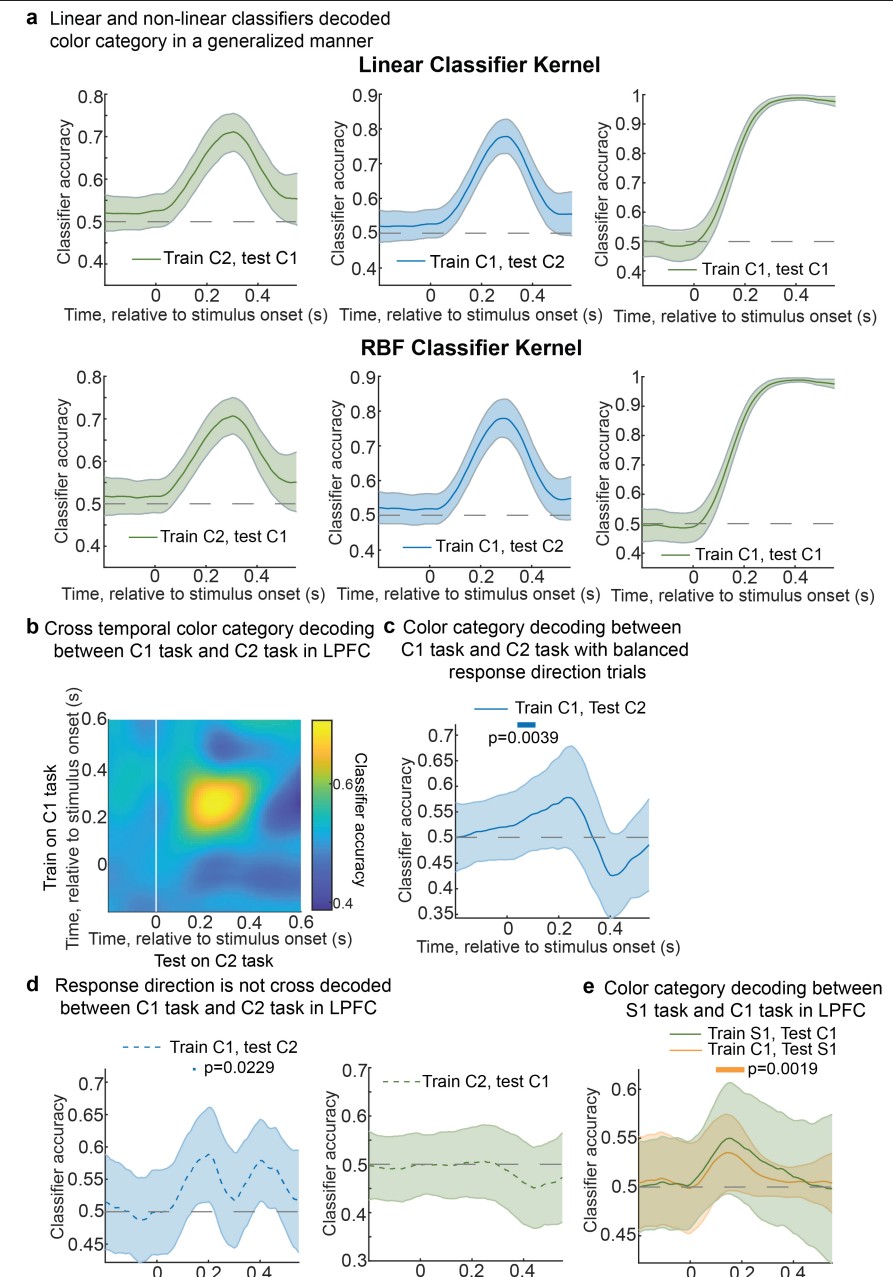

**Extended Data Fig. 5 | Colour category was cross decoded between tasks in LPFC. a**, Accuracy of classifier trained to decode colour category from LPFC neural activity when trained on C2 task and tested C1 task (left); trained on C1 task and tested on C2 task (middle); and trained on C1 task and tested on C1 task (right). Top row shows accuracy when using a linear kernel for classifier and bottom row shows accuracy when using radial basis function (RBF) kernel for classifier. Lines and shading show mean ± s.e.m. over time. Distribution reflects 250 resampled classifiers (see Methods for details). **b**, As in Fig. 2e, but showing the cross-temporal cross-task classification accuracy in decoding shared colour category in LPFC when training on C1 task trials and testing on C2 task trials. **c**, Time course of accuracy of classifier trained to decode colour category on trials of the C1 task, while balancing motor response (as in Extended Data Fig. 4a). Classifier is then tested on the C2 task. Note, reduction in decoding accuracy reflects the reduced number of trials and introduction of error trials. Lines and shading show mean ± s.e.m. classification accuracy after stimulus onset. Distribution reflects 250 iterations of classifiers. Horizontal bars along top indicate above-chance classification (p < 0.05, 0.01, and 0.001 for thin, medium, and thick lines, respectively; two-sided permutation test with cluster mass

correction for multiple comparisons). **d**, Classifiers are unable to decode response direction between axes. Classifiers were trained to decode response direction from LPFC neural activity during the C1 task (left) or C2 task (right) and then tested on accuracy to decode the matched hemifield location during the C2 task (left) or C1 task (right). Lines and shading show mean ± s.e.m. classification accuracy after stimulus onset. Distribution reflects 250 iterations of classifiers. Horizontal bars indicate above-chance classification (p < 0.05, 0.01, and 0.001 for thin, medium, and thick lines, respectively; two-sided permutation test with cluster mass correction for multiple comparisons). **e**, As in Fig. 2f, but showing the accuracy of classifiers trained to decode colour category from LPFC neural activity during the C1 task (orange) and S1 task (green) and then tested on the other task (S1 and C1, respectively). Classifiers were trained only on correct trials. Lines and shading show mean ± s.e.m. classification accuracy after stimulus onset. Distribution reflects 250 iterations of classifiers. Horizontal bars indicate above-chance classification (p < 0.05, 0.01, and 0.001 for thin, medium, and thick lines, respectively; two-sided permutation test with cluster mass correction for multiple comparisons).

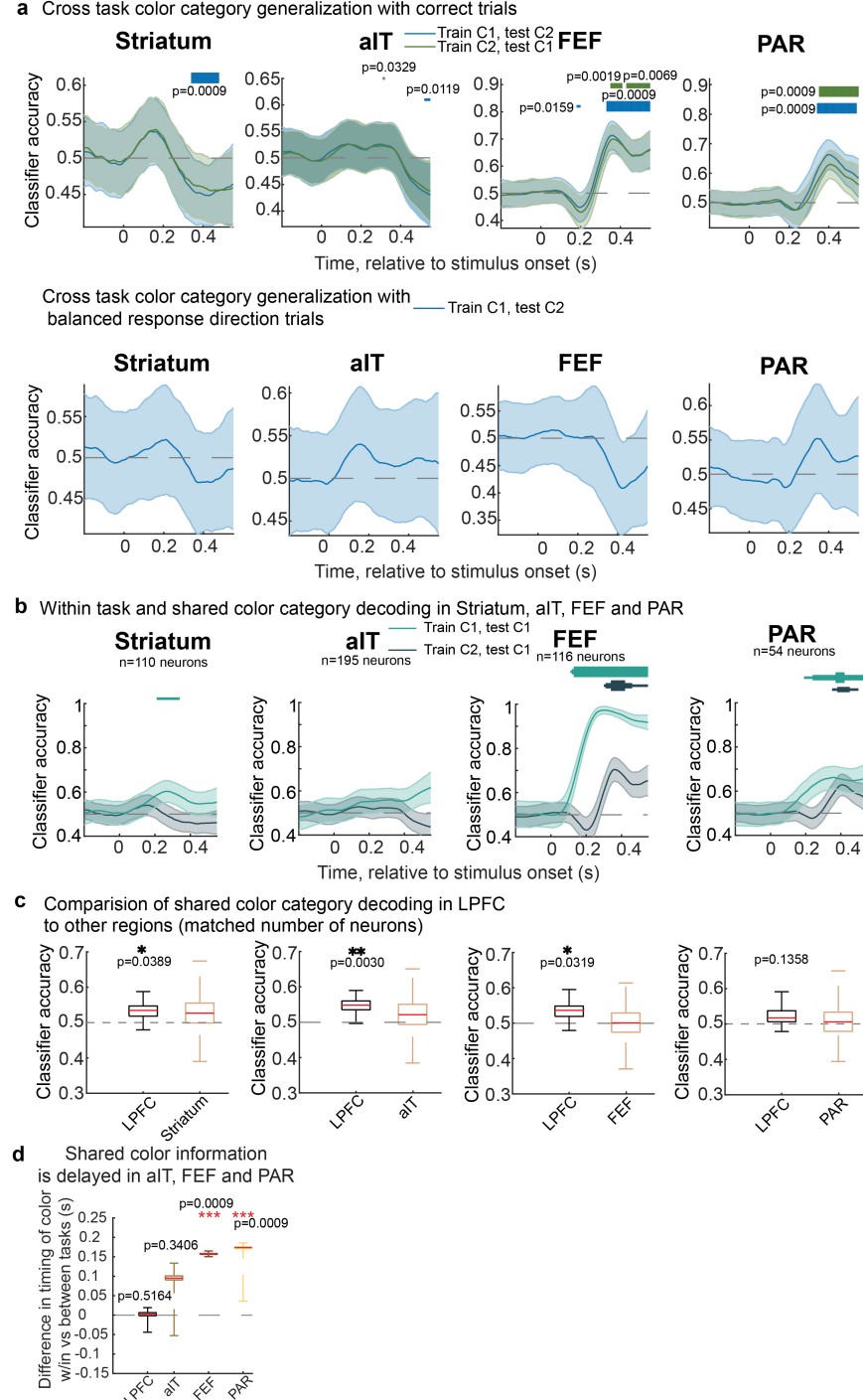

**Extended Data Fig. 6 | Colour category was shared across tasks in a subset of regions. a**, top, Follows Fig. 2f. Time course of accuracy of classifier trained to decode the colour category during the C1 task (blue) and C2 task (green), and then tested on the other task (C2 and C1, respectively). Independent classifiers were trained for each brain region: Striatum, aIT, FEF and PAR (from left to right, across columns). Classifiers were trained on correct trials alone. Lines and shading show mean ± s.e.m. classification accuracy after stimulus onset. Distribution reflects 250 iterations of classifiers. Horizontal bars indicate above-chance classification (p < 0.05, 0.01, and 0.001 for thin, medium, and thick lines, respectively; two-sided permutation test with cluster mass correction for multiple comparisons). bottom, Same as top, but using balanced response direction trials in each colour category. **b**, Time course of accuracy of classifier trained to decode colour category within C1 task and across C1 and C2 tasks. Classifiers were trained separately for each brain region (Striatum, aIT, FEF, and PAR in each column, moving left to right). Lines and shading show mean ± s.e.m. classification accuracy after stimulus onset. Distribution reflects 1000 iterations

of classifiers. Horizontal bars along top indicate above-chance classification (p < 0.05, 0.01, and 0.001 for thin, medium, and thick lines, respectively; one-tailed t-test with no correction). **c**, Comparison of strength of shared colour category responses in LPFC to other regions. All classifiers were trained to decode colour category in C2 task and then tested on C1 task. LPFC neurons were down sampled to match the number of neurons included for this analysis in each other brain region (110 neurons for Striatum, 195 neurons for aIT, 116 neurons for FEF and 54 neurons for PAR). Accuracy was averaged from 50 ms to 150 ms after stimulus onset. Lines and shading show mean ± s.e.m. (p < 0.05, 0.01 and 0.001 for *, **, ***, respectively; one-sided bootstrap test). **d**, Follows Fig. 2g. Difference in the onset of colour information during C1 task when decoded from classifiers trained on C1 task versus generalized from C2 task (vertical dashed lines panels a and f, respectively. p < 0.05, 0.01 and 0.001 for *, **, ***, respectively; two-sided permutation test. LPFC P = 0.5164, aIT P = 0.3406, FEF P = 0.0009, PAR P = 0.0009). Boxes represent interquartile range (25th–75th percentile), and the red horizontal line inside each box is the median.

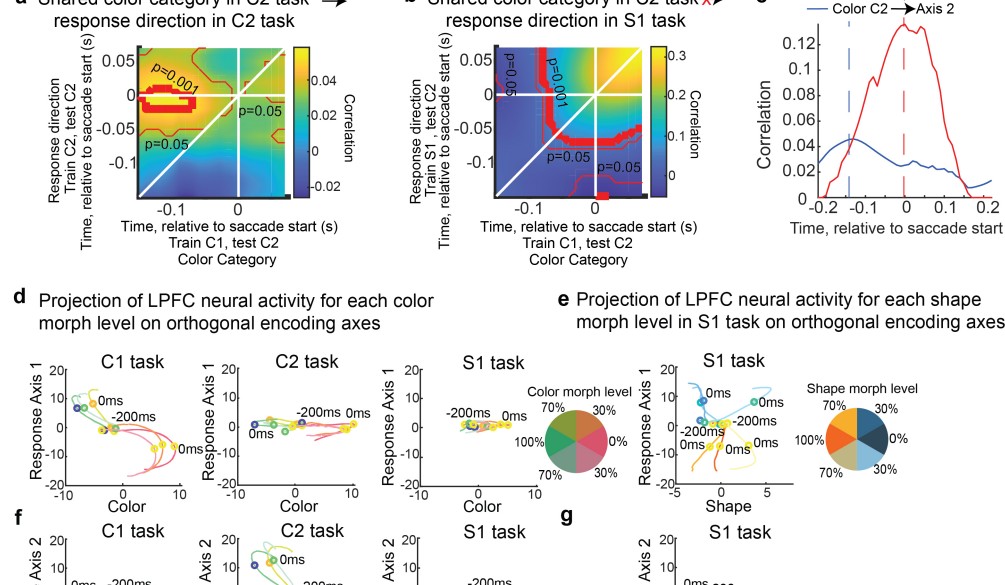

**a** Shared color category in C2 task → response direction in C2 task

**b** Shared color category in C2 task ⟶✗⟶ response direction in S1 task

**c**

**d** Projection of LPFC neural activity for each color morph level on orthogonal encoding axes

**e** Projection of LPFC neural activity for each shape morph level in S1 task on orthogonal encoding axes

**f**

**g**

**Extended Data Fig. 7 | Shared colour category in C2 task was transformed into response on Axis 2 and not Axis 1.** As in Fig. 3c, but shows cross-temporal correlation across trials of shared colour category encoding (trained on C1 task, tested on C2 task) and response direction encoding on Axis 2 (trained on C2 task, tested on C2 task). Thin and thick red lines indicate p ≤ 0.05 and p ≤ 0.001, respectively, uncorrected two-sided t-test. **b**, As in Fig. 3d, but correlating shared colour category encoding (trained on C1 task, tested on C2 task) and response direction encoding on Axis 1 (trained on S1 task, tested on C2 task). Thin and thick red lines indicate p ≤ 0.05 and p ≤ 0.001, respectively, uncorrected two-sided t-test. **c**, As in Fig. 3e, but average cross-temporal correlation along anti-diagonal axis in panel a (shared colour encoding in C2 task predicts response direction in C2 task, blue line) and panel b (shared colour encoding in C2 task predicts response direction in S1 task, red line). Thin and thick red lines in panels b, c

indicate p ≤ 0.05 and p ≤ 0.001, respectively, two-sided uncorrected t-test. **d**, The time course of LPFC neural activity projected on axis encoding colour category and axis encoding response on Axis 1 for C1 (left), C2 (middle) and S1 (right) tasks. Line colours match the actual colour of the stimuli on the colour wheel. Neural activity is aligned to saccade onset time. There is no movement along the axis encoding the response on Axis 1 for S1 because there are equal number of stimuli for each colour morph level that are projected in either direction in Axis 1. **e**, As in panel d, but for LPFC neural activity during S1 task, projected on axis encoding shape and axis encoding response on Axis 1 **f**, As in panel d, but for LPFC neural activity projected on axis encoding colour and axis encoding response on Axis 2. **g**, As in panel e, but for LPFC neural activity in task S1 projected on axis encoding shape and axis encoding response on Axis 2.

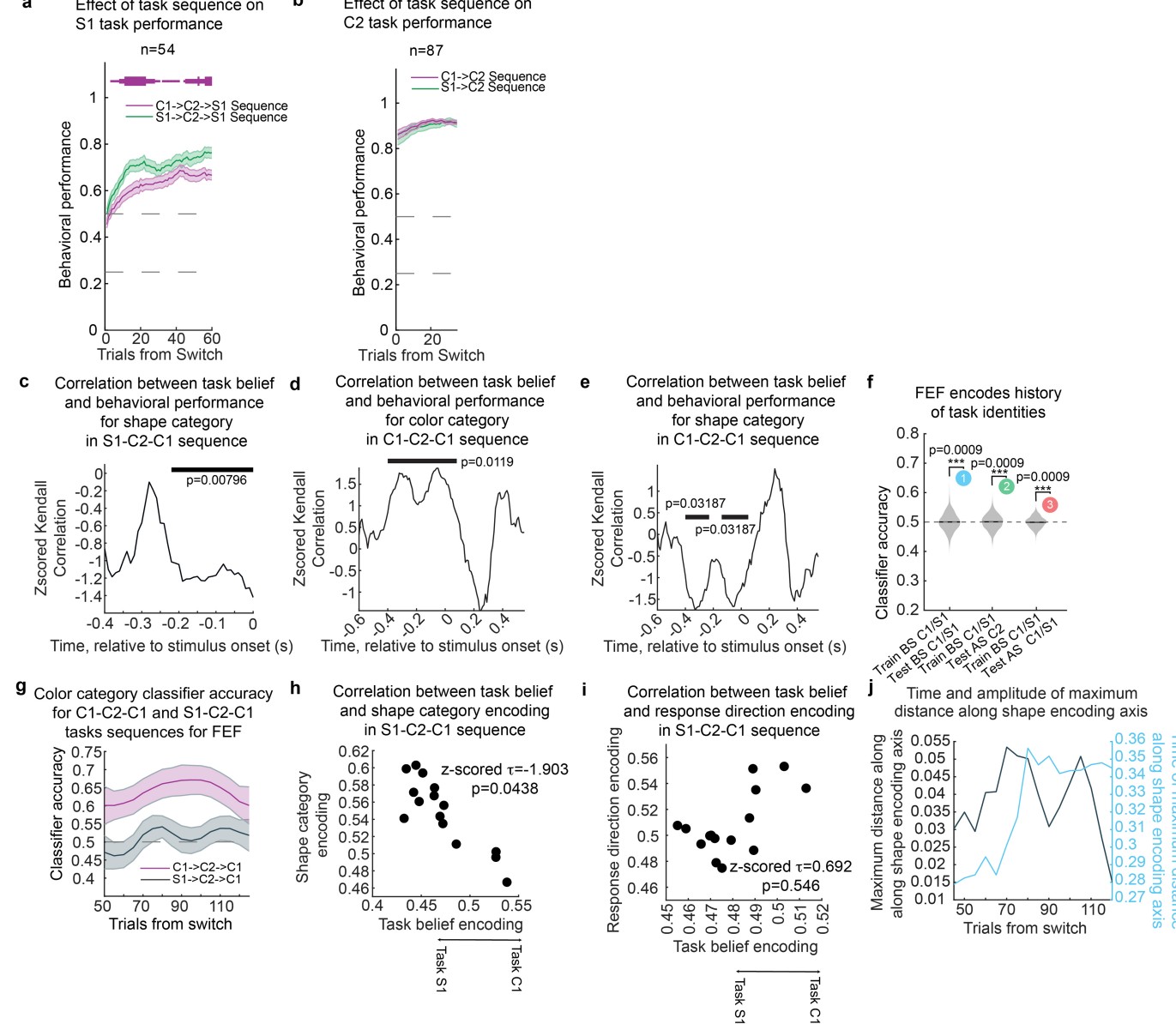

**Extended Data Fig. 8** | See next page for caption.

**Extended Data Fig. 8 | Task sequence affected behavioural performance in S1 and C1 tasks but not in C2 task. a**, Follows Fig. 4a. Comparison of average behavioural performance of both monkeys on S1 task when at the end of a sequence of S1-C2-S1 tasks or C1-C2-S1 tasks. Horizontal bars indicate significant difference between performance in two sequences, p < 0.05, 0.01, and 0.001 for thin, medium, and thick lines, respectively, one-sided chi-squared test, uncorrected for multiple comparisons across trials. Number of included blocks for S1-C2-S1/ C1-C2-S1 sequences were 40/54, respectively. Error bars show mean ± s.e.m. **b**, As in panel a, but comparing average behavioural performance of both monkeys for S1-C2 and C1-C2 task sequences. Number of included blocks for S1-C2/C1-C2 sequences were 82/87, respectively. Lines and shadings show mean ± s.e.m. **c-e**, During task discovery, task belief encoding was inversely correlated with behavioural performance for shape category and positively correlated with behavioural performance for colour category. **c**, Follows Fig. 4e. Correlation between the neurally-estimated task belief (based on task classifier, see Methods) and shape category behavioural performance in C1 task for S1-C2-C1 task sequences. Negative correlation reflects an increase in the behavioural performance in C1 task (thus a decrease in shape performance) when task belief is closer to the C1 task. p < 0.05, 0.01, and 0.001 for thin, medium, and thick lines. Z-scored Kendall's tau with permutation test and cluster mass correction for multiple comparisons across time. **d**, As in panel **c**, but for correlation between the neurally-estimated task belief and colour category behavioural performance in C1 task for C1-C2-C1 task sequences **e**, As in panel **c**, but for correlation between the neurally-estimated task belief and shape category behavioural performance in C1 task for C1-C2-C1 task sequences. **f**, Follows Fig. 4b. Classifier accuracy for decoding the identity of C1 and S1 tasks in FEF using last 50 trials of the C1 and S1 tasks (left), trials of C2 task (middle), and 50 trials after switch of C1 and S1 tasks (right). Numbers inside coloured circles denote corresponding classification task in Fig. 4b. Coloured circles denoted observed values and violins show shuffle distribution. p < 0.05, 0.01, and 0.001 for *, **, and ***, respectively, two-sided permutation test. Similar to LPFC, information about the previous task was present in FEF, both throughout the C2 block and the beginning of the next block (60 neurons, Train before switch C1 and S1, test before switch C1 and S1: 64%, p = 0.0009; Train before switch C1 and S1, test after switch C2: 60%, p = 0.0009; Train before switch C1 and S1, test after switch C1 and S1: 54%, p = 0.0009). The effect was trending in aIT (not shown; 98 neurons, Train before switch C1 and S1, test before switch C1 and S1: 56%, p = 0.087; Train before switch C1 and S1, test after switch C2: 54%, p = 0.032; Train before switch C1 and S1, test after switch C1 and S1: 56%, p = 0.002). There was no significant information about the previous task in PAR or Striatum, although this likely reflects a limited number of neurons recorded in each region. **g**, Comparison of progression of colour category classifier accuracy in C1-C2-C1 task sequence (purple) and S1-C2-C1 task sequence (black) for FEF. Average accuracy was computed using classifier performance in 200 ms to 400 ms after stimulus onset period. Lines and shading show mean ± s.e.m. classification accuracy after stimulus onset. Distribution reflects 250 iterations of classifiers. Δ(average accuracy S1-C2-C1 and C1-C2-C1 sequences)=12.7%, p = 0.004, two-sided uncorrected bootstrap test. On C1-C2-C1 blocks, behavioural performance starts high and only improves slightly during the block (68% on trial 15 to 78% on trial 75 in C1-C2-C1 blocks, compared to 58% on trial 15 to 76% on trial 75 in S1-C2-C1 blocks). Given this, there may not need to be a change in the strength of colour representation during the block. Indeed, the competition from the shape category is low throughout C1-C2-C1 blocks, reflected in the fact that shape information is at or below chance levels (Fig. 4j). In other words, performance may already be 'optimized' and therefore a change may not be needed. If true, one might expect LPFC to become less engaged as the task becomes more certain. Previous work has suggested that LPFC is most strongly engaged when cognitive control is needed – that is, when the task is uncertain[69]. Therefore, the decrease in task representations could reflect the engagement of other neural circuits, such as FEF, as shown here, to represent the colour category during the C1-C2-C1 sequence in comparison to the S1-C2-C1 sequence. **h**, Follows Fig. 4h, correlation between neurally estimated task belief encoding and shape category encoding performance in C1 task for S1-C2-C1 task sequences. Two-sided uncorrected permutation test. **i**, Follows Fig. 4k. Correlation between neurally-estimated task belief and decoded response direction during the C1 task for S1-C2-C1 task sequences. There was a weak correlation between stronger belief encoding and an increase in response direction encoding, but it was not significant. Two-sided uncorrected permutation test. **j**, Follows Fig. 5b. Both the timing (blue) and amplitude (black) of the maximum distance along the shape encoding axis changed over trials as the animals discovered the task. The maximum distance was estimated by fitting a Gaussian CDF function to classifier accuracy in each trial window.

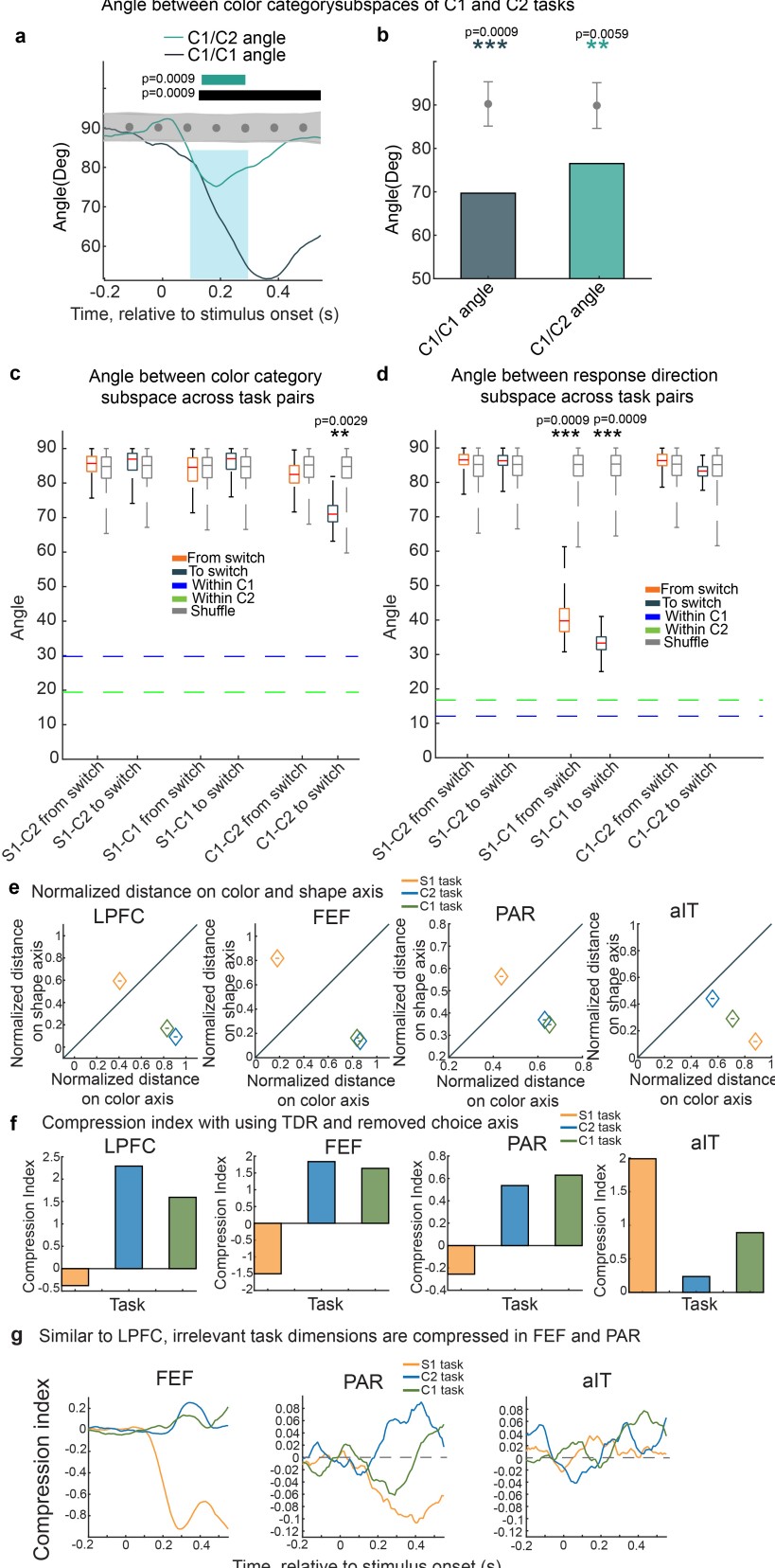

**Extended Data Fig. 9** | See next page for caption.

**Extended Data Fig. 9 | Colour classifiers were aligned in the C1 and C2 tasks.**
**a**, Angle between classifier hyperplanes trained on C1 and C2 tasks (blue) and between classifier hyperplanes trained on split halves of the C1 task (black). Classifiers were trained on n = 24 randomly sampled trials for 250 iterations. Grey shaded region shows mean ± s.e.m. of angles for randomly permuted data. Horizontal bars along top indicate significant angles with respect to chance (two-tailed permutation test, cluster corrected for multiple comparisons). **b**, As in a, but average angle during sensory response period (100 ms - 300 ms after stimulus onset, blue shaded region in panel a). Error bars show mean ± s.e.m. of angles for randomly permuted data. *, **, and *** indicate p < 0.05, 0.01, and 0.001, respectively, one-sided permutation test. **c**, Angle between the axis encoding the colour category in pairs of tasks. Axes were estimated using targeted dimensionality reduction (TDR) using n = 480 LPFC neurons (see Methods) at the beginning of the block ('from switch', orange) and end of the block ('to switch', black). The angle between the colour representation during the C1 task and C2 task was not significantly different from chance at the beginning of the block ($\theta$(C1-C2 after switch)=82.27°, p = 0.234) but was significantly lower than chance at the end of the block ($\theta$(C1-C2 before switch)=71.23°, p = 0.0029). This is consistent with the classifier results in Fig. 4f and suggests the C1 colour representation became aligned with the C2 colour representation as the animals discovered the C1 task. In contrast, the angle between the colour representation in S1 and the C1 or C2 tasks was not significantly different from chance at any point ($\theta$(S1-C1 after switch)=83.75°, p = 0.101, $\theta$(S1-C1 before switch)=86.067°, p = 0.588, $\theta$(S1-C2 after switch)=85.34°, p = 0.358, $\theta$(S1-C2 before switch)= 86.106°, p = 0.780). This suggests colour representations are less shared between the S1 task and C1 or C2 tasks, although this may reflect the reduced colour response during S1. Dashed lines show noise floor estimated by computing the angle between response direction axes of repetitions of C1 (blue) and C2 (green) tasks using all trials during the block ($\theta$(within C1) = 29.78°, p = 0.0009, $\theta$(within C2) = 19.43°, p = 0.0009). Box and whisker plots show distribution of 200 resamples of 75% trials used to fit the GLM model in TDR. Grey plots show null distribution of 1000 random permutations of 75% of trials used to fit the GLM model. All boxes represent interquartile range, red and grey horizontal lines indicate median, whiskers indicate full extent of data. *, **, and ***, indicate p < 0.05, 0.01, and 0.001, respectively, all one-sided permutation tests. **d**, As in panel c, but showing the angle between the axis encoding the response location in pairs of tasks. Consistent with a shared response representation between the C1 and S1 tasks, the angle between the response axes was significantly lower than chance both at the beginning and end of the block of the task ($\theta$ (S1-C1 after switch) =40.92°, p = 0.0009; $\theta$(S1-C1 before switch) =33.35°, p = 0.0009). In contrast, consistent with orthogonal response representations in the C1 or

S1 tasks and C2 task the angle between the C2 response axis and the C1 or S1 response axes were not significantly different from chance ($\theta$(S1-C2 after switch)=86.44°, p = 0.612, $\theta$(S1-C2 before switch)=86.023°, p = 0.715, $\theta$(C1-C2 after switch)=86.28°, p = 0.594, $\theta$(C1-C2 before switch)=83.1203°, p = 0.363). Dashed lines show noise floor estimated by computing the angle between response direction axes of repetitions of C1 (blue) and C2 (green) tasks using all trials during the block ($\theta$(within C1) = 12.05°, p = 0.0009, $\theta$(within C2) = 16.76°, p = 0.0009). Box and whisker plots show distribution of 200 resamples of 75% trials used to fit the GLM model. Grey box and whisker plots show null distribution of 1000 random permutations of 75% of trials used to fit the GLM model. All boxes represent interquartile range, red and grey horizontal lines indicate median, whiskers indicate full extent of data. *, **, and ***, indicate p < 0.05, 0.01, and 0.001, respectively, all one-sided permutation tests **e-g**, Colour and shape representations were scaled by task belief in all regions, except aIT. **e**, Normalized distance on colour and shape axes for LPFC, FEF, PAR and aIT (from left to right). Axes were estimated using TDR to remove motor responses[21] (see Methods). Figure axes show distance of neural response representing different stimuli, projected onto the shape (y-axis) and colour axes (x-axis). Neural activity from 100ms–400ms after stimulus onset. Points above [below] the diagonal indicate a greater distance along the shape [colour] axis compared to the colour [shape] axis (i.e., greater encoding of shape [colour]). As in Fig. 5c, colour representations in LPFC, FEF and PAR were more separated during C1 and C2 tasks while shape representations were more separated during S1 task. Encoding in aIT was not affected by task (colour representation was always greater and did not scale by task in the predicted way). **f**, Compression Index (CPI) of stimulus representations in LPFC, FEF, PAR and aIT, for all three tasks. CPI was taken as the log of the ratio of the separability of stimuli when projected onto the colour and shape axes from TDR (averaged in the period 100 ms to 400 ms after stimulus onset). CPI greater than [less than] zero reflects stronger relative encoding of colour [shape]. **g**, Follows Fig. 5c. CPI of neural responses in FEF, aIT and PAR over time, for all three tasks. Here, CPI is taken as the log of the ratio of the separability of stimuli in colour and shape subspaces using classifier encoding axis. CPI in FEF is similar to LPFC, with greater separation of colour representations during C1 and C2 tasks and greater separation of shape representations during S1 task. CPI was reduced in PAR overall but showed a similar pattern with greater separation of colour in the C2 task and shape in the S1 task (C1 was less clear). CPI in aIT was close to zero for all tasks. This is consistent with previous findings[23] that task-dependent compression of features is limited in sensory regions. Striatum did not show significant colour or shape information and so CPI is not reliable.

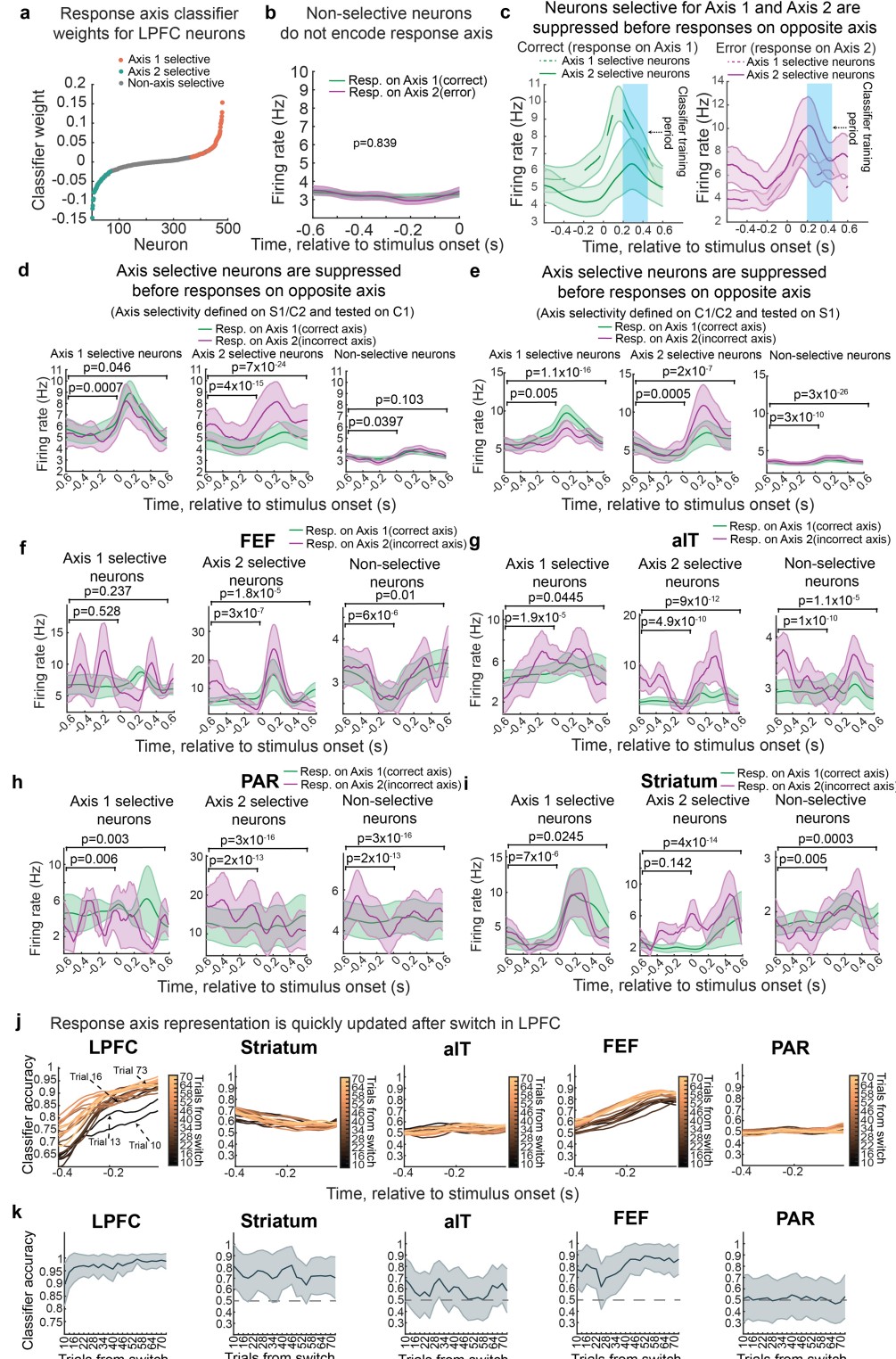

**Extended Data Fig. 10** | See next page for caption.

**Extended Data Fig. 10 | Neurons encoding response on Axis 1 and Axis 2 are suppressed when irrelevant to the task. a**, Classifier weights for classifying axis of response for Fig. 5g. Classifier was trained to decode response axis using C1 and C2 task trials. Neurons with significant weights for Axis 1 and Axis 2 are denoted with green and orange dots, respectively. Non-selective neurons are denoted with grey dots. **b**, Follows Fig. 5g, time course of average firing rate for non-selective neurons in LPFC. Inset shows p-value of difference in firing rate when animal responded on Axis 1 compared to responding on Axis 2 (two-sided paired t-test). Lines and shading show mean ± s.e.m. **c**, Follows Fig. 5g, time course of average firing rate of neurons when the animal responded on Axis 1 (dashed) or Axis 2 (solid). Neurons were grouped by their preferred response on Axis 1 (left) or Axis 2 (right). Shaded blue box indicates the time period used to define the neurons preferred response axis (200ms–450ms after stimulus onset). Lines and shading show mean ± s.e.m, over time. **d**, Follows Fig. 5g. Average firing rate of LPFC neurons during trials when the animal responded on Axis 1 (green) and Axis 2 (pink). Three panels show responses of three categories of neurons: those that responded most strongly on Axis 1 (left), Axis 2 (middle), and were not selective (right) in LPFC. To characterize axis selectivity, classifier was trained to decode response axis using trials from S1 and C2 trials and then firing rate was estimated on withheld C1 trials. Inset shows p-value for of difference between firing rate in response on Axis 1 and response on Axis 2 trials in time periods −600ms-0ms (lower bracket) and −600ms-600ms (upper bracket) after stimulus onset. Note, tests are performed pairwise within each neuron (two-sided paired t-test), whereas lines and shading show mean response across neurons, ± s.e.m, over time. **e**, As in panel d, but axis selectivity was defined by training a classifier to decode response axis using trials from C1 and C2 trials and then firing rate was estimated on withheld S1 trials. Inset shows p-value for of difference between firing rate in response on Axis 1 and response on Axis 2 trials in time periods −600ms-0ms (lower bracket) and −600ms-600ms (upper bracket) after stimulus onset (two-sided paired t-test). Lines and shading show mean ± s.e.m, over time. **f-i**, As in Fig. 5g, but shown for **f**, FEF; **g**, aIT; **h**, PAR and **i**, Striatum. Similar to LPFC, Axis 1 and Axis 2 selective neurons in FEF, aIT and PAR were suppressed during response on opposite axis. Lines and shading show mean ± s.e.m. firing rate across individual neurons for each region. Inset shows p-value for of difference between firing rate in response on Axis 1 and response on Axis 2 trials in time periods −600ms-0ms (lower bracket) and −600ms-600ms (upper bracket) after stimulus onset (two-sided paired t-test). **j**, As in Fig. 5h, but shows classifier accuracy in decoding the response axis for all brain regions. Darker to lighter colour show progression in trial blocks of 10 shifted by 3 trials from switch trial in C1 task. Lines show mean classification accuracy after stimulus onset for 250 iterations of classifiers. Arrows indicate the time course of example trials. **k**, Classifier accuracy in decoding the response axis for all brain regions. Decoding using neural activity in −400ms to 0 ms before stimulus onset period. Lines and shading show mean ± s.e.m. classification accuracy across 250 iterations of classifiers.

*Double-anonymous peer review submissions: write DAPR and your manuscript number here instead of author names.*

# Reporting Summary

## Statistics

For all statistical analyses, confirm that the following items are present in the figure legend, table legend, main text, or Methods section.

| n/a | Confirmed | |
|---|---|---|
| ☐ | ☒ | The exact sample size (*n*) for each experimental group/condition, given as a discrete number and unit of measurement |
| ☐ | ☒ | A statement on whether measurements were taken from distinct samples or whether the same sample was measured repeatedly |
| ☐ | ☒ | The statistical test(s) used AND whether they are one- or two-sided<br>*Only common tests should be described solely by name; describe more complex techniques in the Methods section.* |
| ☐ | ☒ | A description of all covariates tested |
| ☐ | ☒ | A description of any assumptions or corrections, such as tests of normality and adjustment for multiple comparisons |
| ☐ | ☒ | A full description of the statistical parameters including central tendency (e.g. means) or other basic estimates (e.g. regression coefficient) AND variation (e.g. standard deviation) or associated estimates of uncertainty (e.g. confidence intervals) |
| ☐ | ☒ | For null hypothesis testing, the test statistic (e.g. *F*, *t*, *r*) with confidence intervals, effect sizes, degrees of freedom and *P* value noted<br>*Give P values as exact values whenever suitable.* |
| ☒ | ☐ | For Bayesian analysis, information on the choice of priors and Markov chain Monte Carlo settings |
| ☒ | ☐ | For hierarchical and complex designs, identification of the appropriate level for tests and full reporting of outcomes |
| ☐ | ☒ | Estimates of effect sizes (e.g. Cohen's *d*, Pearson's *r*), indicating how they were calculated |

*Our web collection on statistics for biologists contains articles on many of the points above.*

## Software and code

Policy information about availability of computer code

| Data collection | We presented stimuli and recorded behavioral responses using Psychtoolbox (version 3) and  MATLAB (version 2015a). The behavioral training code can be provided upon request. Electrophysiological data was acquired using  Plexon Omniplex recording system (Plexon Inc., Dallas, TX.)  Eye tracking data was gathered with an EyeLink 1000 (SR Research, software version 5.09). Neuron waveforms were sorted using Plexon's Offline Sorter (version 4). |
|---|---|
| Data analysis | The data underwent analysis utilizing both built-in functions and custom code developed in MATLAB (version 2024b & version 2021b). Built-in functions include fitclinear for logistic regression, lassoglm for fitting GLM models, pca for dimensionality reduction, corr for Pearson's correlation, and fminsearch for fitting psychometric curves. These are highlighted in the corresponding methods section. To quantify the statistical significance of trends, we used Trend-Free PreWhitening (TFPW) as implemented in MK_tempAggr.m function (mannkendall Version 1.1.0 ,https://mannkendall.github.io/Matlab) and referenced in Coen et al. 2020. The stimuli were rendered as three-dimensional models using POV-Ray (version 3.7). To compute chi-squared test we used chi2test.m function implemented in MATLAB and referenced in Axensten P, chi2test  (https://www.mathworks.com/matlabcentral/fileexchange/16177-chi2test), MATLAB Central File Exchange (2007). Procedures for non-standard statistical procedures are detailed in the methods. Code for custom functions are either referenced in the manuscript with a pointer to the original source or are included in an online public repository (https://github.com/buschman-lab/CompositionalTasks) and on Zenodo (https://doi.org/10.5281/zenodo.17274345). |

For manuscripts utilizing custom algorithms or software that are central to the research but not yet described in published literature, software must be made available to editors and reviewers. We strongly encourage code deposition in a community repository (e.g. GitHub). See the Nature Portfolio guidelines for submitting code & software for further information.

## Data

Policy information about availability of data

 All manuscripts must include a data availability statement. This statement should provide the following information, where applicable:
- Accession codes, unique identifiers, or web links for publicly available datasets
- A description of any restrictions on data availability
- For clinical datasets or third party data, please ensure that the statement adheres to our policy

> Processed data have been deposited on the FigShare archive and can be accessed at https://figshare.com/articles/dataset/ Data_for_Tafazoli_et_al_2025_Building_Compositional_Tasks_with_Shared_Neural_Subspaces/30276238

## Research involving human participants, their data, or biological material

Policy information about studies with human participants or human data. See also policy information about sex, gender (identity/presentation), and sexual orientation and race, ethnicity and racism.

| | |
|---|---|
| Reporting on sex and gender | N/A |
| Reporting on race, ethnicity, or other socially relevant groupings | N/A |
| Population characteristics | N/A |
| Recruitment | N/A |
| Ethics oversight | N/A |

Note that full information on the approval of the study protocol must also be provided in the manuscript.

# Field-specific reporting

Please select the one below that is the best fit for your research. If you are not sure, read the appropriate sections before making your selection.

☒ Life sciences    ☐ Behavioural & social sciences    ☐ Ecological, evolutionary & environmental sciences

For a reference copy of the document with all sections, see nature.com/documents/nr-reporting-summary-flat.pdf

# Life sciences study design

All studies must disclose on these points even when the disclosure is negative.

| | |
|---|---|
| Sample size | As detailed in the manuscript, a total of 1,081 neurons were recorded from 5 brain regions across 2 animal subjects. The number of subjects (2) and the number of neurons recorded per region (64-480) follows previous work using similar approaches (e.g., Panichello and Buschman et al, 2021, Buschman and Miller, 2007; Mante et al, 2013; Siegel et al, 2015). |
| Data exclusions | To ensure statistical power, neurons were omitted from specific analyses if they did not meet a predetermined number of trials in any condition of interest. This exclusion criterion was predetermined and is delineated for each analysis in the methods section. Additionally, units that were only partially detected during recording sessions were excluded. All analyses presented in this manuscript were conducted on individual neurons, isolated based on their waveform. |
| Replication | Independent experiments were performed in 2 animals, with 1,081 neurons recorded across 34 days. All data was included (except for exclusions noted above). There were no failed replication attempts (i.e., no animals failed to learn the task). Neural recordings were excluded only if the animals did not perform predetermined number of blocks during a recording session. |
| Randomization | Each animal underwent exposure to every task manipulation. Task orders and trial conditions were randomized across trials within each session. Neurons were recorded without bias. Electrodes were positioned to optimize the signal-to-noise ratio of the electrophysiological signal without consideration of neural type or selectivity. |
| Blinding | Since all animals were allocated to a single experimental group, blinding was neither necessary nor feasible during behavioral training. Experimenters were blinded to experimental conditions while recording neurons and sorting waveforms into individual neurons. |

# Reporting for specific materials, systems and methods

We require information from authors about some types of materials, experimental systems and methods used in many studies. Here, indicate whether each material, system or method listed is relevant to your study. If you are not sure if a list item applies to your research, read the appropriate section before selecting a response.

## Materials & experimental systems

| n/a | Involved in the study |
|---|---|
| ☒ ☐ | Antibodies |
| ☒ ☐ | Eukaryotic cell lines |
| ☒ ☐ | Palaeontology and archaeology |
| ☐ ☒ | Animals and other organisms |
| ☒ ☐ | Clinical data |
| ☒ ☐ | Dual use research of concern |
| ☒ ☐ | Plants |

## Methods

| n/a | Involved in the study |
|---|---|
| ☒ ☐ | ChIP-seq |
| ☒ ☐ | Flow cytometry |
| ☒ ☐ | MRI-based neuroimaging |

## Animals and other research organisms

Policy information about studies involving animals; ARRIVE guidelines recommended for reporting animal research, and Sex and Gender in Research

| | |
|---|---|
| Laboratory animals | Two adult male rhesus macaques (Macaca mulatta) participated in the experiment. Monkeys Si and Ch were between 8 and 11 years old and weighed approximately 12.7 and 10.7 kg, respectively. |
| Wild animals | No wild animals were used in this study. |
| Reporting on sex | Both animals were male. |
| Field-collected samples | No field-collected samples were used in this study. |
| Ethics oversight | All experimental procedures were approved by the Princeton University Institutional Animal Care and Use Committee (protocol #3055) and were in accordance with the policies and procedures of the National Institutes of Health. |

Note that full information on the approval of the study protocol must also be provided in the manuscript.

## Plants

| | |
|---|---|
| Seed stocks | N/A |
| Novel plant genotypes | N/A |
| Authentication | N/A |

