## [Peer Review File · Nature]

Building compositional tasks with shared neural subspaces

Corresponding Author: Professor Timothy Buschman

Version 0:

Reviewer comments:

Referee #1

(Remarks to the Author)

Tafazouli et al trained monkeys to discriminate visual stimuli that vary continuously in shape and colour, whilst recording neural signals from multiple brain regions. They trained monkeys to map shape onto response axis [S1], map colour onto response axis 1 [C1] and map colour onto response axis 2 [C2]. Their question concerns the nature of the neural representation that allows monkeys to jointly solve these tasks, and in particular the extent to which representations can be “re-used” to complete new tasks.

This is a high-quality paper that describes important findings. It is beautifully written and presented. The question is timely and will, I think, have significant impact. The authors have also generated an extremely valuable dataset with multi-region recordings from an innovative task. So there is a lot to like here.

Here are some questions that occurred to me as I was reading:

1/ I found the framing of the paper a little odd. The paper doesn't really seem to be about composition, at least not in the sense that we typically use the term in cognitive science. Composition is assembly of new tasks from existing building blocks. So for example, if I can do C2 and S1 then I should be able to “compose” task C1 from the combination of learning about stimulus axis C and response axis 1. But that's not what is happening here. In fact, in task C1, performance is overall hampered by its overlap with task S1, because half of the stimuli create response conflict (whereas they do not in C2 due to the unique response axis). To my mind, a demonstration of composition would be that the monkeys (for example) learn C1 more readily after exposure to S1/C2 compared to (say) some matched control task. Or that they learn faster on the 4th “missing” combination S2 (presumably the authors tried this?).

In support of their claims about composition, the authors make quite a bit of hay about “representational subspaces being reused over tasks” (line 104 onwards). What they are referring to here is the fact that there neurons that care about colour in C1 also care about colour in C2 and those that care about response in C1 also care about response in S1. This is just like saying: some neurons code for sensory variables and others for motor variables. It doesn't really tell us much about composition, I don't think. The sequential engagement of these subspace just means that inputs are converted into outputs. Reassuring, but not spectacularly surprising.

The analyses of task sequences (line 161 onwards) are cool. The monkeys seem to have a prior on task repetition that is relatively enduring. But I wasn't clear about the apparently contradiction between behaviour (Fig. 4a) and classification performance (Fig. 4g). In an C2-C1 sequence, behaviour improves if the previous task was C1 (rather than S1) but the opposite is true for colour decoding; and yet these effects are positively correlated (Fig 4e). The authors sort of gesture to the fact that perhaps the monkey “re-learning” C1 is driving the neural effect, but then why the converse behavioural advantage? This seems to warrant a bit more discussion.

Although superficially symmetric, there is an obvious difference between C2 and the other tasks, in that it the monkey knows unambiguously from the first feedback that this is the rule. I would have been interested in knowing a bit more about how the representation of C2 differs from C1 and S1; can we understand more about how monkeys ensure that colour representations in the C2 task do not inadvertently prompt responses on axis 1? The behavioural advantage for C2 does not

seem to be solely explained by the faster relearning following a switch (because it persists deep into each block (1f) and is actually expressed as heightened psychometric slope and not (say) an increase in bias or lapses (Fig 1e). Given the shared coding between C2 and C1 at the input level, there must be a process that protects C2 in a way that allows its (quite substantially) superior performance. This question isn't really (as far as I could see) addressed by the analysis of compression described below. Do the monkeys monitor for C1 / S1 and treat C2 as an "exception" that is qualitatively distinct in some way?

Perhaps the most impactful results from this study are potentially those relating to compressive coding of irrelevant dimensions. Since 2013, a canonical story has emerged that in this type of task-switching setting, there is no compression of irrelevant alternatives, at least in FEF. This question is hard to pose because in most prefrontal regions most neurons care about the choice axis (so in the Mante / Newsome Sussillo paper the claim about lack of compression concerns the subspace orthogonal to choice; in other work, e.g. that from Miller / Siegel, the choice axis dominates). Here, we have potential evidence for powerful compression; this could be a significant finding, and opens up a new chapter in this debate.

The results in Figure 5 look compelling but I have two questions. Firstly, I'm presuming that the results described are from LPFC exclusively; can the authors comment on what they found in FEF, for direct comparison with Mante 2013 (or in other regions)? Secondly, the monkeys make different responses along response axis 1 in tasks C1 and S1; can the authors be sure that the results are not secondary to neurons simply coding for the choice axis? Given that the response targets are fixed in each block, for example in C1 different colours will map onto different responses whereas different shapes will be orthogonal to this axis; the converse is true in S1. Do the results persist when the choice axis is removed, as in Mante (2013)?

Minor points

Figure 5c - the line colour and legend do not seem to match
Figure 4e – that linear trend looks a bit uncomfortable on that plot

(Remarks on code availability)

Referee #2

(Remarks to the Author)

Tafazoli et al present findings on a "compositional" task in monkeys, in which LPFC activity in particular uses overlapping subspaces for shared stimuli (color with two different saccade axis outputs) or shared outputs (color or shape for the same saccade direction). This is a highly challenging task, and the authors carefully analyze the behavior and use a variety of classifier generalization procedures to demonstrate the central finding. It is technically sound. There is some new science here: the reuse of dimensions is not exactly known in LPFC (though something like it is known elsewhere in frontal cortex), and maybe more novel is that LPFC compresses the irrelevant stimulus dimension dependent on task belief. This is all reasonably interesting, but doesn't have the scientific depth to it that I would have expected at this journal.

The paper does have some real strengths. This task involves blockwise interleaving of 3 different subtasks, which was apparently very challenging to train, and where good performance was attained. The authors recorded from 5 brain areas, though mostly only LPFC was the focus here. The analysis of the task was creative and in-depth, and leads to a good understanding of what the monkey is doing in this complex task. The application of encoding models gives a general idea of what's present in these areas, and the extensive use of linear decoders makes clear that some dimensions are in fact used across the expected related subtasks. The task-belief-dependence of the stimulus representation suppression is, to my knowledge, novel. The pieces are all executed well, and my technical concerns are minor.

Conceptual limitations

The central problem here is that I don't find these results to be all that surprising, and I don't think many in the field would. As I discuss below, I think one of the two main findings was basically known already, and the other has already been largely superseded with additional results and an extensive theoretical framework.

The first main finding is that the stimulus subspace (really a single decoded dimension) is conserved across tasks sharing a stimulus, and the movement subspace (again, a single decoded dimension) is shared across tasks sharing a response. Both were previously found by Mante 2013 in FEF, and are now known in rat FOF in an analogous task (Pagan et al. 2022 bioRxiv, though not yet peer reviewed). Moreover, this is what you would expect given visual inputs to the structure and the widespread prevalence of movement-related activity throughout cortex. To my knowledge this wasn't established in this exact area, and certainly not in this very nice, novel task. This makes the effort worthwhile. But, it's also not very surprising, so the amount of scientific progress it represents is limited.

The second main finding is that LPFC compresses the irrelevant stimulus information, in a task-belief-dependent fashion. This is a different result than Mante 2013, but has now been described as part of a continuum of possible solutions by Pagan 2022 (bioRxiv). Preprints aren't papers, and I don't mean to imply that the existence of that preprint scoops this paper. Rather, the frustrating thing about the present work is that despite the creativity in applying linear decoders, ultimately it's still just a lot of linear decoders; there isn't much translation into new scientific thinking. Here, the authors are still talking about

one dimension at a time, where there are a long list of ways they could have made the science deeper. There is no examination of the geometry between the various dimensions (e.g., what was the angle between the C1 and C2 color encoding dimensions?); there is no consideration of linear vs. nonlinear encoding; there are no dynamics; there is no link from stimulus-related activity to movement-related activity (as in, e.g., Elsayed 2016); there is no modeling of any kind. Even one or two of these would have made for a much bigger potential scientific impact. But as it is this just isn't a big enough scientific (or technical) advance to be a Nature paper for me. If we didn't have a decade of thinking in this direction the current work might be enough, but the field has advanced in our understanding of task representation beyond the thinking in this paper. It ends up being a lot of work for a modest amount of new science.

Minor comments

What potential influences could the training procedure / task training order have had on the learned representation? This has been shown to matter, and should be discussed.

Figure 4g: the color categorization classifier's dependence on sequence doesn't seem consistent with the task classifier results. The explanation (L200-208) doesn't clearly explain it.

Fig. 5b / L238 – the text says that shape representation was attenuated in the color tasks, but that's not what the figure shows; really the peak is delayed and drops back down instead of being sustained. This is also interesting, but not consistent with the text.

Figure S2: Why is task information so time-dependent? How should we think about its role in how the whole system works?

Why does the correlation over time in Fig. S9b strongly flip sign?

Details

Fig. 2f: what is the big red X? Non-significance? Onset time?

Figure 1i: I think the yellow trace (PAR) is missing.

Typo: Figure 4c,d, "indentify"

(Remarks on code availability)

Referee #3

(Remarks to the Author)

The paper presents a compelling study on the population activity supporting the execution of multiple tasks by non human primates. The three tasks are designed in a way that a pair of tasks share stimulus space, while an other pair of tasks shares the decision space. This task design enables the investigation of whether the shared stimulus features and shared decision space are actually reflected in a shared representational space of the neutral population activity. The study reports recording from multiple brain areas, including higher visual area as well as prefrontal regions. The study involves simple population analysis and both statistically well-grounded single cell analysis and properly motivated population level analyses that relies on simple linear decoding. Critically, the complex behavioral paradigm and the elaborate cognitive processes require accurate control over a vast array of confounds, in which the paper excels. The compelling claim of the paper is that it provides evidence that the lpfc hosts separate subspaces for stimuli and decisions, each of which is reused in tasks. Consequently, the lpfc efficiently manages to route information between these subspaces depending on task requirements. Importantly, the tasks are uncued, which requires the animals to discover the current task based on stimulus /decision /reward contingencies. This setting enables the authors to investigate how subjective belief about task shapes the neural representation. The presented analysis provides support for a representation that is highly consistent with theoretical expectations. The paper is very clearly written, presenting the findings in an approachable way.

The paper is very timely since both machine learning, and computational neuroscience identified continual /lifelong learning as a key challenge for learning systems (Hasdell et al, 2020, TICS). The paper also relates to the meta learning literature in machine learning that has attracted attention in the cognitive science community too (Binz et al, 2023, BBS). The key message of the paper, compositional representation has been the focus of a number of recent human behavioral studies and the neural mechanisms have also been the target of a number of studies (Schwartenbeck et al, 2023, Cell; Lake and Baroni, 2023, Nature). The key contribution of the paper is presenting a clever task design that is very challenging to animals (60 months of training) and making a strong case for compositional representation in the brain. Importantly, while a representation is appealing and seems like a 'natural' solution from a theoretical point of view, it is far from the only solution. Recent studies in deep learning have highlighted that there are distinct 'lazy' and 'rich' regimes when training an artificial neural network on multiple tasks, the rich being characterized by a representation similar to the one presented here, and circumstances of training determine which of these prevail (Chizat et al, 2018; Flesch et al, 2022).

The results of the paper gain strong support by careful analyses that look at the problem from different angles, and these analyses build step-by-step a framework that strengthens the interpretation of the authors. One particularly appealing example of this is the analysis of the dependence of performance after task switch on the sequence of earlier blocks: initial uncertainty of the animals about the relevance of the stimulus feature gradually builds up, concurrently with the loss in reliance on the feature that was relevant in a past block.

Despite the above described appeals of the manuscript, several issues require additional care and I also list a number of suggestions.

1. The paper sets off by stating that they test 'this hypothesis' but it is not clearly stated what the actual hypothesis is. The continual learning literature lists quite an array of architectures (Van de Ven et al, 2022, Nat Mach Intell) that might serve computations necessary for task performance, which could be highlighted as alternatives (including the representations associated with the above quoted lazy and rich learning regimes).
2. The paper extensively analyses LPFC in the context of compositional generalisation, and the discussion mentions that generalisation was strongest in PFC (line 277), but I could not identify a compelling comparison of brain regions. Demonstrating specificity of the presented computations would be very informative to establish unique contributions. The five recording locations provide an extremely rich data but comparison of the contributions is limited in the paper.
3. According to task design, there is correlation between the shape category and response when the shape is relevant. Is it ensured here that this correlation is not confounding the results shown on Fig 2a (and related figures in the supplementary materials, Fig S2)? If only correct trials are used, as the methods states, then this correlation might introduce confounds. If, however, responses are balanced then decision-related confound is not present anymore, but in that case the error rate will be higher. In this case one can assume that the trials used in the analysis come from the beginning of the block when the animal is still uncertain about the task being performed. In this case, however, relevance of the feature is not determined by the animal. This potential confound is most relevant at later times during the trial and might affect e.g. FEF responses more substantially. Also, this might cause a drop in across-task decoding of colour late in the trial (Fig 2f).

Minor:

1. The paper argues that a sequence of S1C2C1 has different dynamics than the sequence C1C2C1. For this the animal needs to maintain C1 during C2. A query if this can be identified with similar analysis techniques as the ones used in the paper during C2 would be instructive.
2. The paper uses the terminology of 'learning' a task (e.g. line 174). This might be confusing since the task had been trained exhaustively. To avoid this confusion the term 'apparent learning' was proposed in the context of contextual learning (Heald et al 2021). It might be useful for the community to settle for an unambiguous vocabulary.
3. The paper states that block changes were triggered by a performance level reaching 75%. As the paper also points out, some of the trials are easier to perform: stimuli with congruent properties (colour and shape indicating same choice). According to the methods these trials cover 20% of trials. As a consequence, performing only these perfectly while performing the rest at change could account for 60% performance level. It would therefore be important to see how the correct trials are distributed between congruent and incongruent stimulus property trials.
4. Task identity is identified in neuronal responses through CPD. As later analysis indicates there might also be suppression of responses to irrelevant stimulus properties. Such suppression can actually account for the task identity information as the modulation is task-specific. This might not be the central topic of the paper, but the reader might be confused by the apparent wealth of information present in a humble neural population. The relationship between task representation and stimulus representation can distinguish sensory and higher cortices (e.g. Hajnal et al (2023) biorxiv). The suppression identified in the paper defines a two-dimensional subspace for task representation (along the shape decoder vector and along the colour decoder vector), and it remains unclear if task representation is different from this suppression-related task-specific modulation.
5. Line 201: 'Shared color representation increased during the C1 task (...)' It would be useful to disentangle the referred enhanced representation of C1 from a different effect: more consistent performance in the task (which adds stimulus-consistent decision-related activity). This comment relates to Major #3 above.
6. Mixed selectivity is a key aspect in lPfc and continual learning paradigms distinguish between solutions relying on disjunct and overlapping population the neuron population. Illuminating this aspect of the population responses would be important.

References:

- Binz M, Dasgupta I, Jagadish AK, Botvinick M, Wang JX, Schulz E. Meta-Learned Models of Cognition. Behavioral and Brain Sciences. Published online 2023:1-38. doi:10.1017/S0140525X23003266
- Chizat, L., Oyallon, E., and Bach, F. (2018). On Lazy Training in Differentiable Programming in Neural Networks. arXiv, 1812.07956 <http://arxiv.org/abs/1812.07956>.
- Flesch, T., Juechems, K., Dumbalska, T., Saxe, A., & Summerfield, C. (2022). Orthogonal representations for robust context-dependent task performance in brains and neural networks. *Neuron*, 110(7), 1258-1270.

Hadsell, R., Rao, D., Rusu, A. A., & Pascanu, R. (2020). Embracing Change: Continual Learning in Deep Neural Networks. *Trends in Cognitive Sciences*, 24(12), 1028–1040. <http://doi.org/10.1016/j.tics.2020.09.004>

Hajnal, M. A., Tran, D., Szabó, Z., Albert, A., Safaryan, K., Einstein, M., ... & Orbán, G. (2023). Shifts in attention drive context-dependent subspace encoding in anterior cingulate cortex during decision making. *bioRxiv*.

Heald, J. B., Lengyel, M., & Wolpert, D. M. (2021). Contextual inference underlies the learning of sensorimotor repertoires. *Nature*, 600(7889), 489-493.

Lake, B. M., & Baroni, M. (2023). Human-like systematic generalization through a meta-learning neural network. *Nature*. <https://doi.org/10.1038/s41586-023-06668-3>

Schwartenbeck, P., Baram, A., Liu, Y., Mark, S., Muller, T., Dolan, R., Botvinick, M., Kurth-Nelson, Z., & Behrens, T. (2023). Generative replay underlies compositional inference in the hippocampal-prefrontal circuit. *Cell*, 186(22), 4885-4897.e14. <https://doi.org/10.1016/j.cell.2023.09.004>

Van de Ven, G. M., Tuytelaars, T., & Tolias, A. S. (2022). Three types of incremental learning. *Nature Machine Intelligence*, 4(12), 1185-1197.

(Remarks on code availability)

Version 1:

Reviewer comments:

Referee #1

(Remarks to the Author)

This is an exceptionally responsive review - the authors really have gone the extra mile, both in relation to my comments and those of other reviewers.

I basically buy most of the authors' replies - including to the more sceptical R2 so I don't have further concerns.

(Remarks on code availability)

Referee #2

(Remarks to the Author)

The authors should be commended for a strongly responsive set of revisions, including quite a few new analyses and an improved new Discussion. The improved contrast with prior literature and deeper exploration of several aspects of their data have successfully convinced this reviewer of the novelty and interest of their findings. I now support publication of this manuscript at Nature. I have only very minor remaining changes to suggest.

In the rebuttals, the authors show that training RNNs with different initial conditions can lead to different solutions. (And presumably the same would be true with different curricula.) As I previously noted, it may therefore be that different animals find different solutions, or that small differences in training protocol might bias the system to different solutions. The manuscript cites relevant papers at several points, but doesn't address the central point directly: it's likely that multiple solutions to the task would be possible in the brain, and the solution these animals' brains found may reflect details of training, life experience, chance, and other factors. A sentence raising this possibility seems warranted.

L72: the manuscript says that task C2 was performed better because of its unique response axis. This is probably true, but if I understand correctly C2 was also performed twice as often as C1 or S1. This should be mentioned.

Typos: Figure 3b, "representation"; L284: "Kendell's"

(Remarks on code availability)

We thank all three reviewers for their time and for their thoughtful insights into our manuscript. As we detail below, we have revised the manuscript to address all of the reviewers' concerns. In particular, we have added several analyses to support our conclusions (both in the main figures and extended data figures) and we significantly expanded the methods to describe our analyses and rationale in greater details. Finally, we have completely rewritten the Discussion based on the reviewer's feedback. We feel these revisions have greatly improved the manuscript and, again, would like to thank the reviewers for their feedback.

Here, we provide a detailed description of how we addressed each reviewer's comments. Reviewers' comments are provided in **black**. They are unedited, except for the addition of labels in order to facilitate referencing. Our responses are in **blue**. We also include the relevant sections of text in our response, written in *red italics*. In the manuscript, we have highlighted changes addressing the reviewers' comments in **red**.

Referee #1 (Remarks to the Author):

Tafazouli et al trained monkeys to discriminate visual stimuli that vary continuously in shape and colour, whilst recording neural signals from multiple brain regions. They trained monkeys to map shape onto response axis [S1], map colour onto response axis 1 [C1] and map colour onto response axis 2 [C2]. Their question concerns the nature of the neural representation that allows monkeys to jointly solve these tasks, and in particular the extent to which representations can be "re-used" to complete new tasks.

This is a high-quality paper that describes important findings. It is beautifully written and presented. The question is timely and will, I think, have significant impact. The authors have also generated an extremely valuable dataset with multi-region recordings from an innovative task. So there is a lot to like here.

We thank the reviewer for taking the time to review our paper. We appreciate their feedback and are glad to hear they liked the manuscript!

Here are some questions that occurred to me as I was reading:

Comment 1.1: I found the framing of the paper a little odd. The paper doesn't really seem to be about composition, at least not in the sense that we typically use the term in cognitive science. Composition is assembly of new tasks from existing building blocks. So for example, if I can do C2 and S1 then I should be able to "compose" task C1 from the combination of learning about stimulus axis C and response axis 1. But that's not what is happening here. In fact, in task C1, performance is overall hampered by its overlap with task S1, because half of the stimuli create response conflict (whereas they do not in C2 due to the unique response axis). To my mind, a demonstration of composition would be that the monkeys (for example) learn C1 more readily after exposure to S1/C2 compared to (say) some matched control task. Or that they learn faster on the 4th "missing" combination S2 (presumably the authors tried this?).

We thank the reviewer for raising this important point. We agree with the reviewer's definition of compositionality as assembling tasks from the 'building blocks' of other tasks. We believe our results are consistent with this. For example, our observation that the C1 task can be composed by combining subspaces of neural activity related to color categorization and the response along axis 1. We have revised the Discussion (lines 316-318) to make clear that we are studying one form of compositionality, the sequential composition of building blocks into more complex tasks:

Discussion:

Although our study is limited to three tasks, the underlying mechanism has the capability to be highly expressive – flexibly sequencing task components together could implement a wide variety of behaviors (a form of sequential compositionality⁴)

We also agree with the reviewer that compositionality is thought to be particularly useful when learning to perform new tasks, potentially even allowing for rapid (few shot) configuration of new tasks using the building blocks of other tasks. This is a very interesting question but, as the reviewer points out, would require training separate cohorts of animals with different learning curriculums. While we feel this is out of the scope for the current manuscript, we have revised the Discussion (lines 328-332) to highlight the importance of future work to study this phenomenon:

Discussion:

For example, learning the C2 task shapes the neural computations needed to categorize the color of a stimulus, which could generalize to other tasks that involve categorizing color, such as C1. Future computational and experimental work is needed to test whether having such foundational knowledge can facilitate learning new tasks.

Finally, we note the other advantages of shared representations, such as supporting continual learning:

Discussion (lines 328-331):

For example, learning the C2 task shapes the neural computations needed to categorize the color of a stimulus, which could generalize to other tasks that involve categorizing color, such as C1.

Discussion (lines 347-351):

One reason we found shared representations (rather than task-unique ones) may have been because our task required the animal to continuously learn. Across trials, the monkeys used feedback to infer the task context and to flexibly map stimuli onto motor responses (Fig. 4-5; a form of ‘class-incremental’ continual learning¹⁹). As noted above, shared representations facilitate continual learning by allowing knowledge to generalize between tasks⁵⁸.

We also agree with the reviewer that, on the surface, the C1 task seems to be impaired, relative to C2. As the reviewer suggests, this is likely due to its overlap with the S1 task. Specifically, that the animal is uncertain about the task (i.e., is it S1 or C1?). If true, then directly comparing the S1 and C1 tasks might provide insight into the relative balance between these tasks, as they both suffer from the same uncertainty/interference. Therefore, motivated by the reviewer’s comment, we tested whether there was an advantage to performing the C1 task or S1 task after performing the C2 task. We found behavior was consistently better by 15.1% when performing the C1 task during the first 15 trials of the block ($p=3.3 \times 10^{-16}$, Chi-squared test, Extended Data Fig. 2e,f, reproduced below). Importantly, this difference was not seen at the end of the block, suggesting that it was not due to a general difference in the animals’ ability to do color categorization versus shape categorization (0.5% difference, $p=0.69$, Chi-squared test; the

Extended Data Figure 2 | e-f, Comparison of behavioral performance of S1 (orange) and C1 (green) tasks **e**, in the first 15 trials of the block (after switch) and **f**, for the last 15 trials of the block (before switch). Red and green dotted line show mean behavioral performance for S1 and C1 tasks, respectively.

difference at the beginning of the block was significantly greater than the end of the block, two-way ANOVA to assess the effects of task (S1 vs. C1) and time (beginning vs. end of block) on performance $F(1,379)=300.39, p<0.001$).

We believe this provides behavioral evidence for the advantages of compositionality (with the important caveat that it is not demonstrating the full capabilities of compositionality, as noted above). Since, two out of three of the tasks involve categorizing color, the animal may have been biased in their expectation of the relevant categorization task. This would predict the relatively better performance at the start of the C1 block. These results, and their interpretation, are now included in the results (lines 66-73, Extended Data Fig. 2) and methods (lines 703-705):

Results:

Despite performing both tasks well, monkeys were initially biased towards the C1 task (Extended Data Fig. 2e-f, performance difference for first 15 trials was 15.1%, $p=3.3 \times 10^{-16}$, Chi-squared test). This suggests the monkeys initially expected to perform the C1 task, possibly because they categorized color on two out of three tasks (C1 and C2). In contrast, the animals had high performance on the C2 task immediately after the switch (Fig. 1f, 0.85 at 15 trials, $p<0.001$, binomial test). This was because the axis of response always changed between blocks and the C2 task was the only one to use Axis 2.

Extended Data Figure 2| b, Psychometric curve for congruent trials for C1 and C2 tasks using all trials in the block. Since congruency is undefined for the 50% morph line, there are no corresponding data points available. Behavioral performance on congruent stimuli, where there is no conflict between the C1 and S1 tasks, is similar between the C2 and C1 tasks. This consistent with the idea that the decrease in behavioral performance during the C1 task is due to interference from the S1 task (possibly due to uncertainty over which task to perform).

In addition, inspired by both this comment and the reviewer’s comment 1.5 below, we tested whether there was a difference in behavioral accuracy between the C2 and C1 tasks. If the animals are using the same color discrimination in both tasks, then we should expect to see similar behavioral performance. However, as seen in Figure 1e, there is a difference in overall performance in the two tasks. Again, we believe this reflects uncertainty about whether the C1 or S1 tasks is in effect. This would cause a decrease in performance on trials with incongruent stimuli that conflict between the C1 and S1 tasks. Therefore, to better compare the performance on the C1 and C2 tasks we focused on the congruent stimuli alone. As seen in Extended Data Figure 2b (reproduced above), behavioral performance was similar between the congruent stimuli of the C1 task and the C2 task using all trials. If anything, performance is slightly better on the C1 task (again, possibly reflecting the leakage of the S1 task).

Extended Data Figure 2| c Psychometric curve for color categorization performance when the shape morph level of the stimulus was ambiguous (50%/150%).

In addition, we tested the animal’s behavioral performance when the stimulus’ shape was ambiguous (50%, Extended Data Fig. 2c, reproduced to the right). Again, here, we find a psychometric curve that is closer to the C2 task.

Together, we believe these results support the conclusion that performance on the C1 and C2 tasks was similar, consistent with the observation of a shared category representation. We have revised the main text to note this observation (lines 73-76) and the methods to describe the analysis in detail (lines 706-710):

Results:

While overall behavioral performance was higher on C2 than C1 trials (Fig. 1e), this difference was likely due to the increased uncertainty about the task on C1 trials; there was no difference in performance for congruent trials or when the shape was ambiguous (Extended Data Fig. 2a-d).

Comment 1.2: In support of their claims about composition, the authors make quite a bit of hay about “representational subspaces being reused over tasks” (line 104 onwards). What they are referring to here is the fact that there neurons that care about colour in C1 also care about colour in C2 and those that care about response in C1 also care about response in S1. This is just like saying: some neurons code for sensory variables and others for motor variables. It doesn't really tell us much about composition, I don't think. The sequential engagement of these subspace just means that inputs are converted into outputs. Reassuring, but not spectacularly surprising.

We agree with the reviewer that it is important to distinguish our observations from a simple representation of the sensory stimulus or motor action. A couple pieces of evidence suggest this is not the case.

First, the stimulus representations decoded in our task are not the raw color or shape information, but the category of those stimuli. Therefore, the representation of the stimulus has already been transformed in a task-specific manner. Our results suggest that the same transformed representation is used across multiple tasks. We have revised the manuscript to make this point throughout the manuscript. We now explicitly state when we are quantifying information about the category of the stimulus. For example, in the discussion (lines 305-307):

Discussion:

Different subspaces of neural activity within prefrontal cortex represented different task-relevant information, including the color category and shape category of the stimulus^{24,39} and the motor action⁴⁰.

Second, it is important to note that compositional representations are not a foregone conclusion. In fact, previous theoretical and experimental work has argued against such compositionality, suggesting neurons in the brain respond to the non-linear conjunction of the current task and the sensory and motor components²⁷. Such task-unique representations captured the response of neurons in associative regions, such as prefrontal cortex, and provided a mechanism for learning multiple tasks without interference⁴⁹⁻⁵⁰. Consistent with this alternative, a recent publication suggests animals learn task-specific motor representations⁵¹. We have revised the Discussion to better highlight this alternative hypothesis (lines 324-328):

Discussion:

Our results contrast with previous work that found every task uses a unique representation of sensory and motor information⁴⁹⁻⁵¹. Task-unique representations have the advantage of minimizing interference when learning and performing multiple tasks⁵¹⁻⁵² but have limited ability to generalize learning across tasks^{3,26}. In contrast, shared subspaces increase interference but could speed learning by allowing knowledge to generalize across tasks⁶.

[Redacted text]

[Redacted text and figure]

Motivated by these results, we next tested whether shared representations were observed in other brain regions. Extended Data Figure 9b,c (reproduced below) shows color category information in Striatum, FEF, and PAR did not generalize between tasks or, if they did, the generalization was delayed relative to the task-specific category information (Extended Data Fig. 9b).

To quantify the difference between each region and LPFC, we subsampled the neurons in LPFC to create a population that had an equal number as what was recorded in each other region. While this reduced our statistical power, we consistently observed shared color representation between C1 and C2 tasks in the LPFC (with the exception of when matching to PAR, where neuron counts were too few). In comparison, generalization was reduced in FEF, PAR, STR, and aIT (Extended Data Fig. 9c).

b Within task and shared color category decoding
in Striatum, aIT, FEF and PAR

c Comparison of shared color category decoding in LPFC
to other regions (matched number of neurons)

Extended Data Figure 9| b, Time course of accuracy of classifier trained to decode color category within C1 task and across C1 and C2 tasks. Classifiers were trained separately for each brain region (Striatum, aIT, FEF, and LIP in each column, moving left to right). Lines and shading show mean \pm s.e.m. classification accuracy after stimulus onset. Distribution reflects 1000 iterations of classifiers. Horizontal bars along top indicate above-chance classification ($p \leq 0.05$, 0.01, and 0.001 for thin, medium, and thick lines, respectively; one-tailed t-test with no correction). **c**, Comparison of strength of shared color category responses in LPFC to other regions. All classifiers were trained to decode color category in C2 task and then tested on C1 task. LPFC neurons were down sampled to match each brain region (110 neurons for Striatum, 195 neurons for aIT, 116 neurons for FEF and 54 neurons for PAR). Accuracy was averaged from 100 ms to 300 ms after stimulus onset. Lines and shading show mean \pm s.e.m. ($p \leq 0.001$ for ***; unpaired t-test).

Together, these results suggest non-compositional representations exist in other brain regions, and it suggests prefrontal cortex may play a unique role in compositionally representing sensory and motor representations within the same network. We have revised the manuscript to include these new results in Extended Data Figure 9 and note the uniqueness of prefrontal cortex in results (lines 132-138) and discussion (lines 333-338):

Results:

While color category information was represented in a shared subspace in LPFC, it was represented in task-specific subspaces in other brain regions (Extended Data Fig. 9a). Generalization was weaker in FEF, PAR and aIT and was delayed with respect to task-specific sensory information (and was delayed relative to LPFC, Fig. 2g, *even when controlling for differences in number of neurons across regions, Extended Data Fig. 9b,c*). There was no significant generalization in STR (although stimulus color category could be decoded, Extended Data Fig. 5d).

Discussion:

Representations in prefrontal cortex generalized across tasks (Fig. 2)^{43,53}, although other regions, such as the hippocampus, are likely involved^{39,54}. Less generalization in parietal cortex, visual cortex, and striatum suggests neural representations are more task-unique in these regions.

Differences between regions may allow the brain to use the complementary advantages of both sharing representations (generalizing learning) and task-unique representations (avoiding interference)^{26,55}.

Fourth, motivated by the reviewer's comment (see also comment 5.9), we were curious as to whether there were 'some neurons that code for sensory variables and others for motor variables'. If true, then one would expect separate populations of neurons that encoded the sensory and motor responses. To test this, we used weights from the generalized linear model fits to investigate how task variables were distributed across the population.

We tested whether individual neurons carried significant information about one or more task variables. We found significantly overlapping populations of neurons for the majority of the task variables (Extended Data Fig. 4, reproduced below). In particular, we observed a significant overlap in the neurons encoding the response direction and color category (Extended Data Fig. 4c, $p(\text{response direction-color category})=0.0001$, permutation test) and a significant overlap in the neurons encoding response direction and shape category (Extended Data Fig. 4d, $p(\text{response direction-shape category})=0.0124$, permutation test). Similarly, the population of neurons encoding the color category and the shape category were overlapping (Extended Data Fig. 4j, $p(\text{shape category-color category})=0.0001$). And the neural population encoding reward overlapped with the population encoding response location, shape category, and color category (Extended Data Fig. 4a, f, and g, respectively; $p(\text{reward-response direction})=0.0001$, $p(\text{reward-color category})=0.0025$, $p(\text{reward-shape category})=0.0265$). Finally, while trending towards overlapping, we found more distinct populations of neurons encoded the task identity and reward as well as task identity and shape category ($p(\text{reward-task identity})=0.1085$, $p(\text{task identity-shape category})=0.0930$, permutation test).

These results confirm previous findings that neurons in our recorded regions respond conjunctively to multiple task variables⁴⁹. However, this conjunctive representation is structured²⁶, rather than a nonlinear mixed selective code that leads to a high-dimensional representation. Our findings demonstrate that this

Extended Data Figure 4| Individual neurons represent multiple task variables across regions. Each panel shows the proportion of neurons that significantly encode two task variables (left and middle columns) and encode both (right column). For all panels, *, **, and *** indicate $p \leq 0.05$, 0.01 , and 0.001 , respectively, all permutation tests. **a-d**, Response direction and **a**, reward, **b**, task identity, **c**, color category, and **d**, shape category. **e-g**, reward and **e**, task identity, **f**, color category, and **g**, shape category. **h-i**, task identity and **h**, color category and **i**, shape category. **j**, color category and shape category.

structured conjunctive representation contains shared subspaces for color category and response direction across multiple tasks.

Again, we thank the reviewer for suggesting this analysis as we feel it has added value to the manuscript. These results are now mentioned in the manuscript (lines 91-93):

Results:

All five regions represented task-relevant cognitive variables^{23,24}, including the identity of the current task, the color and shape of the stimulus, the response direction, and whether a reward was received (Fig. 1h-1l). The identity of the task was consistently represented throughout the trial (Fig. 1j). After stimulus onset, information about the color category of the stimulus²⁵ (Fig. 1h) and shape category of the stimulus (Fig. 1i) was followed by information about the direction of the animal's response (Fig. 1l), and then the reward received (Fig. 1k, see Figs. 1m and S3 for average time course across all regions). Individual neurons were 'mixed-selective' in all of these regions, representing multiple task variables (Extended Data Fig. 4)^{26,27}.

Comment 1.3: The analyses of task sequences (line 161 onwards) are cool. The monkeys seem to have a prior on task repetition that is relatively enduring. But I wasn't clear about the apparently contradiction between behaviour (Fig. 4a) and classification performance (Fig. 4g). In an C2-C1 sequence, behaviour improves if the previous task was C1 (rather than S1) but the opposite is true for colour decoding; and yet these effects are positively correlated (Fig 4e). The authors sort of gesture to the fact that perhaps the monkey "re-learning" C1 is driving the neural effect, but then why the converse behavioural advantage? This seems to warrant a bit more discussion.

We thank the reviewer for raising this important point. As noted by the reviewer, the sequence of tasks influences the animals' behavior – they perform better during a C1-C2-C1 sequence than a S1-C2-C1 sequence (Fig. 4a). Consistent with this, the animal's belief that it is performing the C1 task is greater in the C1-C2-C1 sequence compared to the S1-C2-C1 sequence (Fig. 4d). Furthermore, task belief is correlated with behavior in S1-C2-C1 blocks (Fig. 4e).

As detailed in the manuscript, one might expect a similar pattern for the color category classifier – as the animal's belief they are performing the C1 task increases, they should increasingly engage the color category classifier. This seems to be the case for the S1-C2-C1 task sequence (Fig. 4g, grey line; correlation in Fig. 4h). However, there is not a strong and significant correlation between behavioral performance and color category information during the C1-C2-C1 block sequences (Fig. 4g, pink; $\tau = -0.543$, $p = 0.207$, permutation test). Again, we thank the reviewer for noting this, as it motivated several new analyses that led to new insights.

As the reviewer points out, on C1-C2-C1 blocks, behavioral performance starts high and only improves slightly during the block (68% on trial 15 to 78% on trial 75 in C1-C2-C1 blocks, compared to 58% on trial 15 to 76% on trial 75 in S1-C2-C1 blocks). Given this, there may not need to be a change in the strength of color representation during the block. Indeed, the competition from the shape category is low throughout C1-C2-C1 blocks, reflected in the fact that shape information is at or below chance levels (Fig. 4j). In other words, performance may already be 'optimized' and therefore a change may not be needed. If true, one might expect LPFC to become less engaged as the task becomes more certain. Previous work has suggested that LPFC is most strongly engaged when cognitive control is needed – that is, when the task is uncertain³⁴. Therefore, the decrease in task representations could reflect the engagement of other neural circuits representing color during the C1-C2-C1 sequence in comparison to the S1-C2-C1 sequence. To test this, we computed the average accuracy of color category classifier for C1-C2-C1 and S1-C2-C1 sequences for all regions. As seen in Extended Data Figure 11e (reproduced below), color category information in FEF was greater during the C1-C2-C1 sequence compared to S1-C2-C1 sequence ($\Delta(\text{average accuracy C1-C2-C1 and S1-C2-C1 sequences}) = 12.7\%$, $p = 0.004$). This suggests there may be a consolidation of color information into FEF during the C1-C2-C1 sequences.

Again, we thank the reviewer for this comment as we think it has provided deeper insight into the mechanisms allowing the animal to infer, and perform, the current task. These results are now included in the manuscript (lines 236-242):

Results:

Like task representations, color representations depended on the task sequence (Fig. 4g). While shared representations increased when discovering C1 following a S1-C2-C1 sequence, they remained stable during C1-C2-C1 sequences (63% to 62%, $\Delta=-1\%$ from trial 40 to 110, $p=0.808$, permutation test). While this is consistent with better performance at the beginning of the C1 task during C1-C2-C1 sequences, it doesn't explain the improvement in behavior over trials during the C1-C2-C1 sequence (Fig. 4a). This difference in performance was associated with an increase in color information in FEF during C1-C2-C1 sequence (Extended Data Fig. 11e, difference in accuracy between C1-C2-C1 and S1-C2-C1 was 12.7%, $p=0.004$), possibly reflecting the consolidation of color category information from LPFC into FEF as the animal became more certain about the current task.

Extended Data Figure 11| e, Comparison of progression of color category classifier accuracy in C1-C2-C1 task sequence (purple) and S1-C2-C1 task sequence (black) for FEF. Average accuracy was computed using classifier performance in 200ms to 400ms after stimulus onset period. Lines and shading show mean \pm s.e.m. classification accuracy after stimulus onset. Distribution reflects 250 iterations of classifiers. Δ (average accuracy S1-C2-C1 and C1-C2-C1 sequences)=12.7%, $p=0.004$. On C1-C2-C1 blocks, behavioral performance starts high and only improves slightly during the block (68% on trial 15 to 78% on trial 75 in C1-C2-C1 blocks, compared to 58% on trial 15 to 76% on trial 75 in S1-C2-C1 blocks). Given this, there may not need to be a change in the strength of color representation during the block. Indeed, the competition from the shape category is low throughout C1-C2-C1 blocks, reflected in the fact that shape information is at or below chance levels (Fig. 4j). In other words, performance may already be 'optimized' and therefore a change may not be needed. If true, one might expect LPFC to become less engaged as the task becomes more certain. Previous work has suggested that LPFC is most strongly engaged when cognitive control is needed – that is, when the task is uncertain⁷⁰. Therefore, the decrease in task representations could reflect the engagement of other neural circuits, such as FEF, as shown here, to represent the color category during the C1-C2-C1 sequence in comparison to the S1-C2-C1 sequence.

Comment 1.4: Although superficially symmetric, there is an obvious difference between C2 and the other tasks, in that it the monkey knows unambiguously from the first feedback that this is the rule. I would have been interested in knowing a bit more about how the representation of C2 differs from C1 and S1; can we understand more about how monkeys ensure that color representations in the C2 task do not inadvertently prompt responses on axis 1?

This is another interesting question. As noted by the reviewer, a consequence of the experimental design was that the animal was able to quickly infer the C2 task after a block switch. This is reflected in the high level of behavioral performance after a few trials of switching into the C2 task (Fig. 1f). Consistent with the reviewer's prediction, the monkey rarely makes responses on the wrong axis (i.e., onto Axis 1 during the C2 task). We now include this information in Extended Data Figure 2g,h (reproduced below).

This rapid and reliable switch was reflected in the neural activity. To study this, we trained a classifier to decode the axis of response using neural activity during the fixation period (i.e., -400ms-0ms before stimulus onset). Training was restricted to the last 75 trials of the C2 and C1 tasks, when behavioral performance was high. We then applied the classifier to the beginning of the C1 task blocks to assess the

Extended Data Figure 2 | g, Proportion of responses on incorrect axis for C1, C2 and S1 tasks. **h**, Proportion of responses on incorrect axis during the S1 task (left), C2 task (middle) and C1 task (right) did not depend on the sequence of tasks (S1-C2 in orange and C2-C2 in green).

representation of the response axis immediately after the switch. Consistent with behavior, the response axis was decodable immediately after the switch in LPFC (Fig. 5h, reproduced below; behavior is shown in Extended Data Fig. 2g). However, there was a slight, but significant, increase in classifier performance within 3-6 trials, indicating that the belief about the response axis is quickly updated ($\Delta(\text{performance in trial 10 and trial 16 in LPFC})=6.88\%$, $p=0.008$, permutation test). As shown in Extended Data Figure 14 j,k (reproduced below), the response axis was decodable in FEF and Striatum but there was not a significant increase with discovery (and the response axis was not decodable in PAR and aIT during the fixation period, $\Delta(\text{performance in trial 10 and trial 73}):$ Striatum= -16.67% , $p=0.9750$, aIT= -12.17% , $p=0.9670$, FEF= 2.76% , $p=0.2967$, PAR= 4.00% , $p=0.2098$, all one tailed permutation test).

Fig. 5 | h, Classifier accuracy for response axis decoding in LPFC. Darker to lighter color show progression in trial blocks of 10 shifted by 3 trials from switch trial in C1 task in S1-C2-C1 task transition. Lines show mean classification accuracy after stimulus onset for 250 iterations of classifiers. Horizontal bar indicates significant increase in classifier accuracy from trial 10 to trial 16 (permutation test with cluster mass correction for multiple comparisons across time).

Extended Data Figure 14 | j, As in Fig. 5h, but shows classifier accuracy in decoding the response axis for all brain regions. Darker to lighter color show progression in trial blocks of 10 shifted by 3 trials from switch trial in C1 task. Lines show mean classification accuracy after stimulus onset for 250 iterations of classifiers. Arrows indicate the time course of example trials. **k**, Classifier accuracy in decoding the response axis for all brain regions. Decoding using neural activity in -400ms to 0ms before stimulus onset period. Lines and shading show mean \pm s.e.m. classification accuracy across 250 iterations of classifiers.

As suggested by the reviewer, this leads to an interesting question – what is the mechanism by which the animal suppresses Axis 1 responses during the C2 task (and vice versa, suppresses Axis 2 responses during the C1/S1 task). One possibility is that information in the color category subspace is transformed into the task-appropriate response direction. This is consistent with the correlations observed in Fig. 3c-d.

One mechanism for facilitating one transition (e.g., color category to Axis 2) while avoiding another (e.g., color category to Axis 1) is by modulating the neurons encoding the task-relevant and task-irrelevant axes. This would be a similar mechanism to the relative scaling of color and shape representations during C1/S1 task blocks. To test whether we observe modulation of the motor axis representation, we examined the weights of the axis classifier to identify LPFC neurons that encoded the response axis. We then categorized the neurons into three groups based on their classifier weights during the response period (200ms to 450ms after stimulus onset). Neurons with significant classifier weights for Axis 1 or Axis 2 were classified as Axis 1 or Axis 2 selective neurons, respectively (green and orange dots in Extended Data Fig. 14a). Neurons without significant classifier weights were classified as non-axis selective (grey dots in Extended Data Fig. 14a).

Extended Data Figure 14| Neurons encoding response on Axis 1 and Axis 2 are suppressed when irrelevant to the task. **a**, Classifier weights for classifying axis of response for Fig. 5g. Classifier was trained to decode response axis using C1 and C2 task trials. Neurons with significant weights for Axis 1 and Axis 2 are denoted with green and orange dots, respectively. Non-selective neurons are denoted with gray dots. **b**, Follows Fig. 5g, time course of average firing rate for non-selective neurons in LPFC. Inset shows p-value of difference in firing rate when animal responded on Axis 1 compared to responding on Axis 2 (paired t-test). Lines and shading show mean \pm s.e.m. **c**, Follows Fig. 5g, time course of average firing rate of neurons when the animal responded on Axis 1 (dashed) or Axis 2 (solid). Neurons were grouped by their preferred response on Axis 1 (left) or Axis 2(right). Shaded blue box indicates the time period used to define the neurons preferred response axis (200ms-450ms after stimulus onset). **d**, Follows Fig. 5g. Average firing rate of LPFC neurons during trials when the animal responded on Axis 1 (green) and Axis 2 (pink). Three panels show responses of three categories of neurons: those that responded most strongly on Axis 1 (left), Axis 2 (middle), and were not selective (right) in LPFC. To characterize axis selectivity, classifier was trained to decode response axis using trials from S1 and C2 trials and then firing rate was estimated on withheld C1 trials. Inset shows p-value for of difference between firing rate in response on Axis 1 and response on Axis 2 trials in time periods -600ms-0ms (lower bracket) and -600ms-600ms (upper bracket) after stimulus onset (paired t-test). **e**, As in panel d, but axis selectivity was defined by training a classifier to decode response axis using trials from C1 and C2 trials and then firing rate was estimated on withheld S1 trials. **f-i**, As in Fig. 5g, but shown for **f**, FEF; **g**, aIT; **h**, PAR and **i**, Striatum. Similar to LPFC, Axis 1 and Axis 2 selective neurons in FEF, aIT and PAR were suppressed during response on opposite axis. Lines and shading show mean \pm s.e.m. firing rate across individual neurons for each region. Inset shows p-value for of difference between firing rate in response on Axis 1 and response on Axis 2 trials in time periods -600ms-0ms (lower bracket) and -600ms-600ms (upper bracket) after stimulus onset (paired t-test). **j**, As in Fig. 5h, but shows classifier accuracy in decoding the response axis for all brain regions. Darker to lighter color show progression in trial blocks of 10 shifted by 3 trials from switch trial in C1 task. Lines show mean classification accuracy after stimulus onset for 250 iterations of classifiers. Arrows indicate the time course of example trials. **k**, Classifier accuracy in decoding the response axis for all brain regions. Decoding using neural activity in -400ms to 0ms before stimulus onset period. Lines and shading show mean \pm s.e.m. classification accuracy across 250 iterations of classifiers.

Although neurons were defined based on their activity during the motor response, they were still modulated during the baseline time period (Fig. 5g, reproduced below). Axis 1 selective neurons significantly increased their firing rate during Axis 1 trials and were suppressed during Axis 2 trials, even before stimulus onset (Fig. 5g, left). This pattern was reversed for Axis 2 selective neurons (Fig. 5g, right). Non-axis selective neurons showed no significant difference in firing rates during Axis 1 and Axis 2 trials (Extended Data Fig. 14b). Similar patterns were observed when we performed further controls (Extended Data Fig. 14c-e) and in other regions (Extended Data Fig. 14f-i). These results suggest the brain may actively suppress the irrelevant motor response, facilitating the task-relevant transformation (and inhibiting the task-irrelevant transformation).

Figure 5| g, Average firing rate (FR) for Axis 1 selective (left) and Axis 2 selective (right) neurons in LPFC. C1 task trials were divided into two groups: response on Axis 1 (correct axis, green) and response on Axis 2 (incorrect axis, purple). Axis 1 selective neurons had higher activity before stimulus onset when the animal responded on Axis 1 (and vice versa when responding on Axis 2). Inset shows p-value of difference over entire time period (paired t-test)

Together, these results are consistent with the idea that the neurons encoding a task-irrelevant motor axis were quickly suppressed after a switch. This suppression was seen throughout the entire trial and may help to prevent eye movements to the incorrect response axis while performing the C2 or C1/S1 tasks. We thank the reviewer for suggesting these analyses as we think they have added new insight into the mechanisms of task-switching.

These results are now included in the main text (lines 286-299) and Fig. 5g, Fig. 5h and Extended Data Fig. 14:

Results:

In addition to selecting sensory information, the three tasks also require selecting the appropriate motor response: the animal must suppress responses on Axis 1 during the C2 task and Axis 2 during the C1 task. Consistent with scaling acting as a general mechanism for modulating task-(ir)relevant information, neurons selective for each motor axis were suppressed when the animal performed a task requiring a response on the other axis (Fig. 5g, see Extended Data Fig. 14c-e for further controls and Extended Data Fig. 14f-i for other regions). However, in contrast to the gradual suppression of sensory representations during task discovery, the incorrect response axis representation in LPFC was quickly suppressed after a change in task (within 3-6 trials, Fig. 5h, Δ (classifier performance in trial 10 and trial 16)=6.88%, $p=0.008$, permutation test, see Extended Data Fig. 14a,k for other regions). This reflected the rapid inference of response axis seen in the animals' behavior (Extended Data Fig. 2g,h).

Results:

Together, these results suggest the monkeys' belief about the current task dynamically adjusted the magnitude of sensory and motor representations³⁶⁻³⁸. Scaling neural representations was a common mechanism used in all the three tasks and could facilitate task-relevant associations between stimulus category and motor response, while preventing task-irrelevant associations.

And in the Discussion (lines 339-346):

Discussion:

Given that the animals were trained for months on the tasks, one might expect the brain to form independent task representations in order to reduce interference^{4,51,56}. Instead, we found the brain reduced interference by engaging task-relevant sensory and motor representations and suppressing task-irrelevant dimensions^{36,42,57}. Importantly, the degree of engagement/suppression depended on the animals' internal belief about the task state (Fig. 5), allowing the brain to flexibly modulate how color and shape category influenced behavior and ensuring the animal performed the appropriate response (e.g., responding on Axis 1, not Axis 2, during C1 and vice versa for C2).

And the details of the approach are included in the Methods (lines 1220-1268).

Comment 1.5: The behavioural advantage for C2 does not seem to be solely explained by the faster relearning following a switch (because it persists deep into each block (1f) and is actually expressed as heightened psychometric slope and not (say) an increase in bias or lapses (Fig 1e). Given the shared coding between C2 and C1 at the input level, there must be a process that protects C2 in a way that allows its (quite substantially) superior performance. This question isn't really (as far as I could see) addressed by the analysis of compression described below. Do the monkeys monitor for C1 / S1 and treat C2 as an "exception" that is qualitatively distinct in some way?

Extended Data Figure 2| d. Psychometric curve for C1 and C2 tasks when proportion of choices was computed based on responses according to irrelevant feature (shape). Shape affects behavioral responses during the C1 task but not the C2 task. This is again consistent with the idea that the decrease in behavioral performance during the C1 task is due to interference from the S1 task.

We thank the reviewer for raising another interesting question. As noted above, we feel this comment is closely related to Comment 1.1 and so our response is partially overlapping.

As the reviewer suggests, the C2 task is exceptional because there is little uncertainty as to what task the animal should be doing. This is different from the C1 task where the animal can be uncertain as to whether they should be performing the C1 or S1 task (even deep into a block). We believe this uncertainty can explain the difference in psychometric slope between the C2 and C1 tasks.

This is perhaps most directly seen by plotting the effect of the irrelevant shape dimension on behavior during the C2 and C1 tasks. As expected, shape affects behavioral responses during the C1 task but not the C2 task (Extended Data Fig. 2d, reproduced above). This is consistent with the idea that the decrease in behavioral performance during the C1 task is due to interference from the S1 task (possibly due to uncertainty over which task to perform). Several other lines of evidence support this conclusion. Behavioral performance on congruent stimuli, where there is no conflict between the C1 and S1 tasks, is similar between the C2 and C1 tasks (Extended Data Fig. 2b, reproduced to the right). And the animal's behavioral performance when the stimulus' shape was ambiguous (50%) was similar in the C1 and C2 tasks (Extended Data Fig. 2c, reproduced below). Altogether, these results suggest the difference between the two tasks is due to the animal's uncertainty about whether they should be performing the C1 or S1 task.

In addition to the new figures, we now describe the results and their interpretation in the manuscript (lines 73-76):

Results:

While overall behavioral performance was higher on C2 than C1 trials (Fig. 1e), this difference was likely due to the increased uncertainty about the task on C1 trials; there was no difference in performance for congruent trials or when the shape was ambiguous (Extended Data Fig. 2a-d).

Extended Data Figure 2| b, Psychometric curve for congruent trials for C1 and C2 tasks using all trials in the block. Since congruency is undefined for the 50% morph line, there are no corresponding data points available. Behavioral performance on congruent stimuli, where there is no conflict between the C1 and S1 tasks, is similar between the C2 and C1 tasks. This consistent with the idea that the decrease in behavioral performance during the C1 task is due to interference from the S1 task (possibly due to uncertainty over which task to perform).

Extended Data Figure 2| c, Psychometric curve for color categorization performance when the shape morph level of the stimulus was ambiguous (50%/150%, left), intermediate (30%/70%/130%/170%, middle) and prototypical (0%/100%, right).

Comment 1.6: Perhaps the most impactful results from this study are potentially those relating to compressive coding of irrelevant dimensions. Since 2013, a canonical story has emerged that in this type of task-switching setting, there is no compression of irrelevant alternatives, at least in FEF. This question is hard to pose because in most prefrontal regions most neurons care about the choice axis (so in the Mante / Newsome Sussillo paper the claim about lack of compression concerns the subspace orthogonal to choice; in other work, e.g. that from Miller / Siegel, the choice axis dominates). Here, we have potential evidence for powerful compression; this could be a significant finding, and opens up a new chapter in this debate.

We thank the reviewer for their enthusiasm and agree that our results suggest that the degree of compression may be variable across individuals and tasks.

Comment 1.7: The results in Figure 5 look compelling but I have two questions. Firstly, I'm presuming that the results described are from LPFC exclusively; can the authors comment on what they found in FEF, for direct comparison with Mante 2013 (or in other regions)?

We initially focused our analyses on lateral prefrontal cortex because it was the region that showed the greatest sharing of sub-space representations across tasks. However, we agree it would be interesting to look at other regions to provide comparison with LPFC and with other manuscripts.

Extended Data Figure 13c (reproduced below) shows the compression index for additional regions, including the frontal eye fields (FEF), parietal cortex (PAR), and inferotemporal cortex (aIT). The compression index in FEF is similar to the lateral prefrontal cortex (LPFC), with greater separation of color representations during C1 and C2 tasks and greater separation of shape representations during S1 tasks (Extended Data Fig. 13c, left panel). Compression was reduced in PAR overall but showed a similar pattern with greater separation of color in the C2 task and shape in the S1 task (C1 was less clear). The compression index in aIT was close to zero for all tasks. This is consistent with previous findings²⁶ that sensory regions do not show task-dependent compression of features. Finally, it is important to note that the striatum did not show significant color or shape information, and so the compression index is not reliable.

C Similar to LPFC, irrelevant task dimensions are compressed in FEF and PAR

Extended Data Figure 13| Follows Fig. 5c. Compression index of neural responses in FEF, aIT and PAR over time, for all three tasks. Compression index is taken as the log of the ratio of the separability of stimuli in color and shape subspaces using classifier encoding axis. The compression index in FEF is similar to the lateral prefrontal cortex (LPFC), with greater separation of color representations during C1 and C2 tasks and greater separation of shape representations during S1 tasks. Compression was reduced in PAR overall but showed a similar pattern with greater separation of color in the C2 task and shape in the S1 task (C1 was less clear). The compression index in aIT was close to zero for all tasks. This is consistent with previous findings⁷² that sensory regions show limited task-dependent compression of features. Striatum did not show significant color or shape information, and so the compression index is not reliable.

These results are now included in the manuscript. We mentioned them briefly in the main text (lines 280-282) and they are detailed in the legend of Extended Data Figure 13:

Results:

Stimulus representations were also scaled in FEF and PAR, but not aIT, consistent with aIT maintaining a veridical representation of the stimulus (Extended Data Fig. 13a-c)²⁵.

Comment 1.8: Secondly, the monkeys make different responses along response axis 1 in tasks C1 and S1; can the authors be sure that the results are not secondary to neurons simply coding for the choice axis? Given that the response targets are fixed in each block, for example in C1 different colours will map onto different responses whereas different shapes will be orthogonal to this axis; the converse is true in S1. Do the results persist when the choice axis is removed, as in Mante (2013)?

We thank the reviewer for raising this point. Indeed, it is critical to ensure our estimate of the compression of color (and shape) is not simply reflecting the animal's motor action. To address this concern, in order to estimate distance on color axis we use the color classifier from the C2 task and apply it to the C1 task. Because the C2 and C1 task require different responses, cross-task decoding is less influenced by potential confounds between stimulus identity and response (see Extended Data Fig. 8a for cross decoding of response location in C1 and C2 tasks). In addition, when estimating the distance on the shape axis, we trained the shape category classifier on a set of S1 trials with balanced response directions to remove motor action information. As seen in Extended Data Figure 13c (reproduced above) we see differences in the strength of color information in the C1 and S1 tasks, even when controlling for motor responses.

Finally, as suggested by the reviewer, we computed the compression index by removing the response direction axis from tasks. This was done by using the targeted dimensionality reduction (TDR) as described in Mante et al 2013. In short, after defining orthogonal axes of color, shape, response on Axis 1 and response on Axis 2, we projected the activity of the neural population only onto the color and shape axes. Extended Data Figure 13a (reproduced below) shows the average distance along the shape and color axes after stimulus onset for each task (100ms-400ms). Points above the diagonal indicate a greater encoding along the shape axis compared to the color axis (i.e., greater encoding of shape than color). In contrast, points below the diagonal line indicate greater distance along the color axis than the shape axis (i.e., greater color encoding than shape). Similar to our previous findings, LPFC, FEF and PAR all showed greater separation of color representations during the C1 and C2 tasks and greater separation of shape representations during the S1 task (Extended Data Fig. 13b). As above, the encoding in aIT did not seem to be affected by task (always greater representation of color). Overall, these results are similar to observed when the compression index was computed based on the projection of neural activity onto the classifier shape and color encoding axes (Extended Data Fig. 13c, using $CPI = \log\left(\frac{\text{Distance on color axis}}{\text{Distance on shape axis}}\right)$).

We have revised the manuscript (lines 279-282) and figure legends Figure 5a,b and Extended Data Figure 13a,b to make this point clear:

Results:

Color and shape representations were scaled in all three tasks. There was greater separation of color representation during C1 and C2 tasks, and greater separation of shape representations during the S1 task (Fig. 5c; similar results were found when controlling for motor response, Extended Data Fig. 13a,b). Stimulus representations were also scaled in FEF and PAR, but not aIT, consistent with aIT maintaining a veridical representation of the stimulus (Extended Data Fig. 13a-c)²⁵.

Figure legend 5a:

To control for motor response, classifiers were trained to decode color category using trials from C2 task and were tested on trials from C1 task.

Figure legend 5b:

To control for motor response, shape category classifier was trained on balanced response direction trials from S1 task and was tested on trials from C1 task.

Figure legend Extended Data Figure 13a,b:

a, Normalized distance on color and shape orthogonal axes for LPFC, FEF, PAR and aIT (in each column, moving left to right). We computed the compression index by removing the response direction axis from tasks. This was done using targeted dimensionality reduction (TDR), as previously described⁷¹. After defining orthogonal axes of color, shape, response on Axis 1 and response on Axis 2, we projected the activity of the neural population only onto the color and shape axes. Figure shows the average distance along the shape and color axes after stimulus onset for each task (100ms-400ms). Points above the diagonal indicate a greater encoding along the shape axis compared to the color axis (i.e., greater encoding of shape than color). In contrast, points below the diagonal line indicate greater distance along the color axis than the shape axis (i.e., greater color encoding than shape). Similar to panel c and Fig. 5c, LPFC, FEF and PAR all showed greater separation of color representations during the C1 and C2 tasks and greater separation of shape representations during the S1 task. Encoding in aIT was not affected by task (always greater representation of color). b, Compression index computed using TDR method for LPFC, FEF, PAR

Extended Data Figure 13| Color and shape representations were scaled by task belief in all regions, except aIT.

a, Normalized distance on color and shape orthogonal axes for LPFC, FEF, PAR and aIT (in each column, moving left to right). We computed the compression index by removing the response direction axis from tasks. This was done using targeted dimensionality reduction (TDR), as previously described⁷¹. After defining orthogonal axes of color, shape, response on Axis 1 and response on Axis 2, we projected the activity of the neural population only onto the color and shape axes. Figure shows the average distance along the shape and color axes after stimulus onset for each task (100ms-400ms). Points above the diagonal indicate a greater encoding along the shape axis compared to the color axis (i.e., greater encoding of shape than color). In contrast, points below the diagonal line indicate greater distance along the color axis than the shape axis (i.e., greater color encoding than shape). Similar to panel c and Fig. 5c, LPFC, FEF and PAR all showed greater separation of color representations during the C1 and C2 tasks and greater separation of shape representations during the S1 task. Encoding in aIT was not affected by task (always greater representation of color). b, Compression index computed using TDR method for LPFC, FEF, PAR and aIT (in each column, moving left to right). Compression index is taken as the log of the ratio of the separability of stimuli in color and shape orthogonal axes (averaged in the period 100ms to 400ms after stimulus onset).

and aIT (in each column, moving left to right). Compression index is taken as the log of the ratio of the separability of stimuli in color and shape orthogonal axes (averaged in the period 100ms to 400ms after stimulus onset).

And the details of the approach are included in the Methods (lines 1290-1363).

Minor points

Figure 5c - the line colour and legend do not seem to match

We thank the reviewer for noting these issues. We have updated Figure 5c to match the colors.

Figure 4e – that linear trend looks a bit uncomfortable on that plot

We agree and so have updated the correlation analysis to use a non-parametric test, Kendall's tau, with statistical significance determined with a permutation test. Additionally, we have revised Figure 4e (reproduced below) to illustrate the correlation between task belief and behavioral performance during the fixation period, before the onset of the stimuli.

Fig. 4| e, Correlation of behavioral performance and task belief encoding in S1-C2-C1 task sequence. Z-scored Kendall's tau with permutation test and cluster mass correction for multiple comparisons across time.

Referee #2 (Remarks to the Author):

Tafazoli et al present findings on a “compositional” task in monkeys, in which LPFC activity in particular uses overlapping subspaces for shared stimuli (color with two different saccade axis outputs) or shared outputs (color or shape for the same saccade direction). This is a highly challenging task, and the authors carefully analyze the behavior and use a variety of classifier generalization procedures to demonstrate the central finding. It is technically sound. There is some new science here: the reuse of dimensions is not exactly known in LPFC (though something like it is known elsewhere in frontal cortex), and maybe more novel is that LPFC compresses the irrelevant stimulus dimension dependent on task belief. This is all reasonably interesting, but doesn't have the scientific depth to it that I would have expected at this journal.

The paper does have some real strengths. This task involves blockwise interleaving of 3 different subtasks, which was apparently very challenging to train, and where good performance was attained. The authors recorded from 5 brain areas, though mostly only LPFC was the focus here. The analysis of the task was creative and in-depth, and leads to a good understanding of what the monkey is doing in this complex task. The application of encoding models gives a general idea of what's present in these areas, and the extensive use of linear decoders makes clear that some dimensions are in fact used across the expected related subtasks. The task-belief-dependence of the stimulus representation suppression is, to my knowledge, novel. The pieces are all executed well, and my technical concerns are minor.

We thank the reviewer for taking the time to review our manuscript and we are happy to hear they find our work creative, in-depth, and technically sound. We also appreciate their criticisms and, as we detail below, we have revised the manuscript to address them.

Conceptual limitations

The central problem here is that I don't find these results to be all that surprising, and I don't think many in the field would. As I discuss below, I think one of the two main findings was basically known already, and the other has already been largely superseded with additional results and an extensive theoretical framework.

Comment 2.1: The first main finding is that the stimulus subspace (really a single decoded dimension) is conserved across tasks sharing a stimulus, and the movement subspace (again, a single decoded dimension) is shared across tasks sharing a response. Both were previously found by Mante 2013 in FEF, and are now known in rat FOF in an analogous task (Pagan et al. 2022 bioRxiv, though not yet peer reviewed). Moreover, this is what you would expect given visual inputs to the structure and the widespread prevalence of movement-related activity throughout cortex. To my knowledge this wasn't established in this exact area, and certainly not in this very nice, novel task. This makes the effort worthwhile. But, it's also not very surprising, so the amount of scientific progress it represents is limited.

We apologize for not making the contribution of our manuscript clearer. In our view, our results demonstrate that the brain can 'mix-and-match' different subspace representations across multiple tasks. By using three compositionally related tasks, we show that the color representation from one task (C2) and the response representations from a second task (S1) can be combined to explain neural responses on a third task (C1). We have revised the manuscript to make this clearer. In the Introduction, we now state (lines 35-38):

Introduction:

Whether the brain similarly reuses sensory, cognitive, and/or motor representations across tasks remains unknown. Furthermore, we do not yet understand how the brain flexibly engages these representations to continually adapt to the changing demands of the environment¹⁹. To address these questions, we trained monkeys to flexibly switch between three different compositionally related tasks.

And, in the Discussion (lines 304-311), we acknowledge previous work and highlight our advance:

Discussion:

Our results suggest the brain can perform a task by compositionally engaging a series of representational subspaces. Different subspaces of neural activity within prefrontal cortex represented different task-relevant information, including the color category and shape category of the stimulus^{24,39} and the motor action⁴⁰. Consistent with computational models^{4,5,7,41}, these subspaces were shared across multiple tasks (Fig. 2), suggesting they act as 'task components'^{24,42,43}. Subspaces were sequentially engaged, such that information from the relevant sensory subspace was transformed into the appropriate motor response subspace (Fig. 3)⁴⁴.

We also want to note that the observation of shared subspaces is not a foregone conclusion. In fact, previous theoretical and experimental work has argued against such compositionality, suggesting neurons in the brain respond to the non-linear conjunction of the current task and the sensory and motor components²⁷. Such task-unique representations captured the response of neurons in associative regions, such as prefrontal cortex, and provided a mechanism for learning multiple tasks without interference^{49,50}. Consistent with this alternative, a recent publication suggests animals learn task-specific motor representations⁵¹. We have revised the Discussion to better highlight this alternative hypothesis (lines 324-328):

Discussion:

Our results contrast with previous work that found every task uses a unique representation of sensory and motor information⁴⁹⁻⁵¹. Task-unique representations have the advantage of minimizing interference when learning and performing multiple tasks⁵¹⁻⁵² but have limited ability to generalize learning across tasks^{3,26}. In contrast, shared subspaces increase interference but could speed learning by allowing knowledge to generalize across tasks⁶.

[Redacted text and figure]

Motivated by these results, we next tested whether shared representations were observed in other brain regions. Extended Data Figure 9b,c (reproduced below) shows color category information in Striatum, FEF, and PAR did not generalize between tasks or, if they did, the generalization was delayed relative to the task-specific category information (Extended Data Fig. 9b).

To quantify the difference between each region and LPFC, we subsampled the neurons in LPFC to create a population that had an equal number as what was recorded in each other region. While this reduced our statistical power, we consistently observed shared color representation between C1 and C2 tasks in the LPFC (with the exception of when matching to PAR, where neuron counts were too few). In comparison, generalization was reduced in FEF, PAR, STR, and aIT (Extended Data Fig. 9c).

Together, these results suggest non-compositional representations exist in other brain regions, and it suggests prefrontal cortex may play a unique role in compositionally representing sensory and motor representations within the same network. We have revised the manuscript to include these new results in Extended Data Figure 9 and note the uniqueness of prefrontal cortex in results (lines 132-138) and discussion (lines 333-338):

Extended Data Figure 9| b, Time course of accuracy of classifier trained to decode color category within C1 task and across C1 and C2 tasks. Classifiers were trained separately for each brain region (Striatum, aIT, FEF, and LIP in each column, moving left to right). Lines and shading show mean \pm s.e.m. classification accuracy after stimulus onset. Distribution reflects 1000 iterations of classifiers. Horizontal bars along top indicate above-chance classification ($p \leq 0.05$, 0.01 , and 0.001 for thin, medium, and thick lines, respectively; one-tailed t-test with no correction). **c**, Comparison of strength of shared color category responses in LPFC to other regions. All classifiers were trained to decode color category in C2 task and then tested on C1 task. LPFC neurons were down sampled to match each brain region (110 neurons for Striatum, 195 neurons for aIT, 116 neurons for FEF and 54 neurons for PAR). Accuracy was averaged from 100 ms to 300 ms after stimulus onset. Lines and shading show mean \pm s.e.m. ($p \leq 0.001$ for ***; unpaired t-test).

Results:

While color category information was represented in a shared subspace in LPFC, it was represented in task-specific subspaces in other brain regions (Extended Data Fig. 9a). Generalization was weaker in FEF, PAR and aIT and was delayed with respect to task-specific sensory information (and was delayed relative to LPFC, Fig. 2g, *even when controlling for differences in number of neurons across regions, Extended Data Fig. 9b,c*). There was no significant generalization in STR (although stimulus color category could be decoded, Extended Data Fig. 5d).

Discussion:

Representations in prefrontal cortex generalized across tasks (Fig. 2)^{43,53}, although other regions, such as the hippocampus, are likely involved^{39,54}. Less generalization in parietal cortex, visual cortex, and striatum suggests neural representations are more task-unique in these regions. Differences between regions may allow the brain to use the complementary advantages of both sharing representations (generalizing learning) and task-unique representations (avoiding interference)^{26,55}.

Altogether, our results suggest prefrontal cortex may play a unique role in combining shared sensory and motor representations within the same network. This could support the flexible re-combination of representations between tasks. We thank the reviewer for their comment as we believe these changes have better clarified the insights from our study.

Comment 2.2: The second main finding is that LPFC compresses the irrelevant stimulus information, in a task-belief-dependent fashion. This is a different result than Mante 2013, but has now been described as part of a continuum of possible solutions by Pagan 2022 (bioRxiv). Preprints aren't papers, and I don't mean to imply that the existence of that preprint scoops this paper.

We thank the reviewer for noting the Pagan et al (2024) manuscript (now published). We agree with the reviewer that Pagan et al, 2024 is a very interesting manuscript as it builds on the foundational work from Mante et al, 2013 to show different individuals can engage in a continuum of solutions when switching attention between two sensory features. We have revised our Discussion to note this as a possible explanation for the variability in mechanism observed across studies (lines 339-346):

Discussion:

Given that the animals were trained for months on the tasks, one might expect the brain to form independent task representations in order to reduce interference^{4,51,56}. Instead, we found the brain reduced interference by engaging task-relevant sensory and motor representations and suppressing task-irrelevant dimensions^{36,42,57}. Importantly, the degree of engagement/suppression depended on the animals' internal belief about the task state (Fig. 5), allowing the brain to flexibly modulate how color and shape category influenced behavior and ensuring the animal performed the appropriate response (e.g., responding on Axis 1, not Axis 2, during C1 and vice versa for C2).

That being said, we feel our manuscript addresses a fundamentally different question than the Pagan et al, 2024 paper. The focus of our manuscript is on how animals flexibly perform and switch between multiple tasks that involve changes in both sensory processing and motor responses. As noted above, we find both sensory and motor subspaces can be flexibly re-used across different tasks such that monkeys can 'build' the C1 task from components of the C2 and S1 tasks. We have revised our Discussion to highlight this (lines 316-323):

Discussion:

Although our study is limited to three tasks, the underlying mechanism has the capability to be highly expressive – flexibly sequencing task components together could implement a wide variety

of behaviors (a form of sequential compositionality⁴). In this framework, the task representation acts as a control input that selects the appropriate representations and computations^{4,5,34,45}. If, as suggested by our results, the brain can reuse representations and computations across tasks, then this could allow one to rapidly adapt to new situations, either by learning the appropriate task representation through reward feedback^{46,47} or by recalling it from long-term memory⁴⁸.

Furthermore, because the tasks were uncued, we could also gain insight into how animals discover the task in effect and then use their belief about the task to engage the appropriate sensory and motor subspaces. Our results suggest task belief may parametrically use compression as a mechanism to modulate sensory representations (Fig. 5). Based on the reviewers' comments, the revised manuscript demonstrates a similar mechanism for the motor response as well (Fig. 5g,h). In this way, we believe our results are more focused on how task belief can modulate stimulus and response representations in order to flexibly switch between tasks. We have revised our Discussion to highlight this (lines 347-359):

Discussion:

One reason we found shared representations (rather than task-unique ones) may have been because our task required the animal to continuously learn. Across trials, the monkeys used feedback to infer the task context and to flexibly map stimuli onto motor responses (Fig. 4-5; a form of 'class-incremental' continual learning¹⁹). As noted above, shared representations facilitate continual learning by allowing knowledge to generalize between tasks⁵⁸. In addition to reducing interference, scaling representations may also facilitate continual learning by constraining learning to those representations that are currently task relevant⁵⁹⁻⁶². Neural learning rules are activity dependent⁶³ and gated by reward⁶⁴ and so suppressing irrelevant representations may limit learning to task-relevant representations (addressing the credit assignment problem⁶⁵). Future work is needed to understand how suppressing irrelevant features supports continual learning and consider alternative mechanisms¹⁹, including replaying experiences across tasks⁶⁶⁻⁶⁸ and recalling context-specific associations from long-term memory⁴⁸.

Comment 2.2: Rather, the frustrating thing about the present work is that despite the creativity in applying linear decoders, ultimately it's still just a lot of linear decoders; there isn't much translation into new scientific thinking. Here, the authors are still talking about one dimension at a time, where there are a long list of ways they could have made the science deeper. There is no examination of the geometry between the various dimensions (e.g., what was the angle between the C1 and C2 color encoding dimensions?); there is no consideration of linear vs. nonlinear encoding; there are no dynamics; there is no link from stimulus-related activity to movement-related activity (as in, e.g., Elsayed 2016); there is no modeling of any kind. Even one or two of these would have made for a much bigger potential scientific impact. But as it is this just isn't a big enough scientific (or technical) advance to be a Nature paper for me. If we didn't have a decade of thinking in this direction the current work might be enough, but the field has advanced in our understanding of task representation beyond the thinking in this paper. It ends up being a lot of work for a modest amount of new science.

We thank the reviewer for their suggestions for further analyses. At the risk of belaboring our point, we view our manuscript as addressing a different question from Pagan et al (2024) and Mante et al (2013) and therefore have taken a different analysis approach that allows us to focus on those questions. That being said, we appreciate the reviewer's suggestions for deeper insights. Here, we discuss each in turn.

The geometry of the neural representations: We agree this is an interesting question and have performed two analyses to provide further insight. First, if representations are similar between tasks, then one would expect the associated classifiers to be more aligned than chance. To test this, we measured the angle between the hyperplanes of classifiers trained on the C1 and C2 tasks. Consistent with our observation that the representation was shared, the angle between the classifiers decreased after stimulus onset (Extended Data Fig. 12a,b reproduced below). As the exact angle can be affected by noise, we estimated the noise floor by calculating the angle between color classifiers on separate blocks of the C1 task. As seen in Extended Data Fig. 12a, this angle also decreased immediately after stimulus onset. This decrease is similar to what is observed between tasks (Extended Data Fig. 12a,b), with the important

Extended Data Figure 12| Color classifiers were aligned in the C1 and C2 tasks. **a**, Angle between classifier hyperplanes trained on C1 and C2 tasks (blue). For comparison, we measured the angle between classifiers trained on split halves of the C1 task. **b**, Angle between classifiers hyperplanes quantified during sensory response period (shaded region in panel a, 100ms - 300ms after stimulus onset).

caveat that within task cross-block classification is also measure classification of motor responses and so the color representation is likely captured in the neural response immediately following the sample onset.

Second, we were interested in understanding how the subspaces for color, shape and response direction related to one another and how these relationships changed as the animal discovered which task was in effect. To study this, we used Targeted Dimensionality Reduction (TDR, Mante et al, 2013) to identify subspaces of neural activity representing each task variable. We fit a regression model with separate factors for color morph level, shape morph level and response direction for each task (see methods). We then used QR decomposition to define the orthogonal axis of color and response direction i.e. $\beta_{Color,Task i}^\perp$ and $\beta_{Response dir.,Task i}^\perp$ where i is C1, C2 or S1. The angle between the color representation in each pair of tasks (C1-C2, C1-S1, C2-S1) was measured as:

$$\cos(\theta) = \frac{\beta_{color,Task i}^\perp \cdot \beta_{color,Task j}^\perp}{\|\beta_{color,Task i}^\perp\| \|\beta_{color,Task j}^\perp\|}$$

where i and j are task identity of the two tasks. To see how the geometry changed with learning, we quantified the angle using the first 50 trials and last 50 trials in each task block.

Extended Data Figure 12c (reproduced below) shows the angle between the color representation during the C1 task and the color representation during the C2 task. Initially, at the beginning of the block (after switch), this angle was not significantly different from chance ($\theta(C1-C2 \text{ after switch}) = 82.27^\circ$, $p=0.234$, permutation test). However, at the end of the block, the angle between the color representations for the C1 and C2 tasks was significantly lower than chance ($\theta(C1-C2 \text{ before switch}) = 71.23^\circ$, $p=0.003$, permutation test). This suggests the color representation during the C1 task became aligned with the color representation in the C2 task as the animals discovered the C1 task (consistent with classifier results in Fig. 4f). In contrast, the angle between the color representations in the S1 task and the color

Extended Data Figure 12| c, Angle between the axis encoding the color category in pairs of tasks as estimated with targeted dimensionality reduction (TDR; see Methods for details). Error bars show mean \pm s.e.m. . Grey error bars show shuffle distribution. $p \leq 0.05$, 0.01, and 0.001 for *, **, and ***, permutation test, respectively. **d**, same as **c** but for response direction.

representation in the C1 and C2 tasks were not significantly different from chance in both the early and late part of the block ($\theta(\text{S1-C1 after switch})=83.75^\circ$, $p=0.101$, $\theta(\text{S1-C1 before switch})=86.067^\circ$, $p=0.588$, $\theta(\text{S1-C2 after switch})=85.34^\circ$, $p=0.358$, $\theta(\text{S1-C2 before switch})=86.106^\circ$, $p=0.780$, all permutation test). This suggests color representations are less shared between the S1 task and C1/C2 tasks, relative to the C1/C2 tasks, although this may also be due to a reduced color response during S1. To estimate the noise floor, we computed the angle between color subspace within repetitions of C1 and C2 tasks using all trials during the block (blue and green dotted lines, $\theta(\text{within C1})=29.78^\circ$, $p=0.001$, $\theta(\text{within C2})=19.43^\circ$, $p=0.001$, both permutation tests).

Similarly, we measured the angle between response location subspaces across task pairs (Extended Data Fig. 12d, reproduced above). Consistent with a shared response direction representation between the C1 and S1 tasks, the angle between response location subspaces was significantly lower than chance both immediately after a change in task (from switch) and at the end of the block (after switch; $\theta(\text{S1-C1 after switch})=40.92^\circ$, $p=0.001$; $\theta(\text{S1-C1 before switch})=33.35^\circ$, $p=0.001$, both permutation test). In contrast, the angle between the response subspace in the C2 task and the response subspaces of the S1 and C1 tasks was not significantly different from chance ($\theta(\text{S1-C2 after switch})=86.44^\circ$, $p=0.612$, $\theta(\text{S1-C2 before switch})=86.023^\circ$, $p=0.715$, $\theta(\text{C1-C2 after switch})=86.28^\circ$, $p=0.594$, $\theta(\text{C1-C2 before switch})=83.1203^\circ$, $p=0.363$, all permutation test). This is consistent with orthogonal representation of axes of response in C2 task and C1/S1 tasks. To estimate the noise floor, we computed the angle between response direction subspace within repetitions of C1 and C2 tasks using all trials during the block (blue and green dotted lines, $\theta(\text{within C1})=12.05^\circ$, $p=0.001$, $\theta(\text{within C2})=16.76^\circ$, $p=0.001$, both permutation test).

We thank the reviewer for this suggestion as we think it has deepened our understanding of the geometry of neural representations during the tasks. These results are included in the main text (lines 230-232 and 260-261) and in the methods (lines 1269-1289 and 1364-1401).

Results:

Consistent with this, during task discovery, the strength of internal task representation in LPFC during fixation was correlated with the strength of shared color category representation after the onset of the stimulus (Fig. 4h, see Methods for details). Furthermore, the color category subspace in C1 became aligned with the color category subspace in C2 as the animal discovered that the C1 task was in effect (Extended Data Fig. 12a-d).

Results:

In contrast to shared color and shape subspaces, the animals' motor response was stably decoded in a shared subspace (Fig. 4k-l, see Methods for details, $\Delta=3\%$, $p=0.256$, and $\Delta=3\%$, $p=0.680$, for S1-C2-C1 and C1-C2-C1 sequences, respectively; permutation test; see Methods for details and Extended Data Fig. 12b for angle between response direction subspaces).

The dynamics of neural representations: To study how the neural population in each area encoded task variables over time, we used TDR to identify the low-dimensional subspaces of neural activity that captured across-trial variance related to 1) the color morph level of the stimulus, 2) the shape morph level of the stimulus, 3) the response direction on Axis 1, and 4) the response direction on Axis 2. Projecting the population response onto these axes provided a de-mixed estimate of the population dynamics along these task variables.

Overall, our results were consistent with what was observed with linear classifiers (Fig. 3c), suggesting our observations are robust to the exact analytical technique used. However, one advantage of the TDR approach is that it facilitates quantifying the dynamics of task variables by projecting the time course of neural activity in the subspace. As seen in Extended Data Figure 8 (reproduced below), the population response in LPFC first evolved along the axis corresponding to the task's relevant feature before transitioning to move along the axis associated with the corresponding response. For example, in both task C1 and C2, the neural trajectory initially moved horizontally along the axis encoding color with a

larger deflection for prototype morph levels (Extended Data Fig. 8e,g, left, for C1 task and Extended Data Fig. 8e,g, middle, for C2 task, red color morphs versus green color morphs). However, depending on the task, neural activity would then evolve either along the axis encoding responses on Axis 2 or responses on Axis 1 (for the C2 and C1 tasks, respectively). Since color was the irrelevant feature for task S1, the neural activity did not move much along the color axis (Extended Data Fig. 8e,g, right). Instead, the neural activity evolved along axes encoding the shape category of the stimulus and then encoding the response on Axis 1 (Extended Data Fig. 8f, and not on the axis encoding responses on Axis 2, Extended Data Fig. 8h). This highlights the dynamic change in the neural responses during the trial, shifting from sensory encoding to motor responses, and how these dynamics are modulated by the current task. These results are now included in the manuscript (lines 173-176) and in the methods (lines 1290-1401):

Results:

To visualize the dynamics of this transformation, we used Targeted Dimensionality Reduction (TDR)²⁴ to project neural activity onto dimensions encoding the color category (red vs. green) and the motor response along both Axis 1 and Axis 2 (Fig. 3f and Extended Data Fig. 8e-h).

Extended Data Figure 8 | e, The time course of LPFC neural activity projected on axis encoding color category and axis encoding response on Axis 1 for C1 (left), C2 (middle) and S1 (right) tasks. Line colors match the actual color of the stimuli on the color wheel. Neural activity is aligned to saccade onset time. There is no movement along the axis encoding the response on Axis 1 for S1 because there are equal number of stimuli for each color morph level that are projected in either direction in Axis 1. **f,** As in panel e, but for LPFC neural activity during S1 task, projected on axis encoding shape and axis encoding response on Axis 1 **g,** As in panel e, but for LPFC neural activity projected on axis encoding color and axis encoding response on Axis 2. **h,** As in panel e, but for LPFC neural activity in task S1 projected on axis encoding shape and axis encoding response on Axis 2.

Linking stimulus-related activity to movement-related activity. We agree that this is a very interesting question. Indeed, this question initially motivated us to study the correlation between the representations in the sensory subspace and the representations in the motor subspace during the C1 task. As seen in Figure 3, the strength of the stimulus representation is correlated with the future motor response in the Axis 1 subspace, but not the Axis 2 subspace (and vice versa for the C2 task, as seen in Extended Data Fig. 8b-d). We apologize for not making this clearer in the manuscript. We have revised the main text (lines 153-155) and the Discussion (lines 307-315) to make these points more clear:

Results:

To directly test whether this reflected the transformation of information between subspaces, we tested whether information about the stimulus color in the shared color subspace predicted the response in the shared response subspace on a trial-by-trial basis (Fig. 3b).

Discussion:

Consistent with computational models^{4,5,7,41}, these subspaces were shared across multiple tasks (Fig. 2), suggesting they act as ‘task components’^{24,42,43}. Subspaces were sequentially engaged, such that information from the relevant sensory subspace was transformed into the appropriate motor response subspace (Fig. 3)⁴⁴. In this way, a task can be constructed by sequencing together a series of task components. For example, performing the C1 task engaged the subspace representing the color category of the stimulus (shared with the C2 task) and then transformed this information into the motor subspace encoding response along Axis 1 (shared with the S1 task).

This transformation is further emphasized by the TDR analysis. As noted above and seen in the new Figure 3f, TDR shows the task-dependent transformation of stimulus-related activity into movement-related activity. Together, these results highlight how the same sensory input can lead to different behavioral outcomes depending on the task.

We have now included these results in main figures (Fig. 3f), in the main text (lines 173-179) and in the methods (lines 1290-1401).

Results:

To visualize the dynamics of this transformation, we used Targeted Dimensionality Reduction (TDR)²⁴ to project neural activity onto dimensions encoding the color category (red vs. green) and the motor response along both Axis 1 and Axis 2 (Fig. 3f and Extended Data Fig. 8e-h). Consistent with the results from the classifiers, neural activity during the C1 and C2 tasks initially evolved along the color axis according to the stimulus’ category before transforming onto the Axis 1 or Axis 2 dimensions for the C1 and C2 tasks, respectively.

Fig. 3| f, Dynamics of LPFC population activity projected on dimensions of neural activity encoding color category, response on Axis 1, and response on Axis 2 for C1 (dashed line) and C2 (solid line) tasks. Line colors match the color of the stimuli. Time stamps denote time from saccade start.

Motivated by the reviewers’ comments, we were curious as to what mechanism might facilitate task-specific transformations. One hypothesis is that task dependent transformations are facilitated by suppressing task-irrelevant stimulus dimensions *and* task-irrelevant motor responses. This would be a similar mechanism to the relative scaling of color and shape representations during C1/S1 task blocks. To test whether we observe modulation of the motor axis representation, we examined the weights of the

axis classifier to identify LPFC neurons that encoded the response axis. We then categorized the neurons into three groups based on their classifier weights during the response period (200ms to 450ms after stimulus onset). Neurons with significant classifier weights for Axis 1 or Axis 2 were classified as Axis 1 or Axis 2 selective neurons, respectively (green and orange dots in Extended Data Fig. 14a). Neurons without significant classifier weights were classified as non-axis selective (grey dots in Extended Data Fig. 14a).

Extended Data Figure 14| Neurons encoding response on Axis 1 and Axis 2 are suppressed when irrelevant to the task.

a, Classifier weights for classifying axis of response for Fig. 5g. Classifier was trained to decode response axis using C1 and C2 task trials. Neurons with significant weights for Axis 1 and Axis 2 are denoted with green and orange dots, respectively. Non-selective neurons are denoted with gray dots. **b**, Follows Fig. 5g, time course of average firing rate for non-selective neurons in LPFC. Inset shows p-value of difference in firing rate when animal responded on Axis 1 compared to responding on Axis 2 (paired t-test). Lines and shading show mean \pm s.e.m. **c**, Follows Fig. 5g, time course of average firing rate of neurons when the animal responded on Axis 1 (dashed) or Axis 2 (solid). Neurons were grouped by their preferred response on Axis 1 (left) or Axis 2(right). Shaded blue box indicates the time period used to define the neurons preferred response axis (200ms-450ms after stimulus onset). **d**, Follows Fig. 5g. Average firing rate of LPFC neurons during trials when the animal responded on Axis 1 (green) and Axis 2 (pink). Three panels show responses of three categories of neurons: those that responded most strongly on Axis 1 (left), Axis 2 (middle), and were not selective (right) in LPFC. To characterize axis selectivity, classifier was trained to decode response axis using trials from S1 and C2 trials and then firing rate was estimated on withheld C1 trials. Inset shows p-value for of difference between firing rate in response on Axis 1 and response on Axis 2 trials in time periods -600ms-0ms (lower bracket) and -600ms-600ms (upper bracket) after stimulus onset (paired t-test). **e**, As in panel d, but axis selectivity was defined by training a classifier to decode response axis using trials from C1 and C2 trials and then firing rate was estimated on withheld S1 trials. **f-i**, As in Fig. 5g, but shown for **f**, FEF; **g**, aIT; **h**, PAR and **i**, Striatum. Similar to LPFC, Axis 1 and Axis 2 selective neurons in FEF, aIT and PAR were suppressed during response on opposite axis. Lines and shading show mean \pm s.e.m. firing rate across individual neurons for each region. Inset shows p-value for of difference between firing rate in response on Axis 1 and response on Axis 2 trials in time periods -600ms-0ms (lower bracket) and -600ms-600ms (upper bracket) after stimulus onset (paired t-test). **j**, As in Fig. 5h, but shows classifier accuracy in decoding the response axis for all brain regions. Darker to lighter color show progression in trial blocks of 10 shifted by 3 trials from switch trial in C1 task. Lines show mean classification accuracy after stimulus onset for 250 iterations of classifiers. Arrows indicate the time course of example trials. **k**, Classifier accuracy in decoding the response axis for all brain regions. Decoding using neural activity in -400ms to 0ms before stimulus onset period. Lines and shading show mean \pm s.e.m. classification accuracy across 250 iterations of classifiers.

Figure 5| g, Average firing rate (FR) for Axis 1 selective (left) and Axis 2 selective (right) neurons in LPFC. C1 task trials were divided into two groups: response on Axis 1 (correct axis, green) and response on Axis 2 (incorrect axis, purple). Axis 1 selective neurons had higher activity before stimulus onset when the animal responded on Axis 1 (and vice versa when responding on Axis 2). Inset shows p-value of difference over entire time period (paired t-test)

Although neurons were defined based on their activity during the motor response, they were still modulated during the baseline time period (Fig. 5g, reproduced above). Axis 1 selective neurons significantly increased their firing rate during Axis 1 trials and were suppressed during Axis 2 trials, even before stimulus onset (Fig. 5g, left). This pattern was reversed for Axis 2 selective neurons (Fig. 5g, right). Non-axis selective neurons showed no significant difference in firing rates during Axis 1 and Axis 2 trials (Extended Data Fig. 14b). Similar patterns were observed when we performed further controls (Extended Data Fig. 14c-e) and in other regions (Extended Data Fig. 14f-i). These results suggest the brain may actively suppress the irrelevant motor response, facilitating the task-relevant transformation (and inhibiting the task-irrelevant transformation).

Extended Data Figure 2| g, proportion of responses on incorrect axis of response for C1, C2 and S1 tasks.

Next, we were curious as to how quickly the response axis was suppressed after a switch. In other words, how quickly sensory information was routed to the correct motor response. As shown in Extended Data

h Response axis representation is quickly updated after switch

Fig. 5| h, Classifier accuracy for response axis decoding in LPFC. Darker to lighter color show progression in trial blocks of 10 shifted by 3 trials from switch trial in C1 task in S1-C2-C1 task transition. Lines show mean classification accuracy after stimulus onset for 250 iterations of classifiers. Horizontal bar indicates significant increase in classifier accuracy from trial 10 to trial 16 (permutation test with cluster mass correction for multiple comparisons across time).

Figure 2g (reproduced above), the animals make few responses on the incorrect axis, suggesting they quickly infer the change in the response axis.

This rapid and reliable switch was reflected in the neural activity. To study this, we trained a

classifier to decode the axis of response using neural activity during the fixation period (i.e., -400ms-0ms before stimulus onset). Training was restricted to the last 75 trials of the C2 and C1 tasks, when behavioral performance was high. We then applied the classifier to the beginning of the C1 task block to assess the representation of the response axis immediately after the switch. Consistent with behavior, the response axis was decodable immediately after the switch in LPFC (Fig. 5h, reproduced above; behavior is shown in Extended Data Fig. 2g, above). However, there was a slight, but significant, increase in classifier performance within 3-6 trials, indicating that the belief about the response axis is quickly updated ($\Delta(\text{performance in trial 10 and trial 16 in LPFC})=6.88\%$, $p=0.008$, permutation test). As shown in Extended Data Figure 14 j,k (reproduced below), the response axis was decodable in FEF and Striatum but it did not increase as the animal discovered the C1 task (and the response axis was not decodable in PAR and aIT during the fixation period, $\Delta(\text{performance in trial 10 and trial 73}):$ Striatum=-16.67%, $p=0.9750$, aIT=-12.17%, $p=0.9670$, FEF=2.76%, $p=0.2967$, PAR=4.00%, $p=0.2098$, all one tailed permutation test).

j Response axis representation is quickly updated after switch in LPFC

Extended Data Figure 14| j, As in Fig. 5h, but shows classifier accuracy in decoding the response axis for all brain regions. Darker to lighter color show progression in trial blocks of 10 shifted by 3 trials from switch trial in C1 task. Lines show mean classification accuracy after stimulus onset for 250 iterations of classifiers. Arrows indicate the time course of example trials. **k**, Classifier accuracy in decoding the response axis for all brain regions. Decoding using neural activity in -400ms to 0ms before stimulus onset period: Lines and shading show mean \pm s.e.m. classification accuracy across 250 iterations of classifiers.

Altogether, these results suggest the brain supports task-specific transformations by scaling both sensory and motor representations. This would allow the task-relevant sensory to motor transformation to occur, while inhibiting the task-irrelevant transformation. Suppression occurred quickly after a switch in tasks, consistent with the animals' behavior. The suppression was seen throughout the entire trial, even before

the stimulus onset, suggesting the brain is preparing certain eye movements and preventing others in order to ensure they perform the task correctly.

The details of the approach are included in the Methods (lines 1220-1268) and the results are included in the main text (lines 286-299) and Figure 5g-h and Extended Data Fig. 14:

Results:

In addition to selecting sensory information, the three tasks also require selecting the appropriate motor response: the animal must suppress responses on Axis 1 during the C2 task and Axis 2 during the C1 task. Consistent with scaling acting as a general mechanism for modulating task-(ir)relevant information, neurons selective for each motor axis were suppressed when the animal performed a task requiring a response on the other axis (Fig. 5g, see Extended Data Fig. 14c-e for further controls and Extended Data Fig. 14f-i for other regions). However, in contrast to the gradual suppression of sensory representations during task discovery, the incorrect response axis representation in LPFC was quickly suppressed after a change in task (within 3-6 trials, Fig. 5h, Δ (classifier performance in trial 10 and trial 16)=6.88%, $p=0.008$, permutation test, see Extended Data Fig. 14a,k for other regions). This reflected the rapid inference of response axis seen in the animals' behavior (Extended Data Fig. 2g,h).

Results:

Together, these results suggest the monkeys' belief about the current task dynamically adjusted the magnitude of sensory and motor representations³⁶⁻³⁸. Scaling neural representations was a common mechanism used in all the three tasks and could facilitate task-relevant associations between stimulus category and motor response, while preventing task-irrelevant associations.

And in the Discussion (lines 339-346):

Discussion:

Given that the animals were trained for months on the tasks, one might expect the brain to form independent task representations in order to reduce interference^{4,51,56}. Instead, we found the brain reduced interference by engaging task-relevant sensory and motor representations and suppressing task-irrelevant dimensions^{36,42,57}. Importantly, the degree of engagement/suppression depended on the animals' internal belief about the task state (Fig. 5), allowing the brain to flexibly modulate how color and shape category influenced behavior and ensuring the animal performed the appropriate response (e.g., responding on Axis 1, not Axis 2, during C1 and vice versa for C2).

We also highlighted in the Discussion how suppression could mitigate interference:

Discussion (lines 339-342):

Given that the animals were trained for months on the tasks, one might expect the brain to form independent task representations in order to reduce interference^{4,51,56}. Instead, we found the brain reduced interference by engaging task-relevant sensory and motor representations and suppressing task-irrelevant dimensions^{36,42,57}.

And may support continual learning:

Discussion (lines 347-359):

One reason we found shared representations (rather than task-unique ones) may have been because our task required the animal to continuously learn. Across trials, the monkeys used feedback to infer the task context and to flexibly map stimuli onto motor responses (Fig. 4-5; a form of 'class-incremental' continual learning¹⁹). As noted above, shared representations facilitate

continual learning by allowing knowledge to generalize between tasks⁵⁸. In addition to reducing interference, scaling representations may also facilitate continual learning by constraining learning to those representations that are currently task relevant⁵⁹⁻⁶². Neural learning rules are activity dependent⁶³ and gated by reward⁶⁴ and so suppressing irrelevant representations may limit learning to task-relevant representations (addressing the credit assignment problem⁶⁵). Future work is needed to understand how suppressing irrelevant features supports continual learning and consider alternative mechanisms¹⁹, including replaying experiences across tasks⁶⁶⁻⁶⁸ and recalling context-specific associations from long-term memory⁴⁸.

We thank the reviewer for suggesting these analyses as we think they have added new insight into the neural mechanisms of task-switching.

Linear and non-linear encoding: Again, we agree this is an interesting question. In our analyses we use a linear classifier because 1) it makes the fewest assumptions about the underlying representation and 2) it is easier to interpret. Indeed, this allows us to investigate the geometry of the representations, as described above. Nevertheless, we agree with the reviewer that representations could be non-linear. To test whether our results were robust to the classifier type used, we trained non-linear classifier with radial basis function kernel. As shown in Extended Data Figure 7a (reproduced below), non-linear representations of the color category also generalized between the C1 and C2 tasks. Furthermore, the linear and non-linear classifiers performed similarly. These results are now included in the manuscript (lines 123-124):

Results:

Consistent with a shared representation in LPFC, a classifier trained to decode the color category of a stimulus during the C2 task was able to significantly decode the stimulus' color category during the C1 task (Fig. 2e,f, 65ms after stimulus onset in LPFC, $p < 0.001$, permutation test; *similar results were seen when using a non-linear classifier, Extended Data Fig. 7a*).

a Linear and non-linear classifiers decoded color category in a generalized manner

Extended Data Figure 7 | Color category was cross decoded between tasks in LPFC. a, Accuracy of classifier trained to decode color category from LPFC neural activity when trained on C2 task and tested C1 task (left); trained on C1 task and tested on C2 task (middle); and trained on C1 task and tested on C1 task (right). Top row shows accuracy when using a linear kernel for classifier and bottom row shows accuracy when using radial basis function (RBF) kernel for classifier. Lines and shading show mean \pm s.e.m. over time. Distribution reflects 250 resampled classifiers (see Methods for details). Horizontal bars (top right of each plot) indicate above-chance classification ($p \leq 0.05$, 0.01, and 0.001 for thin, medium, and thick lines, respectively; permutation test with cluster mass correction for multiple comparisons).

Computational modeling. We agree with the reviewer that there will be broad interest in building more in-depth computational models of our results. As noted above, we have already begun doing so. However, we feel these models are out of scope for the current paper which is more focused on the behavior, electrophysiology and neural dynamics. We are happy to make our data openly available to others so that future work can further study the computational mechanisms supporting flexible behavior. We have updated the Discussion to propose this as potential next steps (lines 331-332):

Discussion:

*In contrast, shared subspaces increase interference but could speed learning by allowing knowledge to generalize across tasks⁶. For example, learning the C2 task shapes the neural computations needed to categorize the color of a stimulus, which could generalize to other tasks that involve categorizing color, such as C1. **Future computational and experimental work is needed to test whether having such foundational knowledge can facilitate learning new tasks.***

Discussion (lines 356-359):

Future work is needed to understand how suppressing irrelevant features supports continual learning and consider alternative mechanisms¹⁹, including replaying experiences across tasks⁶⁶⁻⁶⁸ and recalling context-specific associations from long-term memory⁴⁸.

Minor comments

Comment 2.3: What potential influences could the training procedure / task training order have had on the learned representation? This has been shown to matter, and should be discussed.

We thank the reviewer for raising this important point. Both animals were trained in the same order of tasks (S1, C2, and then C1). We now clarify this in the Methods (lines 673-674):

Methods

Both animals were trained in the same order of tasks: S1, C2, and then C1.

We agree with the reviewer that this could have an impact on our results. In particular, as noted by Reviewer 1, because compositional representations are thought to be useful when learning tasks, our training regime may have contributed to our observation of shared representations. Future work is needed to understand how training curriculum affects the types of representations in biological networks. We now note this in the Discussion (lines 331-332):

Discussion:

*In contrast, shared subspaces increase interference but could speed learning by allowing knowledge to generalize across tasks⁶. For example, learning the C2 task shapes the neural computations needed to categorize the color of a stimulus, which could generalize to other tasks that involve categorizing color, such as C1. **Future computational and experimental work is needed to test whether having such foundational knowledge can facilitate learning new tasks.***

Comment 2.4: Figure 4g: the color categorization classifier's dependence on sequence doesn't seem consistent with the task classifier results. The explanation (L200-208) doesn't clearly explain it.

We thank the reviewer for raising this important point. As noted by the reviewer, the sequence of tasks influences the animals' behavior – they perform better during a C1-C2-C1 sequence than a S1-C2-C1 sequence (Fig. 4a). Consistent with this, the animal's belief that it is performing the C1 task is greater in the C1-C2-C1 sequence compared to the S1-C2-C1 sequence (Fig. 4d). Furthermore, task belief is correlated with behavior in S1-C2-C1 blocks (Fig. 4e).

As detailed in the manuscript, one might expect a similar pattern for the color category classifier – as the animal's belief they are performing the C1 task increases, they should increasingly engage the color category classifier. This seems to be the case for the S1-C2-C1 task sequence (Fig. 4g, grey line; correlation in Fig. 4h). However, there is not a strong and significant correlation between behavioral performance and color category information during the C1-C2-C1 block sequences (Fig. 4g, pink; $r = -0.543$, $p = 0.207$, permutation test). Again, we thank the reviewer for noting this, as it motivated several new analyses that led to new insights.

As the reviewer points out, on C1-C2-C1 blocks, behavioral performance starts high and only improves slightly during the block (68% on trial 15 to 78% on trial 75 in C1-C2-C1 blocks, compared to 58% on trial 15 to 76% on trial 75 in S1-C2-C1 blocks). Given this, there may not need to be a change in the strength of color representation during the block. Indeed, the competition from the shape category is low throughout C1-C2-C1 blocks, reflected in the fact that shape information is at or below chance levels (Fig. 4j). In other words, performance may already be 'optimized' and therefore a change may not be needed. If true, one might expect LPFC to become less engaged as the task becomes more certain. Previous work has suggested that LPFC is most strongly engaged when cognitive control is needed – that is, when the task is uncertain³⁴. Therefore, the decrease in task representations could reflect the engagement of other neural circuits representing color during the C1-C2-C1 sequence in comparison to the S1-C2-C1 sequence. To test this, we computed the average accuracy of color category classifier for C1-C2-C1 and S1-C2-C1 sequences for all regions. As seen in Extended Data Figure 11e (reproduced below), color category information in FEF was greater during the C1-C2-C1 sequence compared to S1-C2-C1 sequence ($\Delta(\text{average accuracy C1-C2-C1 and S1-C2-C1 sequences}) = 12.7\%$, $p = 0.004$). This suggests there may be a consolidation of color information into FEF during the C1-C2-C1 sequences.

Extended Data Figure 11| e, Comparison of progression of color category classifier accuracy in C1-C2-C1 task sequence (purple) and S1-C2-C1 task sequence (black) for FEF. Average accuracy was computed using classifier performance in 200ms to 400ms after stimulus onset period. Lines and shading show mean \pm s.e.m. classification accuracy after stimulus onset. Distribution reflects 250 iterations of classifiers. $\Delta(\text{average accuracy S1-C2-C1 and C1-C2-C1 sequences}) = 12.7\%$, $p = 0.004$. On C1-C2-C1 blocks, behavioral performance starts high and only improves slightly during the block (68% on trial 15 to 78% on trial 75 in C1-C2-C1 blocks, compared to 58% on trial 15 to 76% on trial 75 in S1-C2-C1 blocks). Given this, there may not need to be a change in the strength of color representation during the block. Indeed, the competition from the shape category is low throughout C1-C2-C1 blocks, reflected in the fact that shape information is at or below chance levels (Fig. 4j). In other words, performance may already be 'optimized' and therefore a change may not be needed. If true, one might expect LPFC to become less engaged as the task becomes more certain. Previous work has suggested that LPFC is most strongly engaged when cognitive control is needed – that is, when the task is uncertain⁷⁰. Therefore, the decrease in task representations could reflect the engagement of other neural circuits, such as FEF, as shown here, to represent the color category during the C1-C2-C1 sequence in comparison to the S1-C2-C1 sequence.

Again, we thank the reviewer for this comment as we think it has provided deeper insight into the mechanisms allowing the animal to infer, and perform, the current task. These results are now included in the manuscript (lines 236-242):

Results:

Like task representations, color representations depended on the task sequence (Fig. 4g). While shared representations increased when discovering C1 following a S1-C2-C1 sequence, they remained stable during C1-C2-C1 sequences (63% to 62%, $\Delta = -1\%$ from trial 40 to 110, $p = 0.808$,

permutation test). While this is consistent with better performance at the beginning of the C1 task during C1-C2-C1 sequences, it doesn't explain the improvement in behavior over trials during the C1-C2-C1 sequence (Fig. 4a). This difference in performance was associated with an increase in color information in FEF during C1-C2-C1 sequence (Extended Data Fig. 11e, difference in accuracy between C1-C2-C1 and S1-C2-C1 was 12.7%, $p=0.004$), possibly reflecting the consolidation of color category information from LPFC into FEF as the animal became more certain about the current task.

Motivated by the reviewers' comments, we were also interested in understanding how task representations persisted from the C1/S1 task, through the C2 task, to influence the next C1/S1 task. To study this, we tested whether we could decode the *previous* C1/S1 task during the C2 block and into the next C1/S1 block.

To this end, we trained a classifier to distinguish between the C1 and S1 tasks using neural activity from LPFC during the fixation period (-400ms to 0ms before stimulus onset). To ensure the animal understood the task in effect, we limited training to the last 50 trials of the C1 and S1 task blocks. We then tested this classifier on withheld trials from three different time periods (see Fig. 4b, left; reproduced below for a schematic).

First, we tested it on withheld trials from the last 50 trials of the C1 and S1 tasks. The classifier significantly decoded the identity of the current task (Fig. 4b (1), right, 244 neurons, 74%, $p=0.001$, permutation test).

Second, we tested whether the same classifier could decode the identity of the *previous* C1/S1 task during the C2 task (i.e., differentiating between S1-C2 and C1-C2 blocks). Although the classifier's performance was lower compared to decoding during the C1/S1 tasks, the classifier significantly decoded the identity of the previous block during the C2 trials (Fig. 4b (2), right, 62%, $p=0.001$, permutation test).

Fig. 4| b, left, Schematic of task classifier trained to decode the identity of the C1/S1 task on the 75 trials before task switch and tested on 1) withheld trials before switch, 2) C2 trials to decode the identity of the previous block, and 3) trials after switch to C1/S1 to decode the identity of the previous block on the Axis 1 (S1 vs. C1). right, Colored circles indicate accuracy of classifier. Grey violin plot is null distribution.

Extended Data Figure 11| d, Follows Fig 2b. Classifier accuracy for decoding the identity of C1/S1 tasks in FEF using last 50 trials of the C1/S1 task(left), trials of C2 task (middle), and 50 trials after switch of C1/S1 task (right). Numbers inside colored circles denote corresponding classification task in Fig. 4b. Colored circles denoted observed values and violins show shuffle distribution. $p \leq 0.05$, 0.01 , and 0.001 for *, **, and ***, respectively, permutation test. Similar to LPFC, information about the previous task was present in FEF and throughout both the C2 block and the beginning of the next block. (60 neurons, Train before switch C1/S1, test before switch C1/S1: 64%, $p=0.001$; Train before switch C1/S1, test after switch C2: 60%, $p=0.001$; Train before switch C1/S1, test after switch C1/S1: 54%, $p=0.001$). The effect was trending in aIT (98 neurons, Train before switch C1/S1, test before switch C1/S1: 56%, $p=0.087$; Train before switch C1/S1, test after switch C2: 54%, $p=0.032$; Train before switch C1/S1, test after switch C1/S1: 56%, $p=0.002$). There was no significant information about the previous task in PAR or Striatum, although this likely reflects a limited number of neurons recorded in each region.

Third, we tested whether the representation of the previous C1/S1 task was maintained through the C2 task and into the *next* block of C1/S1 tasks. Using the first 50 trials of the C1/S1 task, we tested the same C1/S1 classifier to decode the identity of the *previous* C1/S1 task. Again, the classifier successfully decoded the identity of the previous C1/S1 block trials (Fig. 4b (3), right, 60%, $p=0.001$, permutation test).

Altogether, these results suggest that information about the previous task is maintained during the C2 task and transferred to the C1/S1 task in the subsequent block. This may explain the bias in behavior observed during different sequences of blocks (Fig. 4a).

Similar to LPFC, information about the previous task was present in FEF and throughout both the C2 block and the beginning of the next block. (Extended Data Fig. 11d, reproduced above; 60 neurons, train on C1/S1, end of block; test on C1/S1, withheld trials from end of block: 64%, $p=0.001$; test on C2: 60%, $p=0.001$; test on C1/S1 beginning of next block: 54%, $p=0.001$, all permutation tests). The effect was trending in aIT (98 neurons, train on C1/S1, end of block; test on C1/S1, withheld trials from end of block: 56%, $p=0.087$; test on C2: 54%, $p=0.032$; test on C1/S1, beginning of next block: 56%, $p=0.002$, all permutation tests). There was no significant information about the previous task in PAR or Striatum, although this likely reflects a limited number of neurons recorded in each region.

We again thank the reviewer for suggesting this very interesting analysis and now include these results as a main figure panel (Fig. 4b), Extended Data Figure 11d and discuss it in the Results (lines 215-220), Discussion (lines 347-359) and Methods (lines 1402-1417):

Results:

So far, our results suggest the monkeys tracked whether the S1 or C1 task was in effect, and that this belief was maintained during the intervening C2 task. Consistent with this, the task classifier trained during the S1 and C1 tasks could decode the identity of the previous task during the subsequent C2 task (Fig. 4b-2, 62%, $p=0.001$, permutation test) and into the start of the next S1/C1 task (Fig 4b-3, 64%, $p=0.001$, permutation test). Similar results were seen in FEF, but not other regions (Extended Data Fig. 11d).

Discussion:

One reason we found shared representations (rather than task-unique ones) may have been because our task required the animal to continuously learn. Across trials, the monkeys used feedback to infer the task context and to flexibly map stimuli onto motor responses (Fig. 4-5; a form of ‘class-incremental’ continual learning¹⁹). As noted above, shared representations facilitate continual learning by allowing knowledge to generalize between tasks⁵⁸. In addition to reducing interference, scaling representations may also facilitate continual learning by constraining learning to those representations that are currently task relevant⁵⁹⁻⁶². Neural learning rules are activity dependent⁶³ and gated by reward⁶⁴ and so suppressing irrelevant representations may limit learning to task-relevant representations (addressing the credit assignment problem⁶⁵). Future work is needed to understand how suppressing irrelevant features supports continual learning and consider alternative mechanisms¹⁹, including replaying experiences across tasks⁶⁶⁻⁶⁸ and recalling context-specific associations from long-term memory⁴⁸.

Comment 2.5: Fig. 5b / L238 – the text says that shape representation was attenuated in the color tasks, but that’s not what the figure shows; really the peak is delayed and drops back down instead of being sustained. This is also interesting, but not consistent with the text.

We thank the reviewer for noting the shift in the time course of information about the shape category. To track how discovering the C1 task changes the peak and timing of the shape representation, we added Extended Data Figure 11h to show the change in the peak of the distance along the shape axis (black line, left axis) and the timing of that peak (blue line, right axis). This figure is reproduced below.

As noted by the reviewer, this shows the shift in the timing of shape information as the animal discovers the C1 task. This shift in timing results in a reduction of information in the window of sensory processing (100ms-300ms; Fig. 5b). As suggested, we have revised the text to note the shift in the time course (lines 272-273 and lines 251-252):

Results:

Furthermore, this attenuation depended on the animals' internal belief about the task: as they discovered the C1 task was in effect, the representation of color category was magnified (Fig. 5a), while shape representation **was attenuated and delayed in time** (Fig. 5b and Extended Data Fig. 11h).

Results:

In contrast to color, shape representation was significant immediately after the switch to the C1 task during S1-C2-C1 sequences (53%, $p=0.02$, permutation test) **and then decreased, and delayed in time**, as the animal learned (Fig. 4i-j, 49%, $\Delta=-4\%$, $p=0.040$, permutation test).

Extended Data Figure 11| h, Time (blue) and amplitude (black) of the maximum distance along the shape encoding axis. The maximum distance was estimated by fitting a Gaussian CDF function to classifier accuracy in each trial window.

Comment 2.6: Figure S2: Why is task information so time-dependent? How should we think about its role in how the whole system works?

This is an interesting observation. As noted by the reviewer, there is a change in the task representation over time – it increases to about 51.5% of the baseline around the time of the stimulus presentation (Extended Data Fig. 3). While this is less than the change in the other task variables (which tend to be ~438% for shape and color category and ~6300% for reward and response location relative to baseline), this is a significant modulation. We now note this in the manuscript (lines 538-539):

Legend of Extended Data Figure 3

Extended Data Figure 3| Task identity is represented throughout the trial. Follows Fig. 1m. Normalized CPD for task identity averaged for all regions. Lines and shading show mean \pm s.e.m. **Task identity representation increased by 51.5% from baseline during the stimulus processing period.**

Such ramping responses are common in many tasks. For example, a ramping response is observed when animals prepare to perform a motor response during a working memory task (Romo et al. Nature 1999). We've seen similar dynamics in task representations as the animal prepared to process and respond to a visual stimulus (Buschman et al, *Neuron* 2012, Figure 2A). Our interpretation of these dynamics is that it reflects a just-in-time increase in task relevant information to facilitate execution of the task. We now note this in the manuscript (lines 538-539):

Legend of Extended Data Figure 3

Previous research has reported similar dynamics in task representations, suggesting that as an animal prepares to process and respond to a visual stimulus, task-relevant information increases to facilitate task execution⁶⁹.

Comment 2.7: Why does the correlation over time in Fig. S9b strongly flip sign?

As noted, there is a significant negative correlation between task belief (measured during fixation) and the encoding of shape category after stimulus onset (Extended Data Fig. 9b). This is also the time period with significant information about the shape category of the stimulus (Fig. 4i, reproduced to the left). However, as noted by the reviewer, fluctuations in shape encoding before the stimulus onset were positively correlated with the increase in the animal's belief during task discovery. Nevertheless, when using a permutation test, this correlation did not reach statistical significance. It is not clear what might drive this variance; it suggests there is a change in the preparatory state of the network over learning that influences the engagement of the shape subspace.

To avoid confusion, we now limit analysis to the time period with significant trend in shape category during task discovery (Fig. 4i, 200ms-300ms). The new Extended Data Figure 11f shows the correlation between task belief and shape category encoding in S1-C2-C1 task sequence.

Fig. 4i

Extended Data Figure 11f

Fig. 4i, left, Schematic of shape category classifier trained to decode shape category on 75 trials before switch in S1 task and tested on sliding window of 50 trials starting from switch into C1 task. right, Darker to lighter color show progression in trial blocks of 50 shifted by 5 trials from switch trial in S1-C2-C1 task transition. Lines show mean classification accuracy after stimulus onset for 250 iterations of classifiers. Horizontal bars indicate above chance trend for classifier accuracy during task discovery ($p \leq 0.05$, 0.01, and 0.001 for thin, medium, and thick lines, respectively; permutation test with cluster mass correction for multiple comparisons across time).

Extended Data Figure 11f, Follows Fig. 4h, correlation between neurally estimated task belief encoding and shape category encoding performance in C1 task for S1-C2-C1 task sequences.

Comment 2.8: Details

Fig. 2f: what is the big red X? Non-significance? Onset time?

Figure 1i: I think the yellow trace (PAR) is missing.

Typo: Figure 4c,d, "identity"

We thank the reviewer for noting these issues and apologize for the confusion.

The red asterisk and red 'X' in Figure 2 were intended to mark the time of onset. We have edited the figures to add labeled lines instead. We hope these are less confusing.

Shape information was not significant in parietal cortex and so we did not include it in Figure 1i. To avoid confusion, we have added the average CPD for PAR neurons but indicate it was non-significant across the population by making it a dashed line. This is clarified in the legend.

We fixed the typo in Figures 4c and 4d. Thank you for catching this!

Referee #3 (Remarks to the Author):

The paper presents a compelling study on the population activity supporting the execution of multiple tasks by non human primates. The three tasks are designed in a way that a pair of tasks share stimulus space, while an other pair of tasks shares the decision space. This task design enables the investigation of whether the shared stimulus features and shared decision space are actually reflected in a shared representational space of the neural population activity. The study reports recording from multiple brain areas, including higher visual area as well as prefrontal regions. The study involves simple population analysis and both statistically well-grounded single cell analysis and properly motivated population level analyses that relies on simple linear decoding. Critically, the complex behavioral paradigm and the elaborate cognitive processes require accurate control over a vast array of confounds, in which the paper excels. The compelling claim of the paper is that it provides evidence that the lpfc hosts separate subspaces for stimuli and decisions, each of which is reused in tasks. Consequently, the lpfc efficiently manages to route information between these subspaces depending on task requirements. Importantly, the tasks are uncued, which requires the animals to discover the current task based on stimulus /decision /reward contingencies. This setting enables the authors to investigate how subjective belief about task shapes the neural representation. The presented analysis provides support for a representation that is highly consistent with theoretical expectations. The paper is very clearly written, presenting the findings in an approachable way.

The paper is very timely since both machine learning, and computational neuroscience identified continual /lifelong learning as a key challenge for learning systems (Hasdell et al, 2020, TICS). The paper also relates to the meta learning literature in machine learning that has attracted attention in the cognitive science community too (Binz et al, 2023, BBS). The key message of the paper, compositional representation has been the focus of a number of recent human behavioral studies and the neural mechanisms have also been the target of a number of studies (Schwartenbeck et al, 2023, Cell; Lake and Baroni, 2023, Nature). The key contribution of the paper is presenting a clever task design that is very challenging to animals (60 months of training) and making a strong case for compositional representation in the brain. Importantly, while a representation is appealing and seems like a 'natural' solution from a theoretical point of view, it is far from the only solution. Recent studies in deep learning have highlighted that there are distinct 'lazy' and 'rich' regimes when training an artificial neural network on multiple tasks, the rich being characterized by a representation similar to the one presented here, and circumstances of training determine which of these prevail (Chizat et al, 2018; Flesch et al, 2022).

The results of the paper gain strong support by careful analyses that look at the problem from different angles, and these analyses build step-by-step a framework that strengthen the interpretation of the authors. One particularly appealing example of this is the analysis of the dependence of performance after task switch on the sequence of earlier blocks: initial uncertainty of the animals about the relevance of the stimulus feature gradually builds up, concurrently with the loss in reliance on the feature that was relevant in a past block.

Despite the above described appeals of the manuscript, several issues require additional care and I also list a number of suggestions.

We thank the reviewer for taking the time to review our manuscript. We are glad the reviewer found it of interest and appreciate their comments.

Comment 3.1: The paper sets off by stating that they test 'this hypothesis' but it is not clearly stated what the actual hypothesis is. The continual learning literature lists quite an array of architectures (Van de Ven et al, 2022, Nat Mach Intell) that might serve computations necessary for task performance, which could be highlighted as alternatives (including the representations associated with the above quoted lazy and rich learning regimes).

We apologize for the lack of clarity. Here, we were referring to the hypothesis that the brain can perform a task by compositionally re-use subspace representations from other tasks. We have revised the text to make this clearer (lines 35-38):

Introduction:

Whether the brain similarly reuses sensory, cognitive, and/or motor representations across tasks remains unknown. Furthermore, we do not yet understand how the brain flexibly engages these representations to continually adapt to the changing demands of the environment¹⁹.

We have also revised the Discussion to make this more clear (lines 324-332):

Discussion:

Our results contrast with previous work that found every task uses a unique representation of sensory and motor information⁴⁹⁻⁵¹. Task-unique representations have the advantage of minimizing interference when learning and performing multiple tasks⁵¹⁻⁵² but have limited ability to generalize learning across tasks^{3,26}. In contrast, shared subspaces increase interference but could speed learning by allowing knowledge to generalize across tasks⁶. For example, learning the C2 task shapes the neural computations needed to categorize the color of a stimulus, which could generalize to other tasks that involve categorizing color, such as C1. Future computational and experimental work is needed to test whether having such foundational knowledge can facilitate learning new tasks.

Motivated by the reviewer's comment, we have rewritten the Discussion to relate our work to the literature on continual learning (lines 347-359):

Discussion:

One reason we found shared representations (rather than task-unique ones) may have been because our task required the animal to continuously learn. Across trials, the monkeys used feedback to infer the task context and to flexibly map stimuli onto motor responses (Fig. 4-5; a form of 'class-incremental' continual learning¹⁹). As noted above, shared representations facilitate continual learning by allowing knowledge to generalize between tasks⁵⁸. In addition to reducing interference, scaling representations may also facilitate continual learning by constraining learning to those representations that are currently task relevant⁵⁹⁻⁶². Neural learning rules are activity dependent⁶³ and gated by reward⁶⁴ and so suppressing irrelevant representations may limit learning to task-relevant representations (addressing the credit assignment problem⁶⁵). Future work is needed to understand how suppressing irrelevant features supports continual learning and consider alternative mechanisms¹⁹, including replaying experiences across tasks⁶⁶⁻⁶⁸ and recalling context-specific associations from long-term memory⁴⁸.

We also thank the reviewer for the suggested references, which we now include in the manuscript as they broaden the scope of our study and further motivate alternative hypotheses.

Comment 3.2: The paper extensively analyses LPFC in the context of compositional generalisation, and

the discussion mentions that generalisation was strongest in PFC (line 277), but I could not identify a compelling comparison of brain regions. Demonstrating specificity of the presented computations would be very informative to establish unique contributions. The five recording locations provide an extremely rich data but comparison of the contributions is limited in the paper.

We thank the reviewer for this suggestion and agree it will provide further insight into the role of each region. As the reviewer suggested, we have now measured the strength of generalization across all five brain regions. As seen in Extended Data Figure 9b,c (reproduced below), compositionality was strongest in LPFC. In contrast, color category information in Striatum, FEF, and PAR did not generalize between tasks or, if they did, the generalization was delayed relative to the task-specific category information (Extended Data Fig. 9b).

To quantify the difference between each region and LPFC, we subsampled the neurons in LPFC to create a population that had an equal number as what was recorded in each other region. While this reduced our statistical power, we consistently observed shared color representation between C1 and C2 tasks in the LPFC (with the exception of when matching to PAR, where neuron counts were too few). In comparison, generalization was reduced in FEF, PAR, STR, and aIT (Extended Data Fig. 9c).

Together, these results suggest non-compositional representations exist in other brain regions, and it suggests prefrontal cortex may play a unique role in compositionally representing sensory and motor representations within the same network. We have revised the manuscript to include these new results in Extended Data Fig. 9 and note the uniqueness of prefrontal cortex in results (lines 132-138) and discussion (lines 333-338):

Extended Data Figure 9 | b, Time course of accuracy of classifier trained to decode color category within C1 task and across C1 and C2 tasks. Classifiers were trained separately for each brain region (Striatum, aIT, FEF, and LIP in each column, moving left to right). Lines and shading show mean \pm s.e.m. classification accuracy after stimulus onset. Distribution reflects 1000 iterations of classifiers. Horizontal bars along top indicate above-chance classification ($p \leq 0.05$, 0.01, and 0.001 for thin, medium, and thick lines, respectively; one-tailed t-test with no correction). **c**, Comparison of strength of shared color category responses in LPFC to other regions. All classifiers were trained to decode color category in C2 task and then tested on C1 task. LPFC neurons were down sampled to match each brain region (110 neurons for Striatum, 195 neurons for aIT, 116 neurons for FEF and 54 neurons for PAR). Accuracy was averaged from 100 ms to 300 ms after stimulus onset. Lines and shading show mean \pm s.e.m. ($p \leq 0.001$ for ***; unpaired t-test).

Results:

While color category information was represented in a shared subspace in LPFC, it was represented in task-specific subspaces in other brain regions (Extended Data Fig. 9a). Generalization was weaker in FEF, PAR and aIT and was delayed with respect to task-specific sensory information (and was delayed relative to LPFC, Fig. 2g, *even when controlling for differences in number of neurons across regions, Extended Data Fig. 9b,c*). There was no significant generalization in STR (although stimulus color category could be decoded, Extended Data Fig. 5d).

Discussion:

Representations in prefrontal cortex generalized across tasks (Fig. 2)^{43,53}, although other regions, such as the hippocampus, are likely involved^{39,54}. Less generalization in parietal cortex, visual cortex, and striatum suggests neural representations are more task-unique in these regions. Differences between regions may allow the brain to use the complementary advantages of both sharing representations (generalizing learning) and task-unique representations (avoiding interference)^{26,55}.

Altogether, our results suggest prefrontal cortex may play a unique role in combining shared sensory and motor representations within the same network. This could support the flexible re-combination of representations between tasks. We thank the reviewer for their comment as we believe these changes have better clarified the insights from our study.

Comment 3.3: According to task design, there is correlation between the shape category and response when the shape is relevant. Is it ensured here that this correlation is not confounding the results shown on Fig 2a (and related figures in the supplementary materials, Fig S2)? If only correct trials are used, as the methods states, then this correlation might introduce confounds. If, however, responses are balanced then decision-related confound is not present anymore, but in that case the error rate will be higher. In this case one can assume that the trials used in the analysis come from the beginning of the block when the animal is still uncertain about the task being performed. In this case, however, relevance of the feature is not determined by the animal. This potential confound is most relevant at later times during the trial and might affect e.g. FEF responses more substantially. Also, this might cause a drop in across-task decoding of colour late in the trial (Fig 2f).

We thank the reviewer for raising this important point. We agree that it is critical to ensure the classifiers are trained (and tested) in a way that avoids potential confounds. As the reviewer notes, the classifier trained in Figure 2a was trained on correct trials only. Therefore, as the reviewer notes, this includes both color representation and motor representations. For this reason, we performed the same analysis, but training a classifier on a balanced set of rewarded and unrewarded trials. We see similar results (Extended Data Fig. 5a, reproduced below), although with reduced statistical power, as predicted and noted by the reviewer.

a Color category decoding in LPFC with balanced response direction trials

Extended Data Figure 5] Color category and shape category were encoded in all regions. a, Follows Fig. 2a. Time course of accuracy of classifier trained to decode the color category of the stimulus during the C1 task based on neural activity in LPFC. To control for movement, the number of trials with each response direction were balanced within each color category. Only trials with responses on the correct response axis were included (e.g., Axis 1 for task C1). Lines and shading show mean \pm s.e.m. classification accuracy after stimulus onset. Distribution reflects 250 iterations of classifiers. Horizontal bars along top indicate above-chance classification ($p \leq 0.05$, 0.01, and 0.001 for thin, medium, and thick lines, respectively, permutation test with cluster mass correction for multiple comparisons).

One further concern might be that error trials were more common during the beginning of the block, when the animal was less sure as to what task they were performing. To ensure that this did not unduly influence our results, we retrained a set of classifiers on trials from the end of the block (last 100 trials of C1 and S1 tasks) when the behavioral performance was above 70%. Importantly, the number of trials with each motor response was balanced within each color category (i.e., a balanced number of correct and incorrect trials). While this analysis restricted the number of trials we could use, and relied on error trials, we found that classifiers could significantly decode color and shape in C1 and S1 tasks respectively (to an admittedly lesser degree). This figure is reproduced below and is now included as Extended Data Figure 5b,c.

We have also revised the manuscript to note this potential concern and the new results (lines 107-109):

Results:

Task irrelevant stimulus information was attenuated. In LPFC, classification accuracy about the shape of the stimulus was reduced during the C1 and C2 tasks (Fig. 2b) and information about the color of the stimulus was reduced during the S1 task (Fig. 2a; this was also true at the end of the block, when the animal was more certain about the identity of the task, Extended Data Fig. 5b,c).

Finally, to ensure that the observed increase in shared color representation engagement when discovering the C1 task (Fig. 4f) was not due to response-related confounds, we balanced trials for their response direction within each trial window.

Minor:

Comment 3.4: The paper argues that a sequence of S1C2C1 has different dynamics than the sequence C1C2C1. For this the animal needs to maintain C1 during C2. A query if this can be identified with similar analysis techniques as the ones used in the paper during C2 would be instructive.

We thank the reviewer for suggesting this very interesting analysis. As noted by the reviewer, in order for the previous Axis 1 task to affect performance, one would expect information to carry across blocks. To test whether the animal maintains a representation of the C1/S1 task during the C2 task and in the subsequent block of C1/S1 tasks, we tested whether we could decode the *previous* task during the C2 block and even into the subsequent C1/S1 block.

To this end, we trained a classifier to distinguish between the C1 and S1 tasks using neural activity from LPFC during the fixation period (-400ms to 0ms before stimulus onset). To ensure the animal understood the task in effect, we limited training to the last 50 trials of the C1 and S1 task blocks. We then tested this classifier on withheld trials from three different time periods (see Fig. 4b, left, reproduced below, for a schematic).

First, we tested it on withheld trials from the last 50 trials of the C1 and S1 tasks. The classifier significantly decoded the identity of the current task (Fig. 4b (1), right, 244 neurons, 74%, $p=0.001$, permutation test).

Second, we tested whether the same classifier could decode the identity of the *previous* C1/S1 task during the C2 task (i.e., differentiating between S1-C2 and C1-C2 blocks). Although the classifier's performance was lower compared to decoding during the C1/S1 tasks, the classifier significantly decoded the identity of the previous block during the C2 trials (Fig. 4b (2), right, 62%, $p=0.001$, permutation test).

Third, we tested whether the representation of the previous C1/S1 task was maintained through the C2 task and into the *next* block of C1/S1 tasks. Using the first 50 trials of the C1/S1 task, we tested the same C1/S1 classifier to decode the identity of the *previous* C1/S1 task. Again, the classifier successfully decoded the identity of the previous C1/S1 block trials (Fig. 4b (3), right, 60%, $p=0.001$, permutation test).

Altogether, these results suggest that information about the previous task is maintained during the C2 task and transferred to the C1/S1 task in the subsequent block. This may explain the bias in behavior observed during different sequences of blocks (Fig. 4a).

Similar to LPFC, information about the previous task was present in FEF and throughout both the C2 block and the beginning of the next block. (Extended Data Fig. 11d, reproduced above, 60 neurons, train on C1/S1, end of block; test on C1/S1, withheld trials from end of block: 64%, $p=0.001$; test on C2: 60%, $p=0.001$; test on C1/S1 beginning of next block: 54%, $p=0.001$, all permutation tests). The effect was trending in aIT (98 neurons, train on C1/S1, end of block; test on C1/S1, withheld trials from end of block: 56%, $p=0.087$; test on C2: 54%, $p=0.032$; test on C1/S1, beginning of next block: 56%, $p=0.002$, all permutation tests). There was no significant information about the previous task in PAR or Striatum, although this likely reflects a limited number of neurons recorded in each region.

Fig. 4b

Extended Data Figure 11d

Fig 4| b, left, Schematic of task classifier trained to decode the identity of the C1/S1 task on the 75 trials before task switch and tested on 1) withheld trials before switch, 2) C2 trials to decode the identity of the previous block, and 3) trials after switch to C1/S1 to decode the identity of the previous block on the Axis 1 (S1 vs. C1). right, Colored circles indicate accuracy of classifier. Grey violin plot is null distribution.

Extended Data Figure 11| d, Follows Fig 2b. Classifier accuracy for decoding the identity of C1/S1 tasks in FEF using last 50 trials of the C1/S1 task(left), trials of C2 task (middle), and 50 trials after switch of C1/S1 task (right). Numbers inside colored circles denote corresponding classification task in Fig. 4b. Colored circles denoted observed values and violins show shuffle distribution. $p \leq 0.05$, 0.01, and 0.001 for *, **, and ***, respectively, permutation test. Similar to LPFC, information about the previous task was present in FEF and throughout both the C2 block and the beginning of the next block. (60 neurons, Train before switch C1/S1, test before switch C1/S1: 64%, $p=0.001$; Train before switch C1/S1, test after switch C2: 60%, $p=0.001$; Train before switch C1/S1, test after switch C1/S1: 54%, $p=0.001$). The effect was trending in aIT (98 neurons, Train before switch C1/S1, test before switch C1/S1: 56%, $p=0.087$; Train before switch C1/S1, test after switch C2: 54%, $p=0.032$; Train before switch C1/S1, test after switch C1/S1: 56%, $p=0.002$). There was no significant information about the previous task in PAR or Striatum, although this likely reflects a limited number of neurons recorded in each region.

We again thank the reviewer for suggesting this very interesting analysis and now include these results as a main figure panel (Fig. 4b), Extended Data Figure 11d and discuss it in the Results (lines 215-220), Discussion (lines 347-359) and Methods (lines 1402-1417):

Results:

So far, our results suggest the monkeys tracked whether the S1 or C1 task was in effect, and that this belief was maintained during the intervening C2 task. Consistent with this, the task classifier trained during the S1 and C1 tasks could decode the identity of the previous task during the subsequent C2 task (Fig. 4b-2, 62%, $p=0.001$, permutation test) and into the start of the next S1/C1 task (Fig 4b-3, 64%, $p=0.001$, permutation test). Similar results were seen in FEF, but not other regions (Extended Data Fig. 11d).

Discussion:

One reason we found shared representations (rather than task-unique ones) may have been because our task required the animal to continuously learn. Across trials, the monkeys used feedback to infer the task context and to flexibly map stimuli onto motor responses (Fig. 4-5; a form of ‘class-incremental’ continual learning¹⁹). As noted above, shared representations facilitate continual learning by allowing knowledge to generalize between tasks⁵⁸. In addition to reducing interference, scaling representations may also facilitate continual learning by constraining learning to those representations that are currently task relevant⁵⁹⁻⁶². Neural learning rules are activity dependent⁶³ and gated by reward⁶⁴ and so suppressing irrelevant representations may limit learning to task-relevant representations (addressing the credit assignment problem⁶⁵). Future work is needed to understand how suppressing irrelevant features supports continual learning and consider alternative mechanisms¹⁹, including replaying experiences across tasks⁶⁶⁻⁶⁸ and recalling context-specific associations from long-term memory⁴⁸.

While we observed information about the previous C1/S1 task persisted through the C2 task and may have influence behavior on the next C1/S1 block, it did not affect performance in the C2 task (see Extended Data Fig. 10b, reproduced below, for comparison of behavior). One hypothesis for why C2 avoids interference is that the brain suppresses the response on the opposite axis during each task. This would be a similar mechanism to the relative scaling of color and shape representations during C1/S1 task blocks but generalized to modulate the motor response.

b Effect of task sequence on C2 task performance

Extended Data Figure 10| b, Follows Fig. 4a. Comparison of average behavioral performance of both monkeys for S1-C2 and C1-C2 task sequences. Number of included blocks for S1-C2/C1-C2 sequences were 82/87, respectively.

To test whether we observe modulation of the motor axis representation, we examined the weights of the axis classifier to identify LPFC neurons that encoded the response axis. We then categorized the neurons into three groups based on their classifier weights during the response period (200ms to 450ms after

stimulus onset). Neurons with significant classifier weights for Axis 1 or Axis 2 were classified as Axis 1 or Axis 2 selective neurons, respectively (green and orange dots in Extended Data Fig. 14a, reproduced below). Neurons without significant classifier weights were classified as non-axis selective (grey dots in Extended Data Fig. 14a).

Although neurons were defined based on their activity during the motor response, they were still modulated during the baseline time period (Fig. 5g, reproduced below). Axis 1 selective neurons significantly increased their firing rate during Axis 1 trials and were suppressed during Axis 2 trials, even before stimulus onset (Fig. 5g, left). This pattern was reversed for Axis 2 selective neurons (Fig. 5g, right). Non-axis selective neurons showed no significant difference in firing rates during Axis 1 and Axis 2 trials (Extended Data Fig. 14b). Similar patterns were observed when we performed further controls (Extended Data Fig. 14c-e) and in other regions (Extended Data Fig. 14f-i). These results suggest the brain may actively suppress the irrelevant motor response, facilitating the task-relevant transformation (and inhibiting the task-irrelevant transformation).

Extended Data Figure 14| Neurons encoding response on Axis 1 and Axis 2 were suppressed when irrelevant to the task. **a**, Classifier weights for classifying axis of response for Fig. 5g. Classifier was trained to decode response axis using C1 and C2 task trials. Neurons with significant weights for Axis 1 and Axis 2 are denoted with green and orange dots, respectively. Non-selective neurons are denoted with gray dots. **b**, Follows Fig. 5g, time course of average firing rate for non-selective neurons in LPFC. Inset shows p-value of difference in firing rate when animal responded on Axis 1 compared to responding on Axis 2 (paired t-test). Lines and shading show mean \pm s.e.m. **c**, Follows Fig. 5g, time course of average firing rate of neurons when the animal responded on Axis 1 (dashed) or Axis 2 (solid). Neurons were grouped by their preferred response on Axis 1 (left) or Axis 2(right). Shaded blue box indicates the time period used to define the neurons preferred response axis (200ms-450ms after stimulus onset). **d**, Follows Fig. 5g. Average firing rate of LPFC neurons during trials when the animal responded on Axis 1 (green) and Axis 2 (pink). Three panels show responses of three categories of neurons: those that responded most strongly on Axis 1 (left), Axis 2 (middle), and were not selective (right) in LPFC. To characterize axis selectivity, classifier was trained to decode response axis using trials from S1 and C2 trials and then firing rate was estimated on withheld C1 trials. Inset shows p-value for of difference between firing rate in response on Axis 1 and response on Axis 2 trials in time periods -600ms-0ms (lower bracket) and -600ms-600ms (upper bracket) after stimulus onset (paired t-test). **e**, As in panel d, but axis selectivity was defined by training a classifier to decode response axis using trials from C1 and C2 trials and then firing rate was estimated on withheld S1 trials. **f-i**, As in Fig. 5g, but shown for **f**, FEF; **g**, aIT; **h**, PAR and **i**, Striatum. Similar to LPFC, Axis 1 and Axis 2 selective neurons in FEF, aIT and PAR were suppressed during response on opposite axis. Lines and shading show mean \pm s.e.m. firing rate across individual neurons for each region. Inset shows p-value for of difference between firing rate in response on Axis 1 and response on Axis 2 trials in time periods -600ms-0ms (lower bracket) and -600ms-600ms (upper bracket) after stimulus onset (paired t-test). **j**, As in Fig. 5h, but shows classifier accuracy in decoding the response axis for all brain regions. Darker to lighter color show progression in trial blocks of 10 shifted by 3 trials from switch trial in C1 task. Lines show mean classification accuracy after stimulus onset for 250 iterations of classifiers. Arrows indicate the time course of example trials. **k**, Classifier accuracy in decoding the response axis for all brain regions. Decoding using neural activity in -400ms to 0ms before stimulus onset period. Lines and shading show mean \pm s.e.m. classification accuracy across 250 iterations of classifiers.

Fig. 5| g, Average firing rate (FR) for Axis 1 selective (left) and Axis 2 selective (right) neurons in LPFC. C1 task trials were divided into two groups: response on Axis 1 (correct axis, green) and response on Axis 2 (incorrect axis, purple). Axis 1 selective neurons had higher activity before stimulus onset when the animal responded on Axis 1 (and vice versa when responding on Axis 2). Inset shows p-value of difference over entire time period (paired t-test)

Next, we were curious as to how quickly the response axis was suppressed after a switch. In other words, how quickly sensory information was routed to the correct motor response. As shown in Extended Data Figure 2g (reproduced to the right), the animals make few responses on the incorrect axis, suggesting they quickly infer the change in the response axis.

This rapid and reliable switch was reflected in the neural activity. To study this, we trained a classifier to decode the axis of response using neural activity during the fixation period (i.e., -400ms-0ms before stimulus onset). Training was restricted to the last 75 trials of the C2 and C1 tasks, when behavioral performance was high. We then applied the classifier to the beginning of the C1 task blocks to assess the representation of the response axis immediately after the switch. Consistent with behavior, the response axis was decodable immediately after the switch in LPFC (Fig. 5h, reproduced below). However, there was a slight, but significant, increase in classifier

Extended Data Figure 2| g, proportion of responses on incorrect axis of response for C1, C2 and S1 tasks.

Fig. 5| h, Classifier accuracy for response axis decoding in LPFC. Darker to lighter color show progression in trial blocks of 10 shifted by 3 trials from switch trial in C1 task in S1-C2-C1 task transition. Lines show mean classification accuracy after stimulus onset for 250 iterations of classifiers. Horizontal bar indicates significant increase in classifier accuracy from trial 10 to trial 16 (permutation test with cluster mass correction for multiple comparisons across time).

performance within 3-6 trials, indicating that the belief about the response axis is quickly updated ($\Delta(\text{performance in trial 10 and trial 16 in LPFC})=6.88\%$, $p=0.008$, permutation test). As shown in Extended Data Figure 14 j,k (reproduced below), the response axis was decodable in FEF and Striatum but did not significantly change as the animal discovered the task (and the response axis was not decodable in PAR and aIT during the fixation period, $\Delta(\text{performance in trial 10 and trial 73}): \text{Striatum}=-16.67\%$, $p=0.9750$, $\text{aIT}=-12.17\%$, $p=0.9670$, $\text{FEF}=2.76\%$, $p=0.2967$, $\text{PAR}=4.00\%$, $p=0.2098$, all one tailed permutation test). Altogether, these results suggest the brain supports task-specific transformations by scaling both sensory and motor representations. This would allow the task-relevant sensory to motor transformation to occur, while inhibiting the task-irrelevant transformation. Suppression occurred quickly after a switch in tasks, consistent with the animals' behavior. The suppression was seen throughout the entire trial, even before the stimulus onset, suggesting the brain is preparing certain eye movements and preventing others in order to ensure they perform the task correctly.

The details of the approach are included in the Methods (lines 1220-1268) and the results are included in the main text (lines 286-299) and Figure 5g-h and Extended Data Fig. 14:

Results:

In addition to selecting sensory information, the three tasks also require selecting the appropriate motor response: the animal must suppress responses on Axis 1 during the C2 task and Axis 2 during the C1 task. Consistent with scaling acting as a general mechanism for modulating task-

Extended Data Figure 14| j, As in Fig. 5h, but shows classifier accuracy in decoding the response axis for all brain regions. Darker to lighter color show progression in trial blocks of 10 shifted by 3 trials from switch trial in C1 task. Lines show mean classification accuracy after stimulus onset for 250 iterations of classifiers. Arrows indicate the time course of example trials. **k**, Classifier accuracy in decoding the response axis for all brain regions. Decoding using neural activity in -400ms to 0ms before stimulus onset period. Lines and shading show mean \pm s.e.m. classification accuracy across 250 iterations of classifiers.

(ir)relevant information, neurons selective for each motor axis were suppressed when the animal performed a task requiring a response on the other axis (Fig. 5g, see Extended Data Fig. 14c-e for further controls and Extended Data Fig. 14f-i for other regions). However, in contrast to the gradual suppression of sensory representations during task discovery, the incorrect response axis representation in LPFC was quickly suppressed after a change in task (within 3-6 trials, Fig. 5h, $\Delta(\text{classifier performance in trial 10 and trial 16})=6.88\%$, $p=0.008$, permutation test, see Extended Data Fig. 14a,k for other regions). This reflected the rapid inference of response axis seen in the animals' behavior (Extended Data Fig. 2g,h).

Results:

Together, these results suggest the monkeys' belief about the current task dynamically adjusted the magnitude of sensory and motor representations³⁶⁻³⁸. Scaling neural representations was a common mechanism used in all the three tasks and could facilitate task-relevant associations between stimulus category and motor response, while preventing task-irrelevant associations.

And in the Discussion (lines 339-346):

Discussion:

Given that the animals were trained for months on the tasks, one might expect the brain to form independent task representations in order to reduce interference^{4,51,56}. Instead, we found the brain reduced interference by engaging task-relevant sensory and motor representations and suppressing task-irrelevant dimensions^{36,42,57}. Importantly, the degree of engagement/suppression depended on the animals' internal belief about the task state (Fig. 5), allowing the brain to flexibly modulate how color and shape category influenced behavior and ensuring the animal performed the appropriate response (e.g., responding on Axis 1, not Axis 2, during C1 and vice versa for C2).

We also highlighted in the Discussion how suppression could mitigate interference:

Discussion (lines 339-342):

Given that the animals were trained for months on the tasks, one might expect the brain to form independent task representations in order to reduce interference^{4,51,56}. Instead, we found the brain reduced interference by engaging task-relevant sensory and motor representations and suppressing task-irrelevant dimensions^{36,42,57}.

And may support continual learning:

Discussion (lines 347-359):

One reason we found shared representations (rather than task-unique ones) may have been because our task required the animal to continuously learn. Across trials, the monkeys used feedback to infer the task context and to flexibly map stimuli onto motor responses (Fig. 4-5; a form of 'class-incremental' continual learning¹⁹). As noted above, shared representations facilitate continual learning by allowing knowledge to generalize between tasks⁵⁸. In addition to reducing interference, scaling representations may also facilitate continual learning by constraining learning to those representations that are currently task relevant⁵⁹⁻⁶². Neural learning rules are activity dependent⁶³ and gated by reward⁶⁴ and so suppressing irrelevant representations may limit learning to task-relevant representations (addressing the credit assignment problem⁶⁵). Future work is needed to understand how suppressing irrelevant features supports continual learning and consider alternative mechanisms¹⁹, including replaying experiences across tasks⁶⁶⁻⁶⁸ and recalling context-specific associations from long-term memory⁴⁸.

We thank the reviewer for suggesting these analyses as we think they have added new insight into the neural mechanisms of task-switching.

Comment 3.5: The paper uses the terminology of ‘learning’ a task (.g. line 174). This might be confusing since the task had been trained exhaustively. To avoid this confusion the term ‘apparent learning’ was proposed in the context of contextual learning (Heald et al 2021). It might be useful for the community to settle for an unambiguous vocabulary.

We thank the reviewer for raising this point. Honestly, we debated on several different terms and shared the reviewer’s concern about the term “learning” but felt it had the advantage of being simple. We appreciate the reviewer’s suggestion and have edited the manuscript to highlight several of the terms that have been used in this regard, including “apparent learning”, “task inference”, and “task discovery”. These terms are introduced with the topic in lines 61-63:

Results:

The animals were not instructed as to the identity of the new task. So, they had to learn which task was in effect on each new block. Note, that this learning was not de novo but reflected a process of discovering the current task (also known as “apparent learning”²⁰ or “task inference”²¹).

To maintain consistency from that point on in the manuscript, we settled on using a single term “task discovery”. We felt this term was the most approachable to a wide audience and did not imply mechanism (such as inference might). We hope this does not add to the plethora of terms used in the field.

Comment 3.6: The paper states that block changes were triggered by a performance level reaching 75%. As the paper also points out, some of the trials are easier to perform: stimuli with congruent properties (colour and shape indicating same choice). According to the methods these trials cover 20% of trials. As a consequence, performing only these perfectly while performing the rest at chance could account for 60% performance level. It would therefore important to see how the correct trials are distributed between congruent and incongruent stimulus property trials.

This is an important point. Motivated by the author’s comment, we plotted the animals’ performance on each stimulus condition for both congruent and incongruent stimuli (Extended Data Fig. 2a, reproduced below). As seen in the figure, the animals were above chance on all of the conditions ($p < 0.001$, Binomial test). This is now noted in the manuscript (lines 73-76):

Results:

While overall behavioral performance was higher on C2 than C1 trials (Fig. 1e), this difference was likely due to the increased uncertainty about the task on C1 trials; there was no difference in performance for congruent trials or when the shape was ambiguous (Extended Data Fig. 2a-d).

Extended Data Figure 2| Behavior varied for congruent and incongruent stimuli. Follows Fig. 1e. a, Psychometric curve for S1 (left), C2 (middle), and C1 (right) tasks for incongruent (dashed) and congruent (solid) stimuli. *, **, and *** indicate $p \leq 0.05$, 0.01, and 0.001, respectively, binomial test. Since congruency is undefined for the 50% morph line, there are no corresponding data points available.

Comment 3.7: Task identity is identified in neuronal responses through CPD. As later analysis indicates there might also be suppression of responses to irrelevant stimulus properties. Such suppression can actually account for the task identity information as the modulation is task-specific. This might not be the central topic of the paper, but the reader might be confused by the apparent wealth of information present in a humble neural population. The relationship between task representation and stimulus representation can distinguish sensory and higher cortices (e.g. Hajnal et al (2023) biorxiv). The suppression identified in the paper defines a two-dimensional subspace for task representation (along the shape decoder vector and along the colour decoder vector), and it remains unclear if task representation is different from this suppression-related task-specific modulation

We thank the reviewer for raising another interesting question. To minimize the effect of the stimulus response on the task representation, we measured the task representation before the onset of the visual stimulus (Fig. 4c). As expected, compression effects were observed after stimulus onset (Fig. 5c,d). This suggests task representations and the modulation of stimulus information occur at different times during the trial. That isn't to say the two are not related –the strength of the task representation was related to the compression index (Fig. 5e).

We have revised the Methods to more clearly state the motivation for measuring the task representation before the onset of the visual stimulus (lines 1025-1027):

Methods:

To minimize the effect of neural response to visual stimulus, the classifier was trained on neural activity from the fixation period (i.e., before stimulus onset).

We should also note that the coefficient of partial determination (CPD) statistic helps ameliorate this concern. As it measures how much additional explained variance is captured by adding a term, if the neuron was responding solely to the stimulus or response then it would not show a significant change in explained variance with the addition/subtraction of the task term. Therefore, our analyses suggest there is an independent effect of task on the neural activity. Nevertheless, we agree with the reviewer that it is hard (impossible?) to distinguish the encoding of the task *per se* from the effect of the task on other representations. This is almost a philosophical point about the nature of a representation versus its effect on the system. We now note this in the Methods (lines 834-839):

Methods:

Note, because the CPD statistic estimates the additional explained variance captured by each term it controls for potential covariation between terms. Nevertheless, it is important to note that, as with all neurophysiological studies, it is difficult to distinguish between the encoding of a specific cognitive variable (e.g., task) and its effect on the representation of other cognitive variables (e.g., the observed suppression of sensory and/or motor representations).

Similar to the findings reported in Hajnal et al (2023) and Flesch et al (2020), we find task-dependent compression of stimulus features in associative areas (LPFC, FEF and PAR) but no compression in sensory areas (aIT). Extended Data Figure 13 (reproduced below) shows the compression index for FEF, PAR, aIT. As illustrated in the figure, the compression index in FEF and PEAR is similar to the lateral prefrontal cortex (LPFC), with greater separation of color representations during C1 and C2 tasks and greater separation of shape representations during S1 tasks. In contrast, the compression index in aIT was close to zero, aligning with previous findings (Flesch et al (2020)) that sensory regions do not show task-dependent compression of features. Finally, it is important to note that the striatum did not show significant color or shape information, and so the compression index is not reliable.

These results are now included in the manuscript. We mentioned them briefly in the main text (lines 279-282) and they are detailed in the Extended Data Figure 13 legend.

Results:

There was greater separation of color representation during C1 and C2 tasks, and greater separation of shape representations during the S1 task (Fig. 5c; similar results were found when controlling for motor response, Extended Data Fig. 13a,b). Stimulus representations were also scaled in FEF and PAR, but not aIT, consistent with aIT maintaining a veridical representation of the stimulus (Extended Data Fig. 13a-c)²⁵.

C Similar to LPFC, irrelevant task dimensions are compressed in FEF and PAR

Extended Data Fig. 13| Follows Fig. 5c. Compression index of neural responses in FEF, aIT and PAR over time, for all three tasks. Compression index is taken as the log of the ratio of the separability of stimuli in color and shape subspaces using classifier encoding axis. The compression index in FEF is similar to the lateral prefrontal cortex (LPFC), with greater separation of color representations during C1 and C2 tasks and greater separation of shape representations during S1 tasks. Compression was reduced in PAR overall but showed a similar pattern with greater separation of color in the C2 task and shape in the S1 task (C1 was less clear). The compression index in aIT was close to zero for all tasks. This is consistent with previous findings⁷² that sensory regions show limited task-dependent compression of features. Striatum did not show significant color or shape information, and so the compression index is not reliable.

Comment 3.8: Line 201: 'Shared color representation increased during the C1 task (...)' It would be useful to disentangle the referred enhanced representation of C1 from a different effect: more consistent performance in the task (which adds stimulus-consistent decision-related activity). This comment relates to Major #3 above.

We agree with the reviewer that it is important to ensure the changes in classifier performance are due to differences in behavior performance. For this reason, we trained the color classifier on correct trials from the C2 task and tested it on correct trials from C1 task. The number of test trials and their stimulus congruency type was kept constant across all trial windows. This ensured that decision-related activity remains consistent across trial windows. Moreover, as shown in Extended Data Figure 8a, response direction was not cross-decodable between C1 and C2 tasks, suggesting any change in the representation is due to changes in the amount of color information.

Similarly, we trained a classifier to decode shape category using a balanced number of correct and incorrect S1 task trials. This classifier was then tested on correct trials of the C1 task that was used in the color classifier explained above (4 trials, 1 for each stimulus congruency type), in each window of 40 trials (slid every 5 trials). Therefore, even though the overall behavioral performance of the animal increases, their performance on the analyzed trials was kept the same across task discovery.

We apologize for not making this clearer and have revised the Methods to make this more clear (lines 1064-1068):

Methods:

The classifier was trained on only correct trials and the training data was balanced for congruent and incongruent stimuli (16 trials: 4 trials for each of the four stimulus congruency types). This

ensured an equal number of correct trials for each congruency type across all trial windows, controlling for motor response activity during the task discovery period.

Comment 3.9: Mixed selectivity is a key aspect in lPfc and continual learning paradigms distinguish between solutions relying on disjunct and overlapping population the neuron population. Illuminating this aspect of the population responses would be important.

We thank the reviewer for raising another interesting question. To study whether the brain used disjunct or overlapping populations of neurons to represent different task variables, we used a GLM to test whether the same neurons encoded multiple task variables. We found significantly overlapping populations of neurons for the majority of the task variables (Extended Data Fig. 4, reproduced below).

In particular, we observed a significant overlap in the neurons encoding the response direction and color category (Extended Data Fig. 4c, $p(\text{response direction-color category})=0.0001$, permutation test) and a significant overlap in the neurons encoding response direction and shape category (Extended Data Fig. 4d, $p(\text{response direction-shape category})=0.0124$, permutation test). Similarly, the population of neurons encoding the color category and the shape category were overlapping (Extended Data Fig. 4j, $p(\text{shape category-color category})=0.0001$). And the neural population encoding reward overlapped with the population encoding response location, shape category, and color category (Extended Data Fig. 4a, f, and g, respectively; $p(\text{reward-response direction})=0.0001$, $p(\text{reward-color category})=0.0025$, $p(\text{reward-shape category})=0.0265$). Finally, while trending towards overlapping, we found more distinct populations of neurons encoded the task identity and reward as well as task identity and shape category ($p(\text{reward-task identity})=0.1085$, $p(\text{task identity-shape category})=0.0930$, permutation test).

These results confirm previous findings that neurons in our recorded regions respond conjunctively to multiple task variables⁴⁹. However, this conjunctive representation is structured²⁶, rather than a nonlinear mixed selective code that leads to a high-dimensional representation. Our findings demonstrate that this structured conjunctive representation contains shared subspaces for color category and response direction across multiple tasks.

Extended Data Figure 4| Individual neurons represent multiple task variables across regions. Each panel shows the proportion of neurons that significantly encode two task variables (left and middle columns) and encode both (right column). For all panels, *, **, and *** indicate $p \leq 0.05$, 0.01, and 0.001, respectively, all permutation tests. **a-d**, Response direction and **a**, reward, **b**, task identity, **c**, color category, and **d**, shape category. **e-g**, reward and **e**, task identity, **f**, color category, and **g**, shape category. **h-i**, task identity and **h**, color category and **i**, shape category. **j**, color category and shape category.

Again, we thank the reviewer for suggesting this analysis as we feel it has added value to the manuscript. These results are now mentioned in the manuscript (lines 91-93):

Results:

All five regions represented task-relevant cognitive variables^{23,24}, including the identity of the current task, the color and shape of the stimulus, the response direction, and whether a reward was received (Fig. 1h-1l). The identity of the task was consistently represented throughout the trial (Fig. 1j). After stimulus onset, information about the color category of the stimulus²⁵ (Fig. 1h) and shape category of the stimulus (Fig. 1i) was followed by information about the direction of the animal's response (Fig. 1l), and then the reward received (Fig. 1k, see Figs. 1m and S3 for average time course across all regions). Individual neurons were 'mixed-selective' in all of these regions, representing multiple task variables (Extended Data Fig. 4)^{26,27}.

In addition, as noted by the reviewer, these results could constrain the mechanism that supports continual learning. We have added this to our Discussion (lines 333-338):

Discussion:

Representations in prefrontal cortex generalized across tasks (Fig. 2)^{43,53}, although other regions, such as the hippocampus, are likely involved^{39,54}. Less generalization in parietal cortex, visual cortex, and striatum suggests neural representations are more task-unique in these regions. Differences between regions may allow the brain to use the complementary advantages of both sharing representations (generalizing learning) and task-unique representations (avoiding interference)^{26,55}.

References:

- Binz M, Dasgupta I, Jagadish AK, Botvinick M, Wang JX, Schulz E. Meta-Learned Models of Cognition. Behavioral and Brain Sciences. Published online 2023:1-38. doi:10.1017/S0140525X23003266
- Chizat, L., Oyallon, E., and Bach, F. (2018). On Lazy Training in Differentiable Programming in NeuIPS. arXiv, 1812.07956 <http://arxiv.org/abs/1812.07956>.<<http://arxiv.org/abs/1812.07956>>
- Flesch, T., Juechems, K., Dumbalska, T., Saxe, A., & Summerfield, C. (2022). Orthogonal representations for robust context-dependent task performance in brains and neural networks. Neuron, 110(7), 1258-1270.
- Hadsell, R., Rao, D., Rusu, A. A., & Pascanu, R. (2020). Embracing Change: Continual Learning in Deep Neural Networks. Trends in Cognitive Sciences, 24(12), 1028–1040. <http://doi.org/10.1016/j.tics.2020.09.004><<http://doi.org/10.1016/j.tics.2020.09.004>>
- Hajnal, M. A., Tran, D., Szabó, Z., Albert, A., Safaryan, K., Einstein, M., ... & Orbán, G. (2023). Shifts in attention drive context-dependent subspace encoding in anterior cingulate cortex during decision making. bioRxiv.
- Heald, J. B., Lengyel, M., & Wolpert, D. M. (2021). Contextual inference underlies the learning of sensorimotor repertoires. Nature, 600(7889), 489-493.
- Lake, B. M., & Baroni, M. (2023). Human-like systematic generalization through a meta-learning neural network. Nature. <https://doi.org/10.1038/s41586-023-06668-3><<https://doi.org/10.1038/s41586-023-06668-3>>
- Schwartenbeck, P., Baram, A., Liu, Y., Mark, S., Muller, T., Dolan, R., Botvinick, M., Kurth-Nelson, Z., & Behrens, T. (2023). Generative replay underlies compositional inference in the hippocampal-prefrontal circuit. Cell, 186(22), 4885-4897.e14. <https://doi.org/10.1016/j.cell.2023.09.004><<https://doi.org/10.1016/j.cell.2023.09.004>>
- Van de Ven, G. M., Tuytelaars, T., & Tolias, A. S. (2022). Three types of incremental learning. Nature Machine Intelligence, 4(12), 1185-1197.

We thank the reviewers for their time and for their enthusiasm for the manuscript. As we detail below, we have revised the manuscript to address all of the reviewers' concerns. In addition, we have revised the manuscript to address editorial comments and to match the formatting requirements. Those changes are detailed in the cover letter. Here, we describe how we addressed each reviewer's comments.

Reviewers' comments are provided in **black**. They are unedited. Our responses are in **blue**. We also include the relevant sections of text in our response, written in *red italics*. In the manuscript, we have highlighted changes addressing the reviewers' comments in **red**.

Referee #1 (Remarks to the Author):

This is an exceptionally responsive review - the authors really have gone the extra mile, both in relation to my comments and those of other reviewers. I basically buy most of the authors' replies - including to the more sceptical R2 so I don't have further concerns.

We thank the reviewer for taking the time to review our paper and for their feedback, which has greatly improved the manuscript.

Referee #2 (Remarks to the Author):

The authors should be commended for a strongly responsive set of revisions, including quite a few new analyses and an improved new Discussion. The improved contrast with prior literature and deeper exploration of several aspects of their data have successfully convinced this reviewer of the novelty and interest of their findings. I now support publication of this manuscript at Nature. I have only very minor remaining changes to suggest.

We thank the reviewer for taking the time to review our paper and for their previous comments, as they improved the manuscript.

Comment 2.1: In the rebuttals, the authors show that training RNNs with different initial conditions can lead to different solutions. (And presumably the same would be true with different curricula.) As I previously noted, it may therefore be that different animals find different solutions, or that small differences in training protocol might bias the system to different solutions. The manuscript cites relevant papers at several points, but doesn't address the central point directly: it's likely that multiple solutions to the task would be possible in the brain, and the solution these animals' brains found may reflect details of training, life experience, chance, and other factors. A sentence raising this possibility seems warranted.

We have revised the Discussion to make this point more clear:

Discussion (lines 322-324):

Performing multiple tasks can lead to either shared representations or task unique representations^{23,49}. Several previous studies have found different tasks use representations of sensory and motor information that are specific to that task⁴⁴⁻⁴⁵.

Discussion (lines 338-352):

Whether a network uses a shared or task-unique representation depends on training curricula³², initial conditions⁴⁸, and biological factors. Given that the animals were trained for months on the tasks, one might expect the brain to form independent task representations in order to reduce interference^{4,45,49}. Instead, we found the brain reduced interference by dynamically amplifying task-relevant sensory and motor representations and suppressing task-irrelevant dimensions (Figs. 4-5)^{31,37,50}.

One reason we found shared representations (rather than task-unique ones) may be because our task required the animal to continuously learn. Across trials, the monkeys used feedback to infer

the task context, updating their internal belief about the task, and flexibly mapping stimuli onto motor responses (Fig. 4-5; a form of 'class-incremental' continual learning¹⁶). This may allow the brain to flexibly interpolate through a range of behaviors: updating its representation of the current task along a task manifold⁴¹ in order to parametrically modulate how color category and shape category influence decision-making and to ensure the animal performed the appropriate response (e.g., responding on Axis 1, not Axis 2, during C1 and vice versa for C2).

Comment 2.2: L72: the manuscript says that task C2 was performed better because of its unique response axis. This is probably true, but if I understand correctly C2 was also performed twice as often as C1 or S1. This should be mentioned.

This is an important point. As the reviewer suggested, the animals did perform more blocks of the C2 task than the S1 or C1 tasks. This ratio was roughly 1:1:2 for S1:C1:C2 – during recordings the monkeys performed 268:2.77:5.4 blocks of the S1:C1:C2 tasks. However, because block switches depended on performance, the animals also switched away from the C2 task more quickly. This means that, overall, they did slightly less trials of the C2 task – during recording, the ratio was 560:558:301 trials for S1:C1:C2 task.

As suggested by the reviewer, we now include these details in the Methods:

Methods (line 679-681):

On average, animals performed of 560, 558 and 301 trials and 2.68, 2.77 and 5.4 blocks per day for the S1, C1 and C2 tasks, respectively.

And we have revised the main text to note the differences in the number of blocks of each task:

Results (line 71-74):

This was because the axis of response always changed between blocks and the C2 task was the only one to use Axis 2, reducing the animals' uncertainty about the task (and leading to more C2 blocks than C1/S1 blocks, although there were more trials of C1/S1 than C2, see Methods for details).

Comment 2.3: Typos: Figure 3b, “representation”; L284: “Kendell's”

Thank you, we have fixed these typos.